# An Optimization for Reducing the Size of an Existing Urban-like Monitoring Network for Retrieving an Unknown Point Source Emission

Hamza Kouichi[1], Pierre Ngae[1], Pramod Kumar[1], Amir-Ali Feiz[1], and Nadir Bekka[1,2]

[1]LMEE, Université d'Evry Val-d'Essonne, 40 Rue du Pelvoux 91020 Courcouronnes, France
[2]LSA, Université Saad Dahlab-Blida, 09130 Blida, Algérie

**Correspondence:** Pramod KUMAR (pramod.kumar@univ-evry.fr)

**Abstract.** This study presents an optimization methodology for reducing the size of an existing monitoring network of the sensors measuring polluting substances in an urban-like environment in order to estimate an unknown emission source. The methodology is presented by coupling the Simulated Annealing (SA) algorithm with the renormalization inversion technique and the Computational Fluid Dynamics (CFD) modeling approach. This study presents an application of the renormalization data-assimilation theory for optimally reducing the size of an existing monitoring networks in an urban-like environment. Performance of the obtained reduced optimal sensor networks is analyzed by reconstructing the unknown continuous point emission using the concentration measurements from the sensors in that optimized network. This approach is successfully applied and validated with 20 trials of the Mock Urban Setting Test (MUST) tracer field experiment in an urban-like environment. The main results consist in reducing the size of a fixed network of 40 sensors deployed in the MUST experiment. The optimal networks in the MUST urban region are determined which makes it possible to reduce the size of original network (40-sensors) to $\sim 1/3^{rd}$ (13-sensors) and to $1/4^{th}$ (10-sensors). Using measurements from the reduced optimal networks of 10 and 13 sensors, the averaged location errors are obtained 19.20 m and 17.42 m, respectively, which are comparable to 14.62 m from obtained from the original 40-sensors network. In 80% of trials with networks of 10 and 13 sensors, the emission rates are estimated within a factor of two to the actual release rates. These are also comparable to performance of the original network where in 75% of the trials, the releases was estimated within a factor of two to the actual emission rates.

## 1 Introduction

In case of an accidental or deliberate release of a hazardous contaminant in the densely populated urban or industrial regions, it is important to accurately retrieve the location and the intensity of that unknown emission source for the risk assessment, emergency response and mitigation strategies by the concern authorities. This retrieval of an unknown source in various source reconstruction methodologies is completely dependent on the contaminant's concentrations detected by some pre-deployed sensors in that affected or a nearby region. However, pre-deployment of these limited number of sensors in that region required an optimal strategy for the establishment of an optimized monitoring network to achieve maximum a priori information regarding state of emission. It is also required to correctly capture the data while extracting and utilizing information from a limited

and noisy set of the concentration measurements. The optimal monitoring networks for the characterization of the unknown emission sources in complex urban or industrial regions is a challenging problem.

The problem of a monitoring network optimization is complex and may consist a first deployment of the sensors, updating an existing network, reducing the size of an existing network, increasing the size of an existing network. These problems are independent and each one of them have its own requirements. The degree of complexity also depends on (i) the network type (mobile network deployed only on emergency, permanent mobile network, permanent static network), (ii) the scale (local, regional, etc.) and (iii) the topography of the area of interest (flat terrain without obstacles, complex terrain, cities, urban, industrial regions, etc.). It is important to note that the optimization also depends on the objective of a network design as reconstruction of an emitting source, analysis of the air quality, triggering of an alert, etc. This study is focused on reducing the size of an existing network at local scale in an urban-like terrain for source reconstruction.

This study presents an optimization methodology for reducing the size of an existing monitoring network in a geometrically complex urban environment. The measurements from a reduced optimal network can be used for the source term estimation (STE) of an unknown source in an urban region with almost the same level of source detection ability as the original network of a larger number of the samplers. The establishment of an optimal network required the sensor concentration measurements, along with the availability of meteorological data, atmospheric dispersion model, choice of a STE procedure and an optimization algorithm. These type of the networks can have great applications in oil and gas industries for the estimation of the emissions of the greenhouse gases (GHG) like methane. In order to utilize an inversion method to estimate the methane emissions, accurate measurements of the methane at a network of the high precision sensors, downwind of a possible source, is a prerequisite. However, these sensors may not be deployed in a large numbers due to their high cost. Alternatively, the low-cost sensors (which may not be as high precision) can rapidly be deployed specifically for collecting the initial measurements. Using these less accurate measurements and the proposed optimization methodology, a reduced optimal network can quickly be designed to provide the 'best' sensors positions for the deployment of high precision sensors to obtain the accurate methane measurements. These high precision measurements can be utilized in an inversion method to estimate accurate methane emissions. A similar and very useful application of the method proposed here can be applied for the estimation of the methane emissions from landfills.

Ko et al. (1995) showed that the optimization of sensors network is an NP-hard (i.e. Non-deterministic Polynomial-time hardness) problem, which means that it is difficult for an exhaustive search algorithm to solve all instances of the problem because it requires a considerable time. Various optimization algorithms have been proposed to find the best solution, but these methods are not applicable to all the cases especially for large size problems. To solve such problems, the metaheuristic algorithms are efficient. Some studies discussed the optimization of sensor distribution and number for gas emission monitoring, e.g. (Ma et al., 2013; Ngae et al., 2019). Ma et al. (2013) used a direct approach with the Gaussian dispersion model to optimize the sensors networks in homogeneous terrains. However, the present study utilizes an inverse approach by solving the adjoint transport-diffusion equation with the building-resolving Computational Fluid Dynamics (CFD) model for an urban environment. This methodological approach for an optimal monitoring network (i.e. coupling of the optimization algorithm, inverse tracers transport modeling and CFD) includes the geometric and flow complexity inherent in an urban region for the

optimization process. Recently, for a different application point of view, Ngae et al. (2019) has also described an optimization methodology for determining an optimal sensors network in an urban like environment using the available meteorological conditions only. The CFD computations also required a considerable amount of time to compute the flow and dispersion in an urban environment. However in order to apply the proposed methodology in an emergency situation for an area of interest in a

complex urban or industrial environment, an archive database of the CFD calculations can be established for a wide range of meteorological and turbulence conditions and can be utilized in the optimization process.

In this study, the Simulated Annealing (SA) stochastic optimization algorithm (Jiang et al., 2007; Abida et al., 2008; Abida and Bocquet, 2009; Saunier et al., 2009; Kouichi et al., 2016; Kouichi, 2017; Ngae et al., 2019, etc.) is utilized. The SA algorithm was designed in the context of the statistical physics. It incorporates a probabilistic approach to explore the search

space and converges iteratively to the solution. This algorithm is often used and recommended to solve the problems of sensors network optimization (Abida, 2010). The network optimization process consists of finding the best set of sensors that leads to the minimum of a defined cost function. A cost function can be defined as a regularized norm square of the distance between the measurements and forecasts which is also used for the STE (Sharan et al., 2012). In this study, two canonical problems are considered independently. (i) Optimization of the measuring network: the optimization consists of selecting the best positions of

the sensors among a set of potential locations. This choice is operated in a search space constituted by all the possible networks (of a specific size) and based on a cost function that describes quantitatively the quality of the networks. (ii) Identification of the unknown source: the STE is studied in the framework of a parametric approach. Here the challenge is to determine the parameters of the source (intensity and position) using the measurements from the sensors of an optimally designed network.

The reduced optimal networks are validated using an STE technique to estimate the unknown parameters of a continuous

point source. The STE problem for atmospheric dispersion events has been an important topic of much consideration as reviewed in Rao (2007); Hutchinson et al. (2017). Often, the source term is estimated using a network of static sensors deployed in a region. In inverse modeling process, the adjoint source-receptor relationship and concentrations and meteorological datasets are required for the STE. The adjoint source-receptor relationship is defined by an inverse computation of the atmospheric transport dispersion model (Pudykiewicz, 1998). This relationship is often affected by the nonlinearities in the flow-field by

building effects in complex scenarios arising in urban environments, where the backward and forward dispersion concentrations will not match. Various inversion methods can be classified in two major categories: probabilistic and deterministic. The probabilistic category treats source parameters as the random variables associated to the probability distribution. This includes the Bayesian Estimation Theory (Bocquet, 2005; Monache et al., 2008; Yee et al., 2014, etc.), Monte Carlo algorithms using Markov chains (MCMC) (Gamerman and Lopes, 2006; Keats, 2009, etc.) and various stochastic sampling algorithms (Zhang

et al., 2014, 2015, etc.). Deterministic methods use cost functions to assess the difference between observed and modeled concentrations and are based on an iterative process to minimize this difference (Seibert, 2001; Penenko et al., 2002; Sharan et al., 2012, etc.). Among the other approaches, advanced search algorithms like genetic algorithm (Haupt et al., 2006, etc.) or neural network algorithm (Wang et al., 2015, etc.) and other regularization methods (Ma et al., 2017; Zhang et al., 2017, etc.) have been used for the STE. In this study, we utilized the renormalization inversion method (Issartel, 2005) for the STE using

measurements from the optimal networks, which is deterministic in nature and does not require any prior information of the

source parameters. The renormalization inversion approach was successfully applied and validated for retrieval of an unknown continuous point source in flat terrain (Sharan et al., 2009, etc.) and also in urban-like environment (Kumar et al., 2015b). Initially, the renormalization inversion method was proposed to estimate emission of the distributed sources (Issartel, 2005). Sharan et al. (2009) and other studies have shown that this technique is also effective for estimating continuous point sources.

For these applications, the hypothesis of a linear relationship between the receptor and the source was assumed. For homogeneous terrains, the adjoint functions can analytically be computed based on the Gaussian solution of the diffusion transport equation to estimate a continuous point release. However, the flow-field in urban or industrial environments is quite complex and the asymmetry of the flow and the dispersed plume in urban regions is generated mainly by the presence of buildings and other structures. In general, the Gaussian models are unable to capture the effects of complex urban geometries on adjoint

sensitivities between source and receptors and also if dense gases are involved, the Gaussian distribution hypothesis fails. Recently, Kumar et al. (2015b, 2016) have extended the applications of the renormalization inversion technique to retrieve an unknown emission source in the urban environments, where a CFD approach was used to generate the adjoint receptors-source relationship. In this process, a coupled CFD-renormalization source reconstruction approach was described for the identification of an unknown continuous point source located at the ground surface or at a horizontal plane corresponding to a known or

predefined altitude above the ground surface, or an elevated release in an urban area.

This study deals with a case of optimally reducing the size of an existing monitoring network. For this purpose, a predefined network of sensors deployed in an area of interest is considered to determine an optimized network with smaller number of sensors, but, with comparable information. This work explores with two requirements of the optimal networks that modifies the spatial configuration of an existing network by moving the sensors and also reduces the number of sensors of an existing large

network. In real situation, this methodology can be applied for the optimization of mobile networks deployed in emergency situation. The methodological approach to optimally reduce the size of an existing monitoring network in urban environment is presented by coupling the SA stochastic algorithm with the renormalization inversion technique and the CFD modeling approach. The concentration measurements obtained from the optimally reduced sensors networks in 20 trials of the Mock Urban Setting Test (MUST) field tracer experiment are utilize to validate the methodology by estimating an unknown continuous

point source in an urban-like environment.

## 2   Source Term Estimation Method: The Renormalization

In the context of an inversion approach, source parameters are often determined using the concentration measurements at the sensor locations and a source-receptors relationship. The release is considered continuous from a point source located at the ground or at a horizontal plane corresponds to an altitude of a known source height. Since the optimization methodology

presented in the next section utilizes some concepts from the renormalization inversion methodology (Sharan et al., 2009), the renormalization theory to estimate a continuous point release is briefly presented in following subsections.

## 2.1 Source-Receptor relationship

A source-receptor relationship is an important concept in the source reconstruction process and it can be linear or nonlinear. This study deals with the linear relationship, as except from the nonlinear chemical reactions, most of the other processes occurring during the atmospheric transport of trace substances are linear: advection, diffusion, convective mixing, dry and wet deposition, and radioactive decay (Seibert and Frank, 2004). A source-receptor relationship between the measurements and the source function is defined based on a solution of the adjoint transport-diffusion equation that exploits the computed adjoint functions (retroplumes) corresponding to each receptor (Pudykiewicz, 1998; Issartel et al., 2007, etc.). These retroplumes provide a sensitivity information between the source position and the sensor locations. Let's consider a discretized domain of $N$ grid cells in a 2-dimensional space $\mathbf{x} = (x, y)$, a vector of $M$ concentration measurements $\boldsymbol{\mu} = (\mu_1, \mu_2, ..., \mu_M)^T \in \mathbb{R}^M$, and an unknown source vector $\mathbf{s}(\mathbf{x}) \in \mathbb{R}^N$ to estimate. The measurements $\boldsymbol{\mu}$ are related to the source vector $\mathbf{s}$ by the use of sensitivity coefficients (also referred as adjoint functions) (Hourdin and Talagrand, 2006). The sensitivity coefficients describe the backward propagation of information from the receptors toward the unknown source. These vectors are related by the following linear relationship:

$$\boldsymbol{\mu} = \mathbf{A}\mathbf{s} + \boldsymbol{\epsilon} \tag{1}$$

where $\boldsymbol{\epsilon} \in \mathbb{R}^M$ is the total measurements error and $\mathbf{A} \in \mathbb{R}^{M \times N}$ is the sensitivity matrix with $\mathbf{A}(\mathbf{x}) = [\mathbf{a}(\mathbf{x}_1), \mathbf{a}(\mathbf{x}_2), ..., \mathbf{a}(\mathbf{x}_N)]$. Here, each column vector $\mathbf{a}(\mathbf{x}_i) \in \mathbb{R}^M$ of the matrix $\mathbf{A}$ represents the potential sensitivity of a grid cell with respect to all $M$ concentration measurements.

For a given set of the concentration measurements $\boldsymbol{\mu}$, the source estimate function $\mathbf{s}(\mathbf{x})$ in Eq. (1) can easily be estimated by formulating a constrained optimization problem. This optimization problem minimizes a cost function $J(\mathbf{s}) = \mathbf{s}^T\mathbf{s}$, subjected to a constraint $\boldsymbol{\epsilon} = \boldsymbol{\mu} - \mathbf{A}\mathbf{s} = 0$. Using the method of Lagrange multipliers, $\mathbf{s}(\mathbf{x})$ can be estimated as a least-norm solution:

$$\mathbf{s} = \mathbf{A}^T\mathbf{H}^{-1}\boldsymbol{\mu} \tag{2}$$

where $\mathbf{H}^{-1}$ is inverse of the *Gram matrix* $\mathbf{H} = \mathbf{A}\mathbf{A}^T$. This estimate (Eq. (2)) is not satisfactory because it generates artifacts at the grid cells corresponding to the measurement points. Adjoint functions become singular at these points and have very large values. These large values do not represent a physical reality, but rather an artificial information. This was highlighted by Issartel et al. (2007) which reduced this artificial information by a process of renormalization.

## 2.2 Renormalization process

This process involves a weight function in space $\mathbf{W}(\mathbf{x}) \in \mathbb{R}^{N \times N}$, which is purely a diagonal matrix with the diagonal elements $w_{jj} > 0$ such that $\sum_{i=1}^{N} w_{jj} = M$. Introduction of $\mathbf{W}$ transforms the source-receptor relationship in Eq. (1) to:

$$\boldsymbol{\mu} = \mathbf{A}_w\mathbf{W}\mathbf{s} + \boldsymbol{\epsilon}. \tag{3}$$

where the modified sensitivity matrix $\mathbf{A}_w$ is defined as $\mathbf{A}_w = \mathbf{A}\mathbf{W}^{-1} = [\mathbf{a}_w(\mathbf{x}_1), \mathbf{a}_w(\mathbf{x}_2), ..., \mathbf{a}_w(\mathbf{x}_N)]$ in which the column vector $\mathbf{a}_w(\mathbf{x}_i) = \mathbf{a}(\mathbf{x}_i)/w(\mathbf{x}_i)$ of $\mathbf{A}_w$ is the weighted sensitivity vector at $\mathbf{x}_i$. Considering a similar approach that outlined in

previous subsection, a new constrained optimization problem can be formulated for Eq. (3) to estimate $\mathbf{s}(\mathbf{x})$. This optimization problem minimizes a cost function $J(\mathbf{s}) = \mathbf{s}^T \mathbf{W} \mathbf{s}$, subjected to a constraint $\boldsymbol{\epsilon} = \boldsymbol{\mu} - \mathbf{A}_w \mathbf{W} \mathbf{s} = 0$, and deduces the following expression $\mathbf{s}_w$ of $\mathbf{s}$ (Appendix A in Kumar et al., 2016):

$$\mathbf{s}_w = \mathbf{A}_w^T \mathbf{H}_w^{-1} \boldsymbol{\mu} \tag{4}$$

where $\mathbf{H}_w^{-1}$ is the inverse of $\mathbf{H}_w = \mathbf{A}_w \mathbf{W} \mathbf{A}_w^T$.

The weight function in the above discussed renormalization process is computed by using an iterative algorithm demonstrated by Issartel et al. (2007) (Appendix A). A brief derivation for the estimation of an unknown point source (i.e. location and intensity) from the renormalization inversion is described in Appendix B.

## 3  The Combinatorial Optimization of a Monitoring Network

A predefined large network of $n$ sensors deployed in an area of interest is considered to determine an optimized network with smaller number of sensors, but with comparable information. For a given number of $p$ sensors such that $p < n$, one determines an array of $p$ sensors among $n$, which delivers maximum of the information. It is a combinatorial optimization problem that consists of choosing $p$ sensors among $n$, and thus constituting an optimal network. The optimal network will consist of $p$ sensors for which a defined cost function is minimum. The number of possible choices $^nC_p$ (number of combinations of $p$

among $n$) is very high, when an initial network is sufficiently instrumented ($n$ large) and $p$ is small with respect to $n$. As the number of combinations to be tested is very large, minimum of a cost function will be evaluated by a stochastic algorithm, viz. simulated annealing (SA).

### 3.1  Cost function

A cost function is defined (based on the renormalization theory) as a function that minimizes the quadratic distance between

the observed and the simulated measurements according to the $\mathbf{H}_w$ norm (Issartel et al., 2012), where $\mathbf{H}_w$ is the *Gram matrix* defined in a previous section 2.2. A cost function (say $J_s(\mathbf{x})$) to minimize is defined (Appendix C) as follows:

$$J_s(\mathbf{x}) = 1 - \frac{\mathbf{s}_w^2}{\boldsymbol{\mu}^T \mathbf{H}_w^{-1} \boldsymbol{\mu}} \tag{5}$$

where $\mathbf{s}_w$ is given by Eq. (4). A global minimum of the cost function $J_s(\mathbf{x})$ is evaluated by the SA algorithm.

### 3.2  Simulated Annealing (SA) algorithm for the Sensor's Network Optimization

The problem of optimization of a network is solved using the simulated annealing (SA) algorithm. The SA optimization algorithm is utilized here for the determination of the optimal networks by comparing its performance with the Genetic Algorithm (GA)(Kouichi, 2017). These algorithms of different search technics (SA probabilistic and GA evolutionary) are evaluated based on the same cost function. The results showed that the optimal networks retained by the GA and the SA are quantitatively and qualitatively comparable (Kouichi, 2017). The SA has advantageous because it is relatively easy to implement and takes smaller

computational time in comparison to GA. Both SA and GA optimization algorithms in the framework of this approach (based in the renormalization theory) has little influence on the estimation of the parameters of a source (Kouichi, 2017).

The SA is a random optimization technique based on an analogy with thermodynamics. The technique has been introduced to the computational physics over sixty years ago in the classic paper by Metropolis et al. (1953). The algorithm of simulated annealing is initiated by starting from an admissible network. At the subsequent steps, the system moves to another feasible network, according to a prescribed probability, or it remains in the current state. However, it is crucial to explain how this probability is calculated. The mobility of the random walk depends on a global parameter $T$ which is interpreted as 'temperature'. The initial values of $T$ are large, allowing free exploration of large extents of the state space (this corresponds to the "melted state" in terms of the kinetic theory of matter). In the subsequent steps, the temperature is lowered allowing the algorithm to reach a local minimum.

For the SA, each network is considered as a state of a virtual physical system, and the objective function is interpreted as the internal energy of this system in a given state. According to statistical thermodynamics, the probability of a physical system for being in a same state follows the Boltzmann distribution and depends on its internal energy and the temperature level. By analogy, the physical quantities (temperature, energy, etc.) become a pseudo-quantities. And during the minimization process, the probabilistic treatment consists to accept a new network selected in the neighborhood of the current network following the same Boltzmann distribution and depending both on the cost difference between the new and the current networks and on the pseudo-temperature ('temperature'). To find the solution, the SA incorporates the 'temperature' into a minimization procedure. So at high 'temperature' (starting 'temperature'), the space of solution is widely explored, while at lower 'temperature' the exploration is restricted. The algorithm is stopped when the cold 'temperature' is reached. It is necessary to choose the law of decreasing 'temperature', called as cooling schedule. Different approaches to parameterize the SA are explored in Siarry (2016). Kirkpatrick et al. (1983) proposed an average probability to determine the initial (starting) 'temperature'. Nourani and Andresen (1998) compared the most used cooling schedules (exponential, logarithmic, and linear). The SA algorithm starts minimization of an objective function at annealing 'temperature' from a single stochastic point, then it searches for the minimal solutions by attempting all the points in search domain with respect to their value of the 'temperature'. The algorithm is depicted in a flow diagram in Figure 2 and a step by step implementation of the SA procedure for an optimized monitoring network in an urban environment is described as follows:

**Step 1. Parameters setting and initialization**

*Network parameters* ($n$ and $p$): $n$ is the number of possible locations of the sensors and $p$ is the optimal network number of sensors.

*Starting 'temperature'* ($T_0$): $T_0$ is also called the highest 'temperature'. It was determined from the Metropolis law: $T_0 = -\frac{\overline{(\Delta J_s)}}{log(P_0)}$, where $\overline{(\Delta J_s)}$ is an average of the difference of cost functions calculated for a large number of cases. $P_0$ is an acceptance probability and following the recommendations of Kirkpatrick et al. (1983), it was set to 0.8. Start iterations ($I_{tt} = 0$).

*Length of the bearing* $(L_{max})$: A length of the bearing is the number of iterations to be performed at each 'temperature' level. An equilibrium is reached for this number of iterations and any significant improvement of the cost function can be expected. No general rule is proposed to determine a suitable length. This number is often constant and proportional to the size of the problem.

*The 'temperature' decay factor* $(\theta)$: The 'temperature' remains constant for $L_{max}$ iterations corresponding to each bearing. We used the exponential schedule due to its efficiency as denoted by Nourani and Andresen (1998). Then, the 'temperature' decreases law between two bearings varies as: $T_{b+1} = \theta T_b$, with $0 < \theta < 1$, where $b$ represents a bearing. So, it was retained a decay pattern by the bearings.

*The cold 'temperature'* $(T_{cold})$: $T_{cold}$ is often called the stopping 'temperature'. There is no clear rule to set this parameter. It

is possible to stop calculations when no improvement in the cost function is observed during a large number of combinations. One can estimate this number and take into account the maximum length $L_{max}$ of each bearing, thus the cold 'temperature' can be expressed as a fraction of the starting 'temperature' $T_0$.

*Assigning the first best set of sensors,* $\mathbf{x}_{Best} \leftarrow \mathbf{x}_{rand}(p,n)$: $\mathbf{x}_{rand}(p,n)$ corresponds to a vector of $p$ sensors locations randomly chosen among the $n$ possible locations. A new solution is randomly explored. This vector is assigned to the first 'best' set of

sensors.

**Step 2. Assigning a new set of sensors**

$\mathbf{x}_{new} \leftarrow \mathbf{x}_{rand}(p,n)$, where $\mathbf{x}_{rand}(p,n)$ corresponds to a vector of $p$ sensors locations randomly chosen among the $n$ possible locations. This vector is assigned to a new set $(\mathbf{x}_{new})$ of the sensors.

**Step 3. Cost difference**

Given a sensor location $\mathbf{x}_{new}$, the cost function $J_s(\mathbf{x}_{new})$ is computed as follows:

   - set $\boldsymbol{\mu}$ vector by using the measurements at the $\mathbf{x}_{new}$ locations,
   - set rows of matrix $\mathbf{A}$ using the sensitivity at the $\mathbf{x}_{new}$ locations,
   - determine $w(\mathbf{x})$, $\mathbf{H}_w$, and $\mathbf{a}_w$ iteratively using the algorithm in Eq. (A2),
   - compute the source term $\mathbf{s}_w(\mathbf{x})$ using Eq. (4),

- compute the cost function $J_s(\mathbf{x}_{new})$ using Eq. (5).

$J_s(\mathbf{x}_{best})$ is computed like $J_s(\mathbf{x}_{new})$ using the same precedent steps. A cost difference is then calculated using $\Delta J_s = J_s(\mathbf{x}_{new}) - J_s(\mathbf{x}_{best})$. Increment the iterations $(I_{tt} \leftarrow I_{tt} + 1)$.

**Step 4. Test of sign of $\Delta J_s$**

If $\Delta J_s < 0$, the error associated with $\mathbf{x}_{new}$ is less than that with $\mathbf{x}_{best}$ and thus $\mathbf{x}_{new}$ will become the next 'best network' (*Step*

6). If this condition is not satisfied, the algorithm can jump out of a local minimum (*Step* 5).

**Step 5. Conditional jump**

When $\Delta J_s > 0$, the algorithm has ability to jump out any local minima if condition: $P_{01} \leq \exp(-\frac{\Delta J_s}{T})$ is satisfied, where $P_{01}$ is the acceptance probability (a random number between 0 and 1), and $T$ is the current annealing 'temperature'. It means that $\mathbf{x}_{new}$ will be the next 'best network' even if the associated error is greater than that of $\mathbf{x}_{best}$. If $P_{01} > \exp(-\frac{\Delta J_s}{T})$, go to *Step 7*.

**Step 6. Update $\mathbf{x}_{best}$**

In this step, $\mathbf{x}_{best}$ is updated by $\mathbf{x}_{new}$.

**Step 7. Maximum iteration check**

If the maximum number of iterations of a bearing $(L_{max})$ is reached, a state of equilibrium is then achieved for this 'temperature' and one can cool the actual 'temperature' (*Step* 8). If not, continue iterations (*Step* 2).

**Step 8. 'Temperature' cooling**

'Temperature' is cooled using the cooling schedule and iteration variable is reset to zero.

**Step 9. Cold 'temperature' test**

The cold 'temperature' $(T_{cold})$ is also known as the stopping 'temperature'. If this 'temperature' is reached, the algorithm is stopped. When $T_{cold}$ is not reached, other 'temperature' bearing are performed using the cooling schedule.

**Step 10. Optimal network**

At this step, the last best network $\mathbf{x}_{best}$ is the optimal network. Source parameters are then estimated using the concentration measurements and retroplumes only for sensors from the obtained optimal network as: (i) $\mathbf{x}_0$ is estimated at position of the maximum of the source estimate function $\mathbf{s}_w(\mathbf{x})$, and (ii) the intensity $q_0$ is given by $q_0 = \mathbf{s}_w(\mathbf{x}_0)/w(\mathbf{x}_0)$.

In stochastic optimization algorithms, especially in the SA, it was observed that there is no guarantee for the convergence of the algorithm with such a strong cooling (Cohn and Fielding, 1999; Abida et al., 2008). However, chances are that a near-optimal network configuration can be reached. Due to this, one or more near-optimal networks can be obtained from this methodology that satisfy the conditions of near overall optimum condition.

## 4 The Mock Urban Setting Test (MUST) Tracer Field Network

The MUST field experiment was conducted by the Defense Threat Reduction Agency (DTRA) in 2001. It was aimed to help developing and validating the numerical models for flow and dispersion in an idealized urban environment. The experimental design and observations are described in detail in Biltoft (2001) and Yee and Biltoft (2004). In this experiment, an urban

canopy was represented by a grid of 120 containers. These containers were arranged along 12 rows and 10 columns on the army ground in the Utah desert, USA. Each container has dimensions of 2.54 m high, 12.2 m long and 2.42 m wide. The spacing between the horizontal lines is 12.9 m, while the columns are separated by a distance of 7.9 m. The total area thus formed is approximately $200 \times 200$ m$^2$. The experiment consists of 63 releases of a flammable gas (propylene $C_3H_6$) that is not

dangerous or harmful in quantities and could be released through the dissemination system into the open atmosphere (Biltoft, 2001). Different wind conditions (direction, speed, atmospheric stability) as well as different positions for gas emissions (inside or outside the MUST urban canopy at different heights) were considered. These gas emissions were carried out under stable, very stable, and neutral stability conditions. In this study, 20 trials in various atmospheric stability conditions are selected and the meteorological variables are taken from an analysis of meteorological and micro-meteorological observations in Yee and

Biltoft (2004) (Table 1). It is noted that the errors related to meteorological data can affects the accuracy of the source term estimation (Zhang et al., 2014, 2015), although this error is not considered in this study. In each trial, the gas was continuously released for $\approx 15$ min, during which the concentration measurements were made. These concentration measurements were carried out by 48 photoionization detectors (PIDs). 40 sensors were positioned on four horizontal lines at 1.6 m height (Figure 1) and 8 sensors were deployed in vertical direction at a tower located approximately in center of the MUST array.

**5    CFD Modelling for Retroplumes in an Urban Environment**

The flow-field in atmospheric dispersion models in geometrically complex urban or industrial environments cannot be considered as homogeneous throughout the computational domain. This is because the buildings and other structures in that region influence and divert the flow into unexpected directions. Consequently, the dispersion of a pollutant and computations of the adjoint functions (retroplumes) are affected by the flow-field induced by these structures in an urban region. Recently, Kumar

et al. (2015a) utilized a CFD model to compute the flow-field and the forward dispersion in 20 trials of the MUST field experiment. In order to reconstruct an unknown continuous point source, the computed flow-field is then used to compute the retroplumes for all selected trials (Kumar et al., 2015b). The CFD computations of the flow field presented in Kumar et al. (2015a) and retroplumes computed in Kumar et al. (2015b) are utilized in the proposed optimization methodology described in this study to obtain the optimal monitoring networks. In these studies, a CFD model fluidyn-PANACHE was utilized to

calculate the flow-field, considering a subdomain of calculation (whose dimensions are $250 \times 225$ m$^2$ with a height of 100 m) that consists the MUST urban array created by the containers, sources, receptors, and other instruments in this experiment. This subdomain is embedded in a larger computational domain (dimensions of $800 \times 800$ m$^2$ with a height of 200 m) to ensure a smooth transition of the flow between the edges of the domain and the obstacles zone. This extension of the outer domain far from the main experimental site is essential to reduce effects of the inflow boundary conditions imposed at inlet of the outer

domain. A more detailed description about the CFD model and its simulations for the MUST field experiment, e.g., boundary conditions, turbulence model, etc. are presented in Kumar et al. (2015a) and now briefly discussed in the Supplementary Information (SI). An unstructured mesh was generated in both domains with more refinement in the urbanized area in inner subdomain and at the positions of receptors, thus generating 2849276 meshes.

The simulations results with fluidyn-PANACHE in each MUST trial were obtained with inflow boundary conditions from vertical profiles of the wind ($U$), the turbulent kinetic energy ($k$) and its dissipation rate ($\epsilon$). These inflow profiles include: *(i) Wind profile*: Gryning et al. (2007) profiles in stable and neutral conditions and a profile based on the stability function by Beljaars and Holtslag (1991) in very stable conditions, *(ii) Temperature profile:* Monin-Obukhov similarity theory based logarithmic profiles, *(iii) Turbulence profiles:* $k$ and $\epsilon$ profiles are based on an approximate analytical solution of one-dimensional $k - \epsilon$ prognostic equations (Yang et al., 2009). The atmospheric stability effects in the CFD model fluidyn-PANACHE are included through the inflow boundary condition (via advection). fluidyn-PANACHE includes a Planetary Boundary Layer (PBL) model that serves as an interface between the meteorological observations and the boundary conditions required by the CFD solver. The observed turbulence parameters, e.g. (i) sensible heat flux ($Q_h$), the Obukhov length ($L$), (iii) surface friction velocity ($u_*$) and the temperature scale ($\theta_*$) were used to derive the vertical profiles of mean velocity and potential temperature. As an example, the wind velocity vectors around some containers for the trial 11 are shown in SI Figure S1.1. This figure shows the deviations in the wind speed and its direction due to the obstacles in an urban-like environment. It should be noted that the MUST experiment took place under neutral to stable and strongly stable conditions. However, the only atmospheric stability effects included in the CFD model are through the specification of inflow boundary conditions. Atmospheric stability has a profound impact on dispersion and would thus influences the adjoint functions. However, as presented and discussed in our previous study (Kumar et al., 2015a), even with specification of the stability dependent inflow boundary conditions only, the predicting forward concentrations from the CFD model are in good agreement with the measured concentrations for all 20 trials in different atmospheric stability conditions. However, at micro-scales also, small irregularities can break the repeated flow patterns found in a regular array of containers with identical shape (Qu et al., 2011). In addition, uncertainties associated with the thickness and the properties of the material of the container wall also affect flow pattern and the resulted concentrations and adjoint functions (Qu et al., 2011). Accordingly, the atmospheric stratification and stability effects should also be included through surface cooling or heating in the CFD model and stability effects from inflow boundary conditions. Since the released gas propylene is heavier than the air and would behave as a dense gas, a buoyancy model was used to model the body force term in the Navier-Stokes equations. The buoyancy model is suitable for the dispersion of heavy gases where density difference in the vertical direction drives the body force.

In order to compute the retroplumes in each MUST trial, firstly the CFD simulations were performed to compute the converged flow-field in computational domain, secondly the flow-field is reversed and used in the standard advection-diffusion equation to compute the adjoint functions $\mathbf{a}_i(\mathbf{x})$. In this computation of the retroplumes corresponding to each receptor in a selected trial, the advection-diffusion equation is solved by considering a receptor as a virtual point source with unit release rate at the height of that receptor. Also, the meteorological conditions remained invariant during the whole experimental period in a trial. The details about the retroplumes and the correlated theory of the duality verification (i.e. comparison of the concentrations predicted with the forward (direct) model and the adjoint model) for all 20 trials of the MUST field experiment are given in Kumar et al. (2015b) and we have utilized the same retroplumes in this study for the optimization process. Since we are concerned to establish an optimized monitoring network in a domain that contains the MUST urban array, the retroplumes are computed in the inner subdomain only. Consequently all the computations for an optimized monitoring network were carried

out in the inner subdomain only. The sensors in the optimized monitoring network are supposed to deploy on a fixed vertical height above the ground surface. Accordingly, the retroplumes corresponding to only 40 receptors at 1.6 m height were utilized in computations for the optimized monitoring networks in the MUST urban environment.

## 6 Results and Discussion

The calculations were performed by coupling the SA algorithm to a deterministic renormalization inversion algorithm and the CFD adjoint fields to optimally reduce the size of an existing monitoring network in an urban-like environment of the MUST field experiment. The network optimization process consists of finding the best set of sensors that leads to the lowest cost function. In this study, the validation is realized following two separated steps. The first step consists to form two optimal monitoring networks by using the presented optimization methodology which makes it possible to reduce the size of an original
network of 40 sensors to approx. one-third (13 sensors) and one-fourth (10 sensors). The second step consists to compare a posteriori performance of the obtained reduced size optimal networks with the 'MUST existing network' of 40 sensors at 1.6 m above the ground surface. In first step, a comparison (based on a cost function) with networks of the same size (e.g. 10 sensors) was performed implicitly during the optimization process. As the SA is an iterative algorithm, during the optimization process networks of same size are compared at each iteration and the 'best one' is retained. The networks have also been generated
randomly like in Efthimiou et al. (2017); however, the search space of the problem is very large. In our case, the number of the compared networks is equivalent to the number of iterations (as an example for optimal network of 10 sensors $\sim 3 \times 10^4$ configurations are compared). Here, the comparison is based on a cost function and inspired from the renormalized data assimilation method. The cost function quantifies the quadratic distance between the observed and the simulated measurements. The 'optimal network' produces the 'best' description of the observations (i.e. corresponds to the minimal quadratic distance)
and permits a posteriori to estimate the location and emission rate of an unknown continuous point source in an urban-like environment.

The size of the 'MUST predefined (original) network' is 40 sensors and the sizes of the optimized networks are fixed after performing a first optimization with the number of sensors from 4 to 16 (Kouichi, 2017). This first evaluation showed that for some trials, a small number of sensors could not allow to correctly reconstruct the source and divergences in the calculations
have been noted. Accordingly, the source estimation obtained for different trials and network sizes show that, very often, networks of less than 8 sensors may not characterize the source correctly. On the other hand, beyond 13 sensors, the source estimation is not significantly improved, and the associated errors were roughly constant (Kouichi, 2017). Therefore, in order to ensure an acceptable estimate of the source for all the trials, the sizes of the optimized network are fixed as 10 and 13 sensors $(1/4^{th}$ and $\sim 1/3^{rd}$ respectively of the original network of 40 sensors).
The optimization calculations were performed using Matlab on a computer with configuration "Intel® Core™ i7-4790 CPU @ 3.60 GHz and 16 GB RAM". The averaged computational time for optimization of one 10 sensors network was $\approx 2.5$ hrs and $\approx 8.5$ hrs for 13 sensors network. In computations, a value of parameter $T_0 = 10$ was fixed according to the scale of cost function and using the methodology described in *Step* 1 and $T_{cold} = 10^{-13}$ was used for both the optimal sensors networks. $\theta$ is

a decay factor of the 'temperature' for an exponential cooling schedule that describes a procedure of the temperature decrease. The best cooling schedule is the exponential decay as demonstrated by Nourani and Andresen (1998); Cohn and Fielding (1999). $\theta$ was fixed as 0.9 following the recommendation in literature (Siarry, 2014). This value allows a sufficiently slow cooling in order to give more chance to the algorithm to explore a large search space and to avoid the local minima. $L_{max}$ is taken as 100 & 200 for 10 & 13 sensors networks, respectively, following the recommendation in Siarry (2014) and according to number of the possible combinations that increases with the number of sensors ($8.5 \times 10^8$ for 10 sensors and $1.2 \times 10^{10}$ for 13 sensors).

Figure 3 shows the optimal networks of 10 and 13 sensors respectively for three representative trials 5 (very stable), 11 (neutral), and 19 (stable) in the MUST urban array. These three trials correspond to one trial each in neutral, stable, and very stable atmospheric conditions during the release. The optimal monitoring networks of 10 and 13 sensors for all selected 20 MUST trials are shown in SI Figures S2.1&S2.2.

In order to analyze the performance of the optimal monitoring configurations of smaller sizes, the source reconstructions were performed to estimate the unknown location and the intensity of a continuous point release. These source reconstruction results were obtained using the information from the optimal monitoring networks formed by 10 and 13 sensors in each MUST trial. In this performance evaluation process, the retroplumes and the concentration measurements were utilized from the sensors corresponding to these optimal networks. The retroplumes were computed using CFD simulations, considering the dispersion in a complex terrain. The source reconstruction results from both the optimal monitoring networks were also compared with results computed from the initial MUST network formed by 40 sensors (Kumar et al., 2015b). As in practice, the number of measurements is limited, this comparison allowed concluding that in urban areas, the optimally reduction of a networks size is possible without degrading significantly its efficiency for the source estimation.

Source estimation results from the different monitoring networks are shown in Table 2 for all 20 selected trials of the MUST experiment. These results are presented in terms of the location error ($E_l^p$), which is an euclidean distance between the estimated and the true source location, and $E_q^p$, a ratio of the estimated to the true source intensity. The corresponding monitoring network is represented by a superscript $p$ (representing the number of sensors in an optimal network) on $E_l^p$ and $E_q^p$. In order to quantify the uncertainty, a 10% Gaussian noise was added at each measurements. Accordingly, 50 simulations for the source reconstruction were performed with these noise measurements using the optimal networks for each trial. The average and the standard deviation of $E_l^p$ and $E_q^p$ are calculated and the results are also presented in Table 2.

For a given trial, a parameter *skeleton* represents the common sensors between two optimal networks of different sizes (with 10 and 13 sensors). These results exhibit that the SA algorithm coupled with renormalization inversion theory and CFD modeling approach has succeeded to reduce the size of an existing larger network to estimate the unknown emissions with almost similar accuracy in an urban environment.

Figure 4 shows isopleths of the renormalized weight function (also called as the visibility function) and the normalized source estimate function $\mathbf{s}_w^n(\mathbf{x}) = \mathbf{s}_w(\mathbf{x})/\max(\mathbf{s}_w(\mathbf{x}))$ correspond to both optimal monitoring networks for three representative trials (e.g. 5, 11, and 19) of the MUST experiment. These isopleths for all selected 20 MUST trials are shown in SI Figures S3. As already discussed in the literature, the visibility function includes the natural information associated with a monitoring

network for the source retrieval in a domain and physically interprets the extent of regions seen by the network (Issartel, 2005; Sharan et al., 2009). This function is independent of the effective values of the concentration measurements and depends only on geometry of the monitoring network. Hence, this leads to a priori information about the unknown source apparent to the monitoring network. A statistical parameter, factor of $g$ (FA$g$), for the source reconstruction results from each monitoring network is presented in Table 3, where FA$g$ represents the percentage no. of trials in which the source intensity is estimated within a factor of $g$ to the actual emission rates. The statistics calculated with original 40 sensors network show that the average location error for all 20 trials is 14.62 m, and in 75% of the trials, the intensity of the source is estimated within a factor of two to the actual emission rates. In 90% of the trials, intensity was estimated within a factor of three and within a factor of four in 95% trials (Table 3). If trial 2 is considered, large location errors (greater than 30 m) and the intensity values ranged between a factor of three to five, were observed (Table 2) independently of the number of sensors in the networks . If we consider the trials 15, 16, & 20, it was noted from the numerical results that the larger location errors do not necessarily correspond to the high intensity errors (Table 2).

From distribution of the optimized sensors in networks in Figures 3 for trials 5, 11 & 19 and SI Figures S2.1&S2.2 for all selected trials, it was noted that a larger number of sensors are close to the source position in the optimal networks in most of the trials. The source reconstruction results from the optimal monitoring networks formed by 10 sensors have an averaged location error ($E_l^{10}$) of 19.20 m for all 20 trials in the MUST experiment (Tables 2&3). In most of the trials, the location and the intensity of a continuous point emission are estimated accurately and close to the true source parameters. The location error is minimum in trial 14 ($E_l^{10}$ = 5.50 m) and maximum in trial 2 ($E_l^{10}$ = 56.88 m) (Table 2). For this configuration of the optimal sensors network, the source intensity in 80% of the trials are estimated within a factor of two to their true release rates (Tables 2&3).

For all 20 trials, the averaged location error $E_l^{13}$ is 17.42 m for the optimal networks formed by 13 sensors, which is smaller than the averaged $E_l^{10}$ = 19.20 m obtained with 10 sensors (Tables 2&3). The location error is observed minimum in trial 5 ($E_l^{13}$ = 2.13 m) and maximum in trial 16 ($E_l^{13}$ = 63.04 m) (Table 2). For this optimal network, in 80% of the trials, the source intensity is estimated within a factor of two to the actual emission rates. It was noted from the evaluation results that the increase in the number of sensors in a network has little influence on the accuracy of the estimated intensity (Tables 2&3).

In some trials, it was also noted that the distance of an estimated source to real source decreases with a decrease in sensors number and are also increases with the number of sensors in some other cases. It may be because the information added by a new sensor was not necessarily beneficial. As it is noticeable that in a particular meteorological condition (i.e. wind direction, speed and atmospheric stability), some of the sensors in a network may have little contribution to the STE. So, increasing the number of the sensors may not always provide the best estimation because with addition of the more no. of sensors, we also add more model and measurements errors in the estimation process. These errors can affect the source estimation results in some trials. In some cases, it may also depend on sensitiveness of the added sensor's position in an extended optimal network to the source estimation. It is also noted that for a monitoring network, not only the number of sensors but also the sensors distribution form (or sensor position) affect the information captured from network.

In fact, both optimal networks for each trial show a diversity of structures independently of the number of sensors considered. For this, the *skeleton* was used to analyze the heterogeneity of the structures of different optimal networks. A skeleton with 7 sensors is considered as a strong common base for the networks. This is the case for trials 3,6,14,15&20 (Table 2). It is noted that the overall results obtained are comparable (little differences between the results obtained by the networks). For these networks, a strong common base leads to a near global optimum. If we consider networks with a weak common base, the *skeleton* was formed of up to 3 sensors, particularly in trials 1 and 11. The performances do not systematically converge independently of the size of networks. Thus, for trial 1, better results were obtained with a network formed by 13 sensors compared to that by 10 sensors. This result reflects that the algorithm with the network formed by 13 sensors, converges probably toward a near global optimum. For trial 11 also, it was noted that the performances obtained by the two networks are identical. This shows that the networks with different sensors configurations may lead to a near overall optimum. This result is in coherence with Kovalets et al. (2011) and Efthimiou et al. (2017). Considering a network of 10 sensors, they shown for the same experimental data that the best source reconstruction is possible with only 5% or 10% of the total network combinations, randomly selected.

Considering the networks of intermediate structures, with *skeletons* varying from 4 to 6 sensors, notedly for trials 2,4,5,7,8,9, 12,13,16,17,18,&19, no obvious trend is noticed. These results tend to show that for a given trial, one or more optimal networks can satisfy the conditions of a near overall optimum (to be minimized). The obtained optimal networks may have a more or less common structure (having a greater or lesser number of *skeleton*).

Moreover, uncertainties calculated for different network sizes do not show an obvious trend. Indeed, a general relationship between the number of samplers and the uncertainties is not obvious. One notice that changing size of the network (increasing or decreasing the number of sensors) can lead to the growth or diminution of the uncertainties in the source parameters estimation. As an example, for Trial 7 uncertainties grow while for Trial 17 uncertainties diminish (Table 2).

It should be noted that this study deals with the case of reducing number of sensors in order to obtain an optimal network from an existing large network. This optimization was carried out under the constraints of an existing network of the original 40 sensors in the MUST field experiment. If one compares the performances of the obtained optimal monitoring networks of smaller sizes with the initial (original) network of 40 sensors in MUST environment, both optimal networks provide satisfactory estimations of unknown source parameters. The 40 sensors network gives an averaged location error of 14.62 m for all trials and the release rate were estimated within a factor of two in 75% trials. However, reducing the number of sensors to $\sim 1/3^{rd}$ from the original 40 sensors, the 13 sensors optimal networks also give comparable source estimations performance with an averaged location error of 17.42 m. Even with the 13 sensors optimal networks, source intensities in 80% trials were accurately estimated within a factor of two. Similarly for 10 sensors optimal networks, the averaged location error (=19.20 m) is slightly larger than that obtained from 13 and 40 sensors networks. However, reducing the number of sensors to $1/4^{th}$ gives extra advantages in case of the limited available sensors for a network in emergency scenarios of an accidental or deliberated releases in complex urban environments.

It should also be noted that the optimization evaluation in this study is performed using the MUST set of measurements and this makes it more likely that the resulting sensor configuration performs well reconstructing the source (that "the same

measurements shouldn't be used for the optimization and for the reconstructions"). However, this doesn't limit the application of the proposed methodology for some important practical applications like the accurate emissions estimation. In fact, this can be considered as a limitation of the used data for this application as for a complete process of the optimization and then the evaluation one requires a sufficiently long set of measurements so that the whole data can be divided into two parts: (i) first part for the designing an optimal sensor network and (ii) the second part for the evaluation of the designed optimal network. However, the durations of the releases in the MUST field experiment were not sufficiently large to divide the whole data from a test release separately into two parts for designing the optimal sensor network and then its evaluation. However, in further evaluation of the resulting optimal sensor configuration, a different set of the concentration measurements can be constructed by adding some noise to the measurements. For a continuous release in steady atmospheric conditions, the average value of the steady concentration in a test release is not expected to deviate drastically from the mean values in each segment of the complete data. So this new set of the concentration measurements with added noise can partially fulfill the purpose of the evaluation of a designed optimal network. As shown in Table 2, the errors in the estimated source parameters are small even with the new sets of concentration measurements constructed by adding 10% Gaussian noise. This exercise shows that even if we have utilized a partially different set of the measurements for the evaluation of the optimal networks, the optimal networks have almost the same level of the source detection ability in an urban-like environment. However, a realistic data is required for further evaluation of the optimization methodology.

Although the MUST field experiment has been widely utilized for validation of the atmospheric dispersion models and the inversion methodologies for unknown source reconstruction in an urban-like environment, its experimental domain was only approx. 200 m × 200 m (with buildings represented by a grid of containers) and can be considered small for a real urban environment. Thus, it may not quite represent a real urban region in terms of scale, meteorological variability, or non-uniform terrain or roughness/canopy structure. However, the methodology presented here is general in nature to apply to a real urban environment also. The methodology involves the utilization of the CFD model which generally can include the effects of the urban geometry, meteorological variability, or non-uniform terrain or roughness/canopy structure in a real urban environment. It is also to note that, the optimal network design would also depend on diurnal and spatial variability in meteorological conditions which may increase or decrease the optimum number of sensors and also may change the 'best positions' to be instrumented by sensors.

# 7   Conclusions

This study describes an approach for optimally reducing the size of an existing monitoring network of the sensors in a geometrically complex urban environment. It is a matter of reducing the size of networks while retaining the capabilities of estimating an unknown source in an urban region. Given an urban-like environment of the MUST field experiment, the renormalization inversion method was chosen for the Source Term Estimation. It was coupled with the CFD model fluidyn-PANACHE for generation of the adjoint fields. Combinatorial optimization by the simulated annealing consisted in choosing a set of sensors which leads to an optimal monitoring network and allows an accurate unknown source estimation. This study demonstrates

how the renormalization inversion technique can be applied to optimally reducing the size of an existing large network of the concentration samplers for quantifying a continuous point source in an urban-like environment with almost the same level of source detection ability as the original network with larger number of samplers.

The numerical calculations were performed by coupling the simulated annealing stochastic algorithm to the renormalization inversion technique and the CFD modeling approach to optimally reduce the size of an existing monitoring network in urban-like environment of the MUST field experiment. The optimal networks were constructed to reduce size of the original networks (40 sensors) to approx. one-third (13 sensors) and to one-fourth (10 sensors). The 10 and 13 sensors optimal networks have estimated the averaged location errors of 19.20 m and 17.43 m, respectively, and have comparable source estimations performance with an averaged location error of 14.62 m from the original 40 sensors network. In 80% of trials with optimal networks of 10 and 13 sensors, the emission rates are estimated within a factor of two to the actual release rates. These are also comparable to performance of the original 40-sensors network where in 75% of the trials, the releases was estimated with a factor of two to the actual release.

It was shown that in most of the MUST trials, the number of sensors in optimal networks slightly influences the location error of an estimated source and this error tends to increase as the number of sensors decreases. In 20 MUST trials, an analysis of the networks formed by 10 & 13 sensors revealed the heterogeneity of their structures in an urban domain. It was observed that for some trials, optimal networks had a strong common structure. This tends to prove that a certain number of sensors have a primordial role in reconstructing an unknown source. It would reflect a fact that the disjoint sets of sensors can lead to the best estimate of an unknown source in an urban region. This opens enormous prospects for assessing the relative importance of each sensor in a source reconstruction process in an urban environment. Defining a global optimal network for all meteorological conditions is a complex problem, but of greater importance that one may want to pursue. This challenge consists to define an optimal static network able to reconstruct the sources in all varied meteorological conditions. This information can be of great importance to determine an optimal monitoring network by reducing the number of sensors for characterization of the unknown emissions in the complex urban or industrial environments.

*Data availability.* The authors received access to the MUST field experiment dataset from Dr. Marcel Koñig of Leibniz Institute for Tropospheric Research. The MUST database was officially available from the Defense Threat Reduction Agency (DTRA) at https://must-dpg.dpg.army.mil/.

## Appendix A: Weight function

Issartel et al. (2007) demonstrated that a weight function, which reduces the artifacts of the adjoint functions at the measurement points, must verify the following renormalization criterion:

$$\mathbf{a}_w^T(\mathbf{x})\mathbf{H}_w^{-1}\mathbf{a}_w(\mathbf{x}) \equiv 1 \tag{A1}$$

Following an iterative algorithm by Issartel et al. (2007), $w(\mathbf{x})$ is determined as:

$$w_0(\mathbf{x}) = 1, \quad \text{and} \quad w_{k+1}(\mathbf{x}) = w_k(\mathbf{x})\sqrt{\mathbf{a}_{wk}^T(\mathbf{x})\mathbf{H}_{wk}^{-1}\mathbf{a}_{wk}(\mathbf{x})} \tag{A2}$$

## Appendix B: Identification of point source

Consider a point source of continuous release at a position $\mathbf{x_o} = (x_o, y_o)$ and with the intensity $q_o$. The point source is thus

expressed as a function of the preceding parameters: $\mathbf{s}(\mathbf{x}) = q_o \delta(\mathbf{x} - \mathbf{x_o})$. The relationship between the source and the measurements (Eq. (3)) becomes: $\boldsymbol{\mu} = q_o \mathbf{a}_w(\mathbf{x_o})w(\mathbf{x_o}) + \boldsymbol{\epsilon}$. By replacing the measurement term in Eq. (4), one obtains:

$$\mathbf{s}_w = q_o w(\mathbf{x_o})\mathbf{A}_w^T \mathbf{H}_w^{-1} \mathbf{a}_w(\mathbf{x_o}). \tag{B1}$$

$\mathbf{s}_w$ reaches its maximum at position $\mathbf{x_o}$ as the renormalization criterion (Eq. (A1)) is satisfied only at this position $\mathbf{x_o}$. Thus, $\mathbf{s}_w(\mathbf{x})$ at $\mathbf{x_o}$ becomes:

$$\mathbf{s}_w(\mathbf{x_o}) = q_o w(\mathbf{x_o}), \tag{B2}$$

which estimates the source intensity $q_o = \mathbf{s}_w(\mathbf{x_o})/w(\mathbf{x_o})$.

## Appendix C: Derivation of the Cost function

A cost function is defined (based on the renormalization theory) as a function that minimizes the quadratic distance between the observed and the simulated measurements according to the $\mathbf{H}_w$ norm (Issartel et al., 2012). $\mathbf{H}_w$ is the *Gram matrix* defined

in section 2.2. The quadratic distance between the real and the simulated concentration measurements according to the $\mathbf{H}_w$ norm is given by :

$$J = \|\boldsymbol{\mu} - \hat{\boldsymbol{\mu}}\|_{\mathbf{H}_w^{-1}}^2 = \frac{1}{2}\left[(\boldsymbol{\mu} - \hat{\boldsymbol{\mu}})^T \mathbf{H}_w^{-1}(\boldsymbol{\mu} - \hat{\boldsymbol{\mu}})\right] \tag{C1}$$

When considering a point source, $\hat{\boldsymbol{\mu}}$ is written by $\hat{\boldsymbol{\mu}} = q_o \mathbf{a}_w(\mathbf{x})w(\mathbf{x})$, where $q_o$ and $\mathbf{x}$ are respectively the intensity and the position of a point source. By replacing $\hat{\boldsymbol{\mu}}$ in Eq. (C1), one obtains (Sharan et al., 2012; Issartel et al., 2012):

$$J = J(q_o, \mathbf{x}) = \frac{1}{2}\left[(\boldsymbol{\mu} - q_o \mathbf{a}_w(\mathbf{x})w(\mathbf{x}))^T \mathbf{H}_w^{-1}(\boldsymbol{\mu} - q_o \mathbf{a}_w(\mathbf{x})w(\mathbf{x}))\right] \tag{C2}$$

For a fixed $\mathbf{x}$ in Eq. (C2), $J$ reaches a strict local minimum if following two conditions are satisfied:

$$\frac{\partial J(q_o, \mathbf{x})}{\partial q_o} = 0 \tag{C3}$$

$$\frac{\partial^2 J(q_o, \mathbf{x})}{\partial q_o^2} > 0 \tag{C4}$$

For each fixed $\mathbf{x}$, the first condition (Eq. (C3)) gives an estimate ($\tilde{q}_0$) of $q_0$ as: $\tilde{q}_0 = \frac{\mathbf{a}_w^T(\mathbf{x})\mathbf{H}_w^{-1}\boldsymbol{\mu}}{w(\mathbf{x})}$. The second condition (Eq. (C4)) is always satisfied as $\frac{\partial^2 J(q_o,\mathbf{x})}{\partial q_o^2} = w^2(\mathbf{x}) > 0, \forall\mathbf{x}$ (Sharan et al., 2012). Corresponding to the estimate $\tilde{q}_0$ from the first condition (Eq. (C3)), the cost function $J$ from Eq. (C2) leads to the following expression (Issartel et al., 2012):

$$J(\tilde{q}_0, \mathbf{x}) = \frac{\boldsymbol{\mu}^T\mathbf{H}_w^{-1}\boldsymbol{\mu}}{2}\left[1 - \frac{\mathbf{s}_w^2}{\boldsymbol{\mu}^T\mathbf{H}_w^{-1}\boldsymbol{\mu}}\right] \tag{C5}$$

where $\mathbf{s}_w$ is same as given in Eq. (4) and $\boldsymbol{\mu}^T\mathbf{H}_w^{-1}\boldsymbol{\mu}$ is a positive constant. Considering Eq. (C5), it is obvious that the minimization of $J$ also corresponds to the maximization of the term $\frac{\mathbf{s}_w^2}{\boldsymbol{\mu}^T\mathbf{H}_w^{-1}\boldsymbol{\mu}}$ or minimization of term $\left[1 - \frac{\mathbf{s}_w^2}{\boldsymbol{\mu}^T\mathbf{H}_w^{-1}\boldsymbol{\mu}}\right]$. Accordingly, the minimum value of the cost function $J$ in Eq (C5) leads to the following expression of the cost function (say $J_s(\mathbf{x})$) to minimize:

$$J_s(\mathbf{x}) = 1 - \frac{\mathbf{s}_w^2}{\boldsymbol{\mu}^T\mathbf{H}_w^{-1}\boldsymbol{\mu}} \tag{C6}$$

A global minimum of the cost function $J_s(\mathbf{x})$ is evaluated by the SA algorithm.

*Acknowledgements.* The authors would like to thank Dr. Marcel Koňig of Leibniz Institute for Tropospheric Research and the Defense Threat Reduction Agency (DTRA) for providing access to the MUST field experiment dataset. The authors gratefully acknowledge Fluidyn France for use of the CFD model fluidyn-PANACHE. We also thank to Dr. Claude Souprayen from Fluidyn France for useful discussions. Finally we thank to the reviewers Dr. J. Pudykiewicz, Dr. G.C. Efthimiou, one anonymous reviewer, and the topical editor Dr. Ignacio Pisso for their detailed and technical comments that helped to improve this study.

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

**Table 1.** The values of the meteorological (wind speed ($S_{04}$), wind direction ($\alpha_{04}$) at 4 m level of mast S), turbulence (the Obukhov length ($L$), friction velocity ($u_*$), turbulent kinetic energy ($k$) at 4 m level of tower T), and source parameters (source height ($z_s$), release duration ($t_s$), release rate ($q_s$)) in 20 selected trials of the MUST field experiment (Biltoft, 2001; Yee and Biltoft, 2004). Here, Trial Nos. 1-20 are assigned for just continuation and simplicity and these are not correspond to the same assigned trial no. for a given Trial name in the MUST experiment.

| Trial No. | Trial Name (JJJhhmm) | $q_s$ (l/min) | $t_s$ (min) | $z_s$ (m) | $S_{04}$ (m/s) | $\alpha_{04}$ (deg) | $u_*$ (m/s) | $L$ (m) | $k$ ($m^2s^{-2}$) |
|---|---|---|---|---|---|---|---|---|---|
| 1 | 2640138 | 175 | 21 | 0.15 | 2.35 | 17 | 0.26 | 91 | 0.359 |
| 2 | 2640246 | 200 | 15 | 0.15 | 2.01 | 30 | 0.25 | 62 | 0.306 |
| 3 | 2671852 | 200 | 22 | 0.15 | 3.06 | -49 | 0.32 | 330 | 0.436 |
| 4 | 2671934 | 200 | 15 | 1.8 | 1.63 | -48 | 0.08 | 5.8 | 0.148 |
| 5 | 2672033 | 200 | 15 | 1.8 | 2.69 | -26 | 0.17 | 4.8 | 0.251 |
| 6 | 2672101 | 200 | 14 | 0.15 | 1.89 | -10 | 0.16 | 7.7 | 0.218 |
| 7 | 2672150 | 200 | 16 | 0.15 | 2.30 | 36 | 0.35 | 150 | 0.409 |
| 8 | 2672213 | 200 | 15 | 1.8 | 2.68 | 30 | 0.35 | 150 | 0.428 |
| 9 | 2672235 | 200 | 15 | 2.6 | 2.32 | 36 | 0.26 | 48 | 0.387 |
| 10 | 2672303 | 200 | 19 | 1.8 | 2.56 | 17 | 0.25 | 74 | 0.367 |
| 11 | 2681829 | 225 | 15 | 1.8 | 7.93 | -41 | 1.10 | 28000 | 1.46 |
| 12 | 2681849 | 225 | 16 | 0.15 | 7.26 | -50 | 0.76 | 2500 | 0.877 |
| 13 | 2682256 | 225 | 15 | 0.15 | 5.02 | -42 | 0.66 | 240 | 0.877 |
| 14 | 2682320 | 225 | 15 | 2.6 | 4.55 | -39 | 0.50 | 170 | 0.718 |
| 15 | 2682353 | 225 | 15 | 5.2 | 4.49 | -47 | 0.44 | 120 | 0.727 |
| 16 | 2692054 | 225 | 22 | 1.3 | 3.34 | 39 | 0.36 | 170 | 0.362 |
| 17 | 2692131 | 225 | 17 | 1.3 | 4.00 | 39 | 0.42 | 220 | 0.582 |
| 18 | 2692157 | 225 | 15 | 2.6 | 2.98 | 43 | 0.39 | 130 | 0.505 |
| 19 | 2692223 | 225 | 15 | 1.3 | 2.63 | 26 | 0.35 | 120 | 0.484 |
| 20 | 2692250 | 225 | 17 | 1.3 | 3.38 | 36 | 0.37 | 130 | 0.537 |

**Table 2.** Source estimation results from the different monitoring networks for each selected trial of the MUST field experiment. $E_l^p$ and $E_q^p$ respectively denote the location error (m) and ratio of the estimated to true source intensity with the corresponding monitoring network. Here, the superscript $p$ on $E_l^p$ & $E_q^p$ represents the no. of sensors in an optimal network. *Skeleton* refers to the number of sensors common to the optimal networks of 10 and 13 sensors for a given MUST trial.

| Run No. | $E_l^{40}$ (m) | $E_l^{13}$ (m) | $E_l^{10}$ (m) | $E_q^{40}$ | $E_q^{13}$ | $E_q^{10}$ | *Skeleton* sensors |
|---|---|---|---|---|---|---|---|
| 1 | 3.3±1.3 | 19.60±12.13 | 33.76±5.30 | 0.92±0.08 | 1.04±0.23 | 1.24±0.22 | 3 |
| 2 | 42.9±23.8 | 31.91±8.80 | 56.88±9.51 | 4.01±1.57 | 3.21±0.41 | 5.12±3.63 | 4 |
| 3 | 10.8±1.6 | 9.01±2.47 | 9.01±3.02 | 1.17±0.27 | 0.71±0.16 | 0.71±0.16 | 7 |
| 4 | 22.8±7.7 | 18.07±1.84 | 18.07±2.61 | 0.27±0.35 | 0.83±0.21 | 0.83±0.26 | 6 |
| 5 | 21.9±2.1 | 2.13±2.54 | 11.56±4.21 | 0.57±0.07 | 0.95±0.05 | 0.67±0.05 | 6 |
| 6 | 5.0±1.6 | 6.96±0.19 | 6.96±0.00 | 2.14±0.60 | 1.04±0.06 | 1.04±0.04 | 7 |
| 7 | 12.4±9.1 | 18.85±9.08 | 12.99±1.67 | 0.41±0.49 | 3.11±0.51 | 1.06±0.07 | 4 |
| 8 | 15.8±12.1 | 12.86±1.28 | 15.79±1.05 | 2.22±0.90 | 1.32±0.34 | 1.76±0.11 | 6 |
| 9 | 7.7±1.2 | 8.20±0.35 | 8.08±0.00 | 1.37±0.07 | 3.06±0.17 | 7.55±0.39 | 5 |
| 10 | 8.8±3.0 | 8.00±4.57 | 8.00±5.68 | 1.08±0.19 | 1.08±0.77 | 1.08±1.07 | 8 |
| 11 | 19.8±5.0 | 17.19±12.00 | 17.19±7.06 | 1.67±0.12 | 1.62±0.40 | 1.62±0.26 | 3 |
| 12 | 7.4±6.6 | 5.43±11.69 | 10.22±9.10 | 0.95±0.06 | 0.85±0.28 | 0.20±0.04 | 4 |
| 13 | 7.7±0.6 | 8.63±4.36 | 8.63±3.86 | 0.97±0.07 | 0.78±0.18 | 0.78±2.05 | 4 |
| 14 | 2.2±1.9 | 5.50±2.98 | 5.50±3.88 | 1.42±0.17 | 0.88±0.24 | 0.88±0.40 | 7 |
| 15 | 1.1±1.0 | 30.23±2.14 | 37.98±0.72 | 1.88±0.09 | 0.57±0.07 | 0.17±0.01 | 7 |
| 16 | 26.7±4.9 | 63.04±6.84 | 29.80±9.86 | 1.70±0.06 | 0.29±0.06 | 0.67±0.23 | 5 |
| 17 | 7.0±1.9 | 14.07±2.78 | 23.05±10.44 | 0.90±0.05 | 1.10±0.04 | 1.52±0.16 | 6 |
| 18 | 14.3±11.0 | 12.83±4.18 | 12.83±4.61 | 1.15±0.46 | 1.15±0.16 | 1.15±0.21 | 6 |
| 19 | 22.3±6.4 | 10.77±4.25 | 13.46±4.8 | 1.76±0.16 | 0.99±0.20 | 0.83±0.25 | 6 |
| 20 | 32.5±1.8 | 45.23±1.78 | 44.29±0.31 | 0.83±0.04 | 1.68±0.06 | 1.56±0.06 | 7 |

**Table 3.** Statistics for the source reconstruction results from each monitoring network. Here, $E_l^p$ is the averaged location error for all 20 trials corresponding to each network. FA$g$ represents the percentage number of trials in which the source intensity is estimated within a factor of $g$.

| Sensors ($p$) | $E_l^p$ (m) | FA4 | FA3 | FA2 |
|---|---|---|---|---|
| 40 | 14.62 | 95 | 90 | 75 |
| 13 | 17.42 | 100 | 80 | 80 |
| 10 | 19.20 | 80 | 80 | 80 |

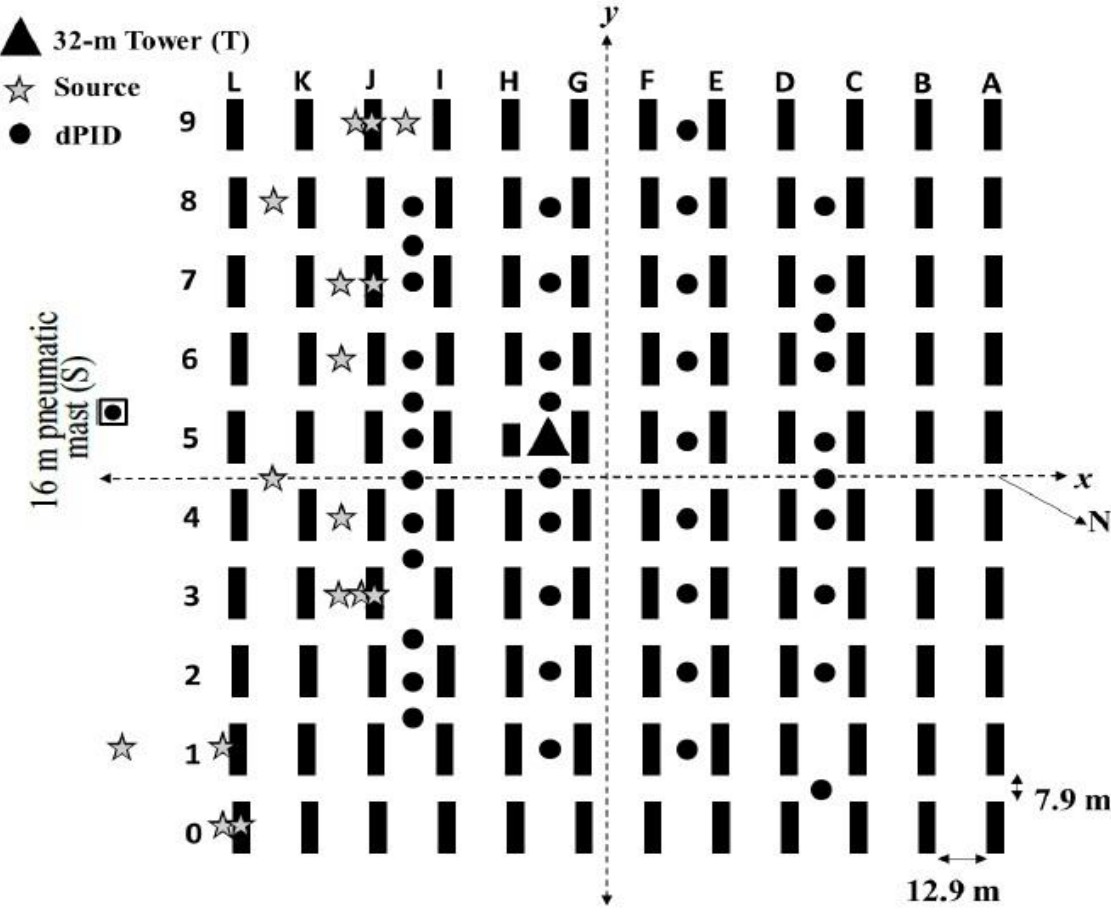

**Figure 1.** A schematic diagram of the MUST geometry showing 120 containers and source (stars) and receptors (black filled circles) locations. In a given trial - only one source was operational.

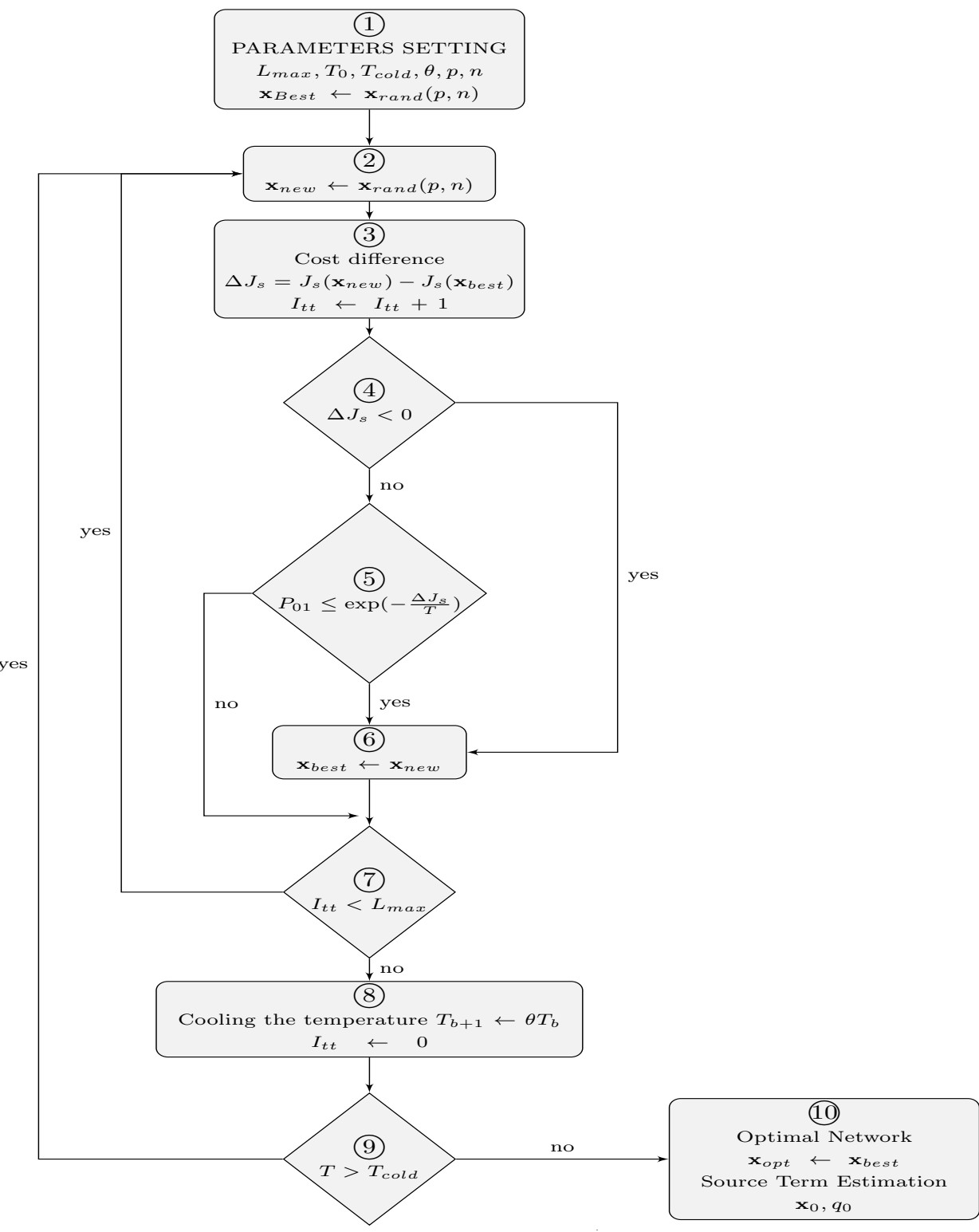

**Figure 2.** Flow diagram of the Simulated Annealing procedures to determine an optimized monitoring network.

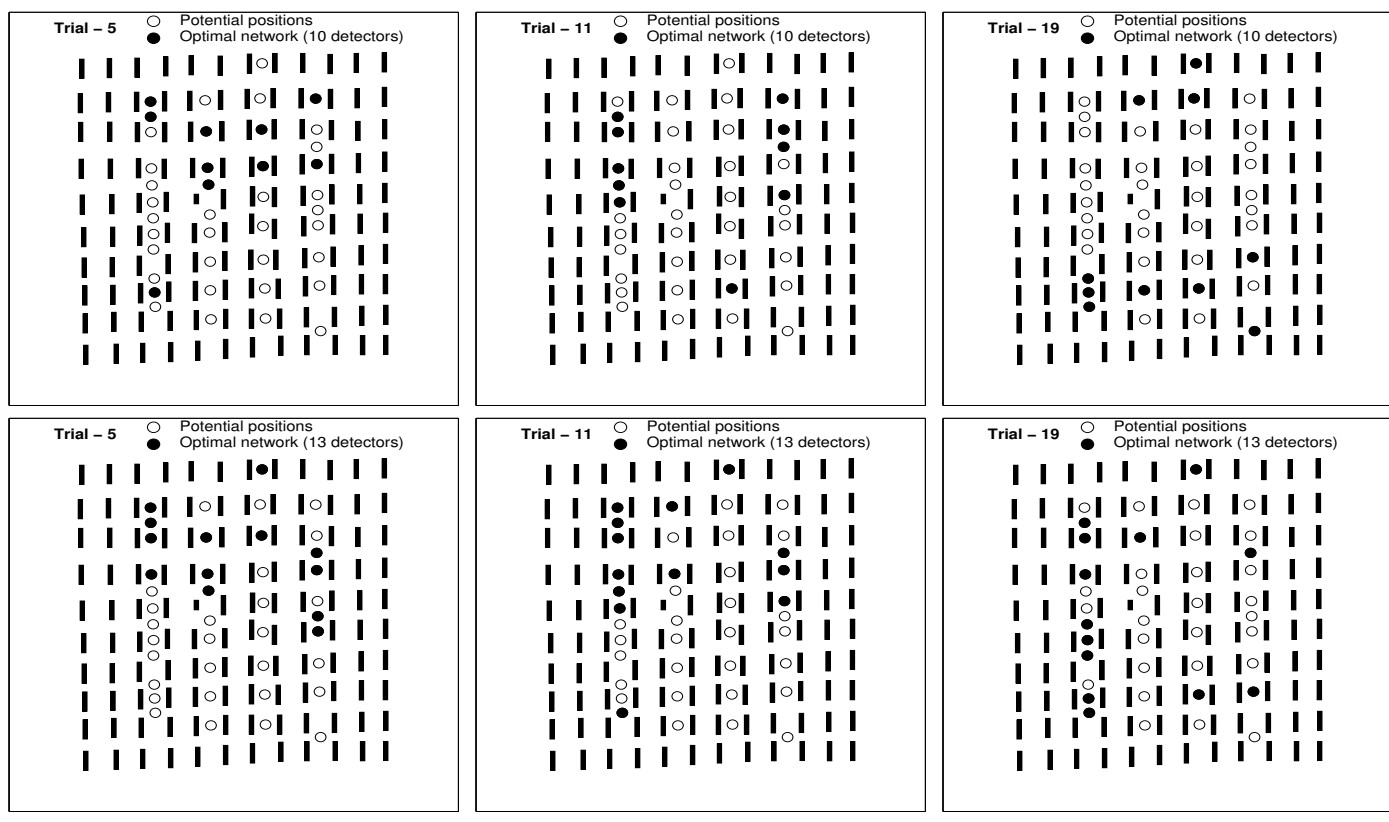

**Figure 3.** The optimal networks of 10 (first row) and 13 sensors (second row) respectively for trials 5 (very stable), 11 (neutral), and 19 (stable). Blank and filled black circles respectively represent the all (40) potential positions and the optimal positions of sensors.

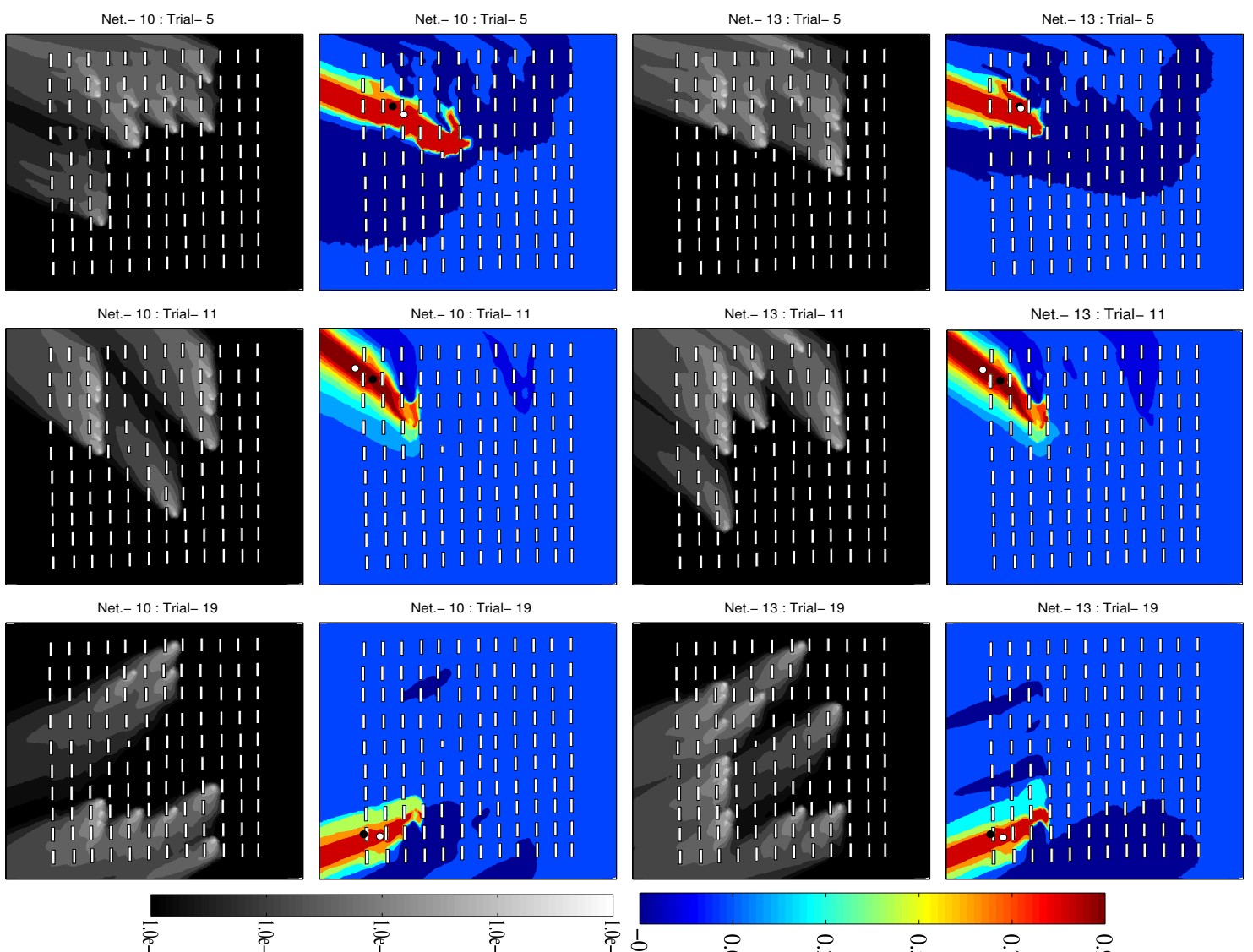

**Figure 4.** Isopleths of the renormalized weight function $(w(\mathbf{x}))$ (gray colored in first and third columns) and the normalized source estimate function $(\mathbf{s}_w^n(\mathbf{x}) = \mathbf{s}_w(\mathbf{x})/\max(\mathbf{s}_w(\mathbf{x})))$ (colored in second and fourth columns) for both optimal networks of 10 and 13 sensors respectively for trials 5 (very stable), 11 (neutral), and 19 (stable). The black and white filled circles respectively represent the true and estimated source locations.

# An Optimization for Reducing the Size of an Existing Urban-like Monitoring Network for Retrieving an Unknown Point Source Emission

Hamza Kouichi[1], Pierre Ngae[1], Pramod Kumar[1], Amir-Ali Feiz[1], and Nadir Bekka[1,2]

[1]LMEE, Université d'Evry Val-d'Essonne, 40 Rue du Pelvoux 91020 Courcouronnes, France
[2]LSA, Université Saad Dahlab-Blida, 09130 Blida, Algérie

**Correspondence:** Pramod KUMAR (pramod.kumar@univ-evry.fr)

**Abstract.** This study presents a methodology for the optimization of a an optimization methodology for reducing the size of an existing monitoring network of sensors measuring the the sensors measuring polluting substances in an urban-like environment with a view in order to estimate an unknown emission source. The methodology was is presented by coupling the Simulated Annealing (SA) algorithm with the renormalization inversion technique and the Computational Fluid Dynamics (CFD) modeling approach. Performance of an obtained optimal network was This study presents an application of the renormalization data-assimilation theory for optimally reducing the size of an existing monitoring networks in an urban-like environment. Performance of the obtained reduced optimal sensor networks is analyzed by reconstructing the unknown continuous point emission using the concentration measurements from the sensors in that optimized network. This approach was is successfully applied and validated with 20 trials of the Mock Urban Setting Test (MUST) tracer field experiment in an urban-like environment. The main results consist in reducing the size of a fixed network of 40 sensors deployed in the MUST experiment. The optimal networks in the MUST urban region are determined which makes it possible to reduce the size of original network (40-sensors) to $\sim 1/3^{rd}$ (13-sensors) and to $1/4^{th}$ (10-sensors). The Using measurements from the reduced optimal networks of 10 and 13 sensors optimal networks have estimated , the averaged location errors of are obtained 19.20 m and 17.42 m, respectively, which are comparable to 14.62 m from obtained from the original 40-sensors network. In 80% trials, emission rates with the of trials with networks of 10 and 13 sensors networks were , the emission rates are estimated within a factor of two which to the actual release rates. These are also comparable to 75% from performance of the original network . This study presents an application of the renormalization data-assimilation theory for determining the optimal monitoring networks to estimate a continuous point source emission in an urban-like environment where in 75% of the trials, the releases was estimated within a factor of two to the actual emission rates.

## 1 Introduction

In case of an accidental or deliberated release of a hazardous contaminant in the densely populated urban or industrial regions, it is important to accurately retrieve the location and the intensity of that unknown emission source for the risk assessment, emergency response and mitigation strategies by the concern authority authorities. This retrieval of an unknown source in various source reconstruction methodologies is completely dependent on the contaminant's concentrations detected by some pre-

deployed sensors in that affected or a nearby region. However, pre-deployment of these limited number of sensors in that region required an optimal strategy for the establishment of an optimized monitoring network to achieve maximum a priori information regarding state of emission. It is also required to correctly capture the data while extracting and utilizing information from a limited and noisy set of the concentration measurements. The optimal monitoring networks for the characterization of the

unknown emission sources in complex urban or industrial regions is a challenging problem.

The problem to optimize of a monitoring network is common and consists in optimization is complex and may consist a first deployment of the sensors, updating an existing network, reducing the size of a networkof sensors at the level of a city, county, or a neighborhood while retaining its properties. The positions of a small number of sensors are thus optimally determined so as to preserve the objectives of the initial monitoring network . These objectives are generally diverse, e.g. , an existing network, increasing the size of an existing network. These problems are independent and

each one of them have its own requirements. The degree of complexity also depends on (i) the network type (mobile network deployed only on emergency, permanent mobile network, permanent static network), (ii) the scale (local, regional, etc.) and (iii) the topography of the area of interest (flat terrain without obstacles, complex terrain, cities, urban, industrial regions, etc.). It is important to note that the optimization also depends on the objective of a network design as reconstruction of an emitting source, analysis of the air quality, triggering of an alert, etc. This study will be focused with an objective to reconstruct

an unknown continuous point source's release in an is focused on reducing the size of an existing network at local scale in an urban-like environment. Using the concentration measurements from an optimally monitoring sensors network , the determination of an unknown or fabricated pollutant emissions from some industrial and accidental releases can be useful for mitigation strategies and also to impose strict actions on such pollutant sources. terrain for source reconstruction.

This study presents a methodology for the sensor's locations choice, leading to the best network for the estimation of an unknown point point source an optimization methodology for reducing the size of an existing monitoring network in a geometrically complex urban environment.

This type of network is of great interest in case of an accidental or intentional pollutants release because this makes it possible to estimate the sources of pollution with limited number of measurements from an optimal sensors network . In these conditions, it is necessary to know the location, and the evolution of the spatial extent of the contaminant for an emergency response. The intensity, location and time of the release are often unknown and should be inferred from sensor measurements. The The measurements from a reduced optimal network can be used for the source term estimation (STE) from the measurements is an inverse modeling problem. of an unknown source in an urban region with almost the same level of source detection ability as the original network of

a larger number of the samplers. The establishment of an optimal network may require required the sensor concentration measurements, along with the availability of meteorological data, atmospheric dispersion model, choice of a STE procedure and an optimization algorithm. These type of the networks can have great applications in oil and gas industries for the estimation of the emissions of the greenhouse gases (GHG) like methane. In order to utilize an inversion method to estimate the methane emissions, accurate measurements of the methane at a network of the high precision sensors, downwind of a

possible source, is a prerequisite. However, these sensors may not be deployed in a large numbers due to their high cost. Alternatively, the low-cost sensors (which may not be as high precision) can rapidly be deployed specifically for collecting the initial measurements. Using these less accurate measurements and the proposed optimization methodology, a reduced optimal network can quickly be designed to provide the 'best' sensors positions for the deployment of high precision sensors to obtain the accurate methane measurements. These high precision measurements can be utilized in an

inversion method to estimate accurate methane emissions. A similar and very useful application of the method proposed here can be applied for the estimation of the methane emissions from landfills.

Ko et al. (1995) showed that the optimization of sensors network is an NP-hard (i.e. Non-deterministic Polynomial-time hardness) problem, which means that it is difficult for an exhaustive search algorithm to solve all instances of the problem because it requires a considerable time. Various optimization algorithms have been proposed to find the best solution, but these methods are not applicable to all the cases especially for large size problems. To solve such problems, the metaheuristic algorithms are efficient. Some studies discussed the optimization of sensor distribution and number for gas emission monitoring, e.g. Ma et al. (2013)(Ma et al., 2013; Ngae et al., 2019). Ma et al. (2013) used a direct approach with the Gaussian dispersion model to optimize the sensors networks in homogeneous terrains. However, the present study utilizes an inverse approach by solving the adjoint transport-diffusion equation with the building-resolving Computational Fluid Dynamics (CFD) model for an urban environment. This methodological approach for an optimal monitoring network (i.e. coupling of the optimization algorithm, inverse tracers transport modeling and Computational Fluid DynamicsCFD) includes the geometric and flow complexity inherent in an urban region for the optimization process. Recently, for a different application point of view, Ngae et al. (2019) has also described an optimization methodology for determining an optimal sensors network in an urban like environment using the available meteorological conditions only. The CFD computations also required a considerable amount of time to compute the flow and dispersion in an urban environment. However in order to apply the proposed methodology in an emergency situation for an area of interest in a complex urban or industrial environment, an archive database of the CFD calculations can be established for a wide range of meteorological and turbulence conditions and can be utilized in the optimization process.

In this study, the Simulated Annealing (SA) stochastic optimization algorithm (Jiang et al., 2007; Abida et al., 2008; Abida and Bocquet, 2009; Saunier et al., 2009; Kouichi et al., 2016; Kouichi, 2017, etc.) (Jiang et al., 2007; Abida et al., 2008; Abida and Bocquet, 2009; Saunier et al., 2009; Kouichi et al., 2016; Kouichi, 2017; Ngae et al., 2019, etc.) is utilized. The SA algorithm was designed for in the context of the statistical physics. It incorporates a probabilistic approach to explore the search space and converges iteratively to the solution. This algorithm is often used and recommended to solve the problems of sensors network optimization (Abida, 2010). The network optimization process consists of finding the best set of sensors that leads to the minimum of a defined cost function. A cost function can be defined as a regularized norm square of the distance between the measurements and forecasts which is also used for the STE (Sharan et al., 2012). In this study, two canonical problems are considered independently. (i) Optimization of the measuring network: the optimization consists of selecting the best positions of the sensors among a set of potential locations. This choice is operated in a search space constituted by all the possible networks (of a specific size) and based on a cost function that describes quantitatively the quality of the networks. (ii) Identification of the unknown source: the STE is studied in the framework of a parametric approach. Here the challenge is to determine the parameters of the source (intensity and position) using the measurements from the sensors of an optimally designed network.

The reduced optimal networks are validated using an STE technique to estimate the unknown parameters of a continuous point source. The STE problem for atmospheric dispersion events has been an important topic of much consideration

as reviewed in Rao (2007); Hutchinson et al. (2017). Often, the source term is estimated using a network of static sensors deployed in a region. In inverse modeling process, the adjoint source-receptor relationship and concentrations and meteorological datasets are required for the STE. The adjoint source-receptor relationship is defined by an inverse computation of the atmospheric transport dispersion model (Pudykiewicz, 1998). This relationship is often affected by the nonlinearities in the

flow-field by building effects in complex scenarios arising in urban environments, where the backward and forward dispersion concentrations will not match. Various inversion methods can be classified in two major categories: probabilistic and deterministic. The probabilistic category treats source parameters as the random variables associated to the probability distribution. This includes the Bayesian Estimation Theory (Bocquet, 2005; Monache et al., 2008; Yee et al., 2014, etc.), Monte Carlo algorithms using Markov chains (MCMC) (Gamerman and Lopes, 2006; Keats, 2009, etc.) and various stochastic sampling

algorithms (Zhang et al., 2014, 2015, etc.). Deterministic methods use cost functions to assess the difference between observed and modeled concentrations and are based on an iterative process to minimize this difference (Seibert, 2001; Penenko et al., 2002; Sharan et al., 2012, etc.). Among the other approaches, advanced search algorithm algorithms like genetic algorithm (Haupt et al., 2006, etc.) or neural network algorithm (Wang et al., 2015, etc.) and other regularization methods (Ma et al., 2017; Zhang et al., 2017, etc.) have been used for the STE. In this study, we focused on utilized the renormalization inversion method (Issartel,

2005) for the STE using measurements from the optimal networks, which is deterministic in nature and does not require any prior information of the source parameters. The renormalization inversion approach was successfully applied and validated for retrieval of an unknown continuous point source in flat terrain (Sharan et al., 2009, etc.) and also in urban-like environment (Kumar et al., 2015b). Initially, the renormalization inversion method was proposed to estimate emission of the distributed sources (Issartel, 2005). Sharan et al. (2009) and other studies have shown that this technique is also effective for estimating

continuous point sources. For these applications, the hypothesis of a linear relationship between the receptor and the source was assumed. For homogeneous terrains, the adjoint functions can analytically be computed based on the Gaussian solution of the diffusion transport equation to estimate a continuous point release. However, the flow-field in urban or industrial environments is quite complex and the asymmetry of the flow and the dispersed plume in urban regions is generated mainly by the presence of buildings and other structures. In general, the Gaussian models are unable to capture the effects of complex

urban geometries on adjoint sensitivities between source and receptors and also if dense gases are involved, the Gaussian distribution hypothesis fails. Recently, Kumar et al. (2015b, 2016) have extended the applications of the renormalization inversion technique to retrieve an unknown emission source in the urban environments, where a CFD approach was used to generate the adjoint receptors-source relationship. In this process, a coupled CFD-renormalization source reconstruction approach was described for the identification of an unknown continuous point source located at the ground surface or at a horizontal plane

corresponding to a known or predefined altitude above the ground surface, or an elevated release in an urban area.

In this study, two canonical problems are considered separately: (i) optimization of the measuring network: here, the optimization consists of selecting the best positions to be instrumented by the sensors among a set of potential locations. This choice is operated in a space of search constituted of all possible networks (of a specific size) and based on a cost function that describes quantitatively the quality of the networks. The cost function is defined from the inverse renormalization method and is quantified during the searching process. (ii) Identification of the unknown source: the STE is studied in the framework of a parametric approach using the renormalization technique. Here the challenge is to determine the

parameters of the source (intensity and position) using any measurements vector (in practice the number of measurements is limited). The evoqued canonical problems are coupled in order to evaluate the performance of the proposed methodology.

The main objective of this study is to determine the sensors locations choice in an urban domain for an optimal monitoring network dedicated to estimate the location and intensity of a continuously polluting point source. A methodology is proposed to determine an optimal network formed by a predetermined number of sensors, to better characterize a source of pollutant in a complex urban environment. This study deals with a case of reducing the number of sensors in order to obtain an optimal network from an existing optimally reducing the size of an existing monitoring network. For this purpose, a predefined network of sensors deployed in an area of interest is considered to determine an optimized network with smaller number of sensors, but, with comparable information. This work explores with two requirements of the optimal networks that modifies the spatial configuration of an existing network by moving the sensors and also reduces the number of sensors of an existing large network. In real situation, this methodology can be applied for the optimization of mobile networks deployed in emergency situation. The methodological approach to optimize the optimally reduce the size of an existing monitoring network in urban environment was is presented by coupling the SA stochastic algorithm with the renormalization inversion technique and the CFD modeling approach. The concentration measurements from these optimized networks of sensors obtained from the optimally reduced sensors networks in 20 trials of the Mock Urban Setting Test (MUST) field tracer experiment were are utilize to validate the methodology to retrieve by estimating an unknown continuous point source in an urban-like environment.

## 2 Source Term Estimation Method: The Renormalization

In the context of an inversion approach, source parameters are often determined using the concentration measurements at the sensor locations and a source-receptors relationship. The release is considered continuous from a point source located at the ground or at a horizontal plane corresponds to an altitude of a known source height. Since the optimization methodology presented in the next section utilizes some concepts from the renormalization inversion methodology (Sharan et al., 2009), the renormalization theory to estimate a continuous point release is briefly presented in following subsections.

### 2.1 Source-Receptor relationship

A source-receptor relationship is an important concept in the source reconstruction process and it can be linear or nonlinear. This study deals with the linear relationship, as except from the nonlinear chemical reactions, most of the other processes occurring during the atmospheric transport of trace substances are linear: advection, diffusion, convective mixing, dry and wet deposition, and radioactive decay (Seibert and Frank, 2004). A source-receptor relationship between the measurements and the source function is defined based on a solution of the adjoint transport-diffusion equation that exploits the computed adjoint functions (retroplumes) corresponding to each receptor (Pudykiewicz, 1998; Issartel et al., 2007, etc.). These retroplumes provide a sensitivity information between the source position and the sensor locations. Let's consider a discretized domain of $N$ grid cells in a 2-dimensional space $\mathbf{x} = (x, y)$, a vector of $M$ concentration measurements $\boldsymbol{\mu} = (\mu_1, \mu_2, ..., \mu_M)^T \in \mathbb{R}^M$, and an unknown source vector $\mathbf{s}(\mathbf{x}) \in \mathbb{R}^N$ to estimate. The measurements $\boldsymbol{\mu}$ are related to the source vector $\mathbf{s}$ by the use of sensitivity coefficients (also referred as adjoint functions) (Hourdin and Talagrand, 2006). The sensitivity coefficients describe

the backward propagation of information from the receptors toward the unknown source. These vectors are related by the following linear relationship:

$$\boldsymbol{\mu} = \mathbf{A}\mathbf{s} + \boldsymbol{\epsilon} \tag{1}$$

where $\boldsymbol{\epsilon} \in \mathbb{R}^M$ is the total measurements error and $\mathbf{A} \in \mathbb{R}^{M \times N}$ is the sensitivity matrix with $\mathbf{A}(\mathbf{x}) = [\mathbf{a}(\mathbf{x}_1), \mathbf{a}(\mathbf{x}_2), ..., \mathbf{a}(\mathbf{x}_N)]$. Here, each column vector $\mathbf{a}(\mathbf{x}_i) \in \mathbb{R}^M$ of the matrix $\mathbf{A}$ represents the potential sensitivity of a grid cell with respect to all $M$ concentration measurements.

For a given set of the concentration measurements $\boldsymbol{\mu}$, the source estimate function $\mathbf{s}(\mathbf{x})$ in Eq. (1) can easily be estimated by formulating a constrained optimization problem. This optimization problem minimizes a cost function $J(\mathbf{s}) = \mathbf{s}^T\mathbf{s}$, subjected to a constraint $\boldsymbol{\epsilon} = \boldsymbol{\mu} - \mathbf{A}\mathbf{s} = 0$. Using the method of Lagrange multipliers, $\mathbf{s}(\mathbf{x})$ can be estimated as a least-norm solution:

$$\mathbf{s} = \mathbf{A}^T\mathbf{H}^{-1}\boldsymbol{\mu} \tag{2}$$

where $\mathbf{H}^{-1}$ is inverse of the *Gram matrix* $\mathbf{H} = \mathbf{A}\mathbf{A}^T$. This estimate (Eq. (2)) is not satisfactory because it generates artifacts at the grid cells corresponding to the measurement points. Adjoint functions become singular at these points and have very large values. These large values do not represent a physical reality, but rather an artificial information. This was highlighted by Issartel et al. (2007) which reduced this artificial information by a process of renormalization.

## 2.2 Renormalization process

This process involves a weight function in space $\mathbf{W}(\mathbf{x}) \in \mathbb{R}^{N \times N}$, which is purely a diagonal matrix with the diagonal elements $w_{jj} > 0$ such that $\sum_{i=1}^{N} w_{jj} = M$. Introduction of $\mathbf{W}$ transforms the source-receptor relationship in Eq. (1) to:

$$\boldsymbol{\mu} = \mathbf{A}_w\mathbf{W}\mathbf{s} + \boldsymbol{\epsilon}. \tag{3}$$

where the modified sensitivity matrix $\mathbf{A}_w$ is defined as $\mathbf{A}_w = \mathbf{A}\mathbf{W}^{-1} = [\mathbf{a}_w(\mathbf{x}_1), \mathbf{a}_w(\mathbf{x}_2), ..., \mathbf{a}_w(\mathbf{x}_N)]$ in which the column vector $\mathbf{a}_w(\mathbf{x}_i) = \mathbf{a}(\mathbf{x}_i)/w(\mathbf{x}_i)$ of $\mathbf{A}_w$ is the weighted sensitivity vector at $\mathbf{x}_i$. Considering a similar approach that outlined in previous subsection, a new constrained optimization problem can be formulated for Eq. (3) to estimate $\mathbf{s}(\mathbf{x})$. This optimization problem minimizes a cost function $J(\mathbf{s}) = \mathbf{s}^T\mathbf{W}\mathbf{s}$, subjected to a constraint $\boldsymbol{\epsilon} = \boldsymbol{\mu} - \mathbf{A}_w\mathbf{W}\mathbf{s} = 0$, and deduces the following expression $\mathbf{s}_w$ of $\mathbf{s}$ (Appendix A in Kumar et al., 2016):

$$\mathbf{s}_w = \mathbf{A}_w^T\mathbf{H}_w^{-1}\boldsymbol{\mu} \tag{4}$$

where $\mathbf{H}_w^{-1}$ is the inverse of $\mathbf{H}_w = \mathbf{A}_w\mathbf{W}\mathbf{A}_w^T$.

The weight function in the above discussed renormalization process is computed by using an iterative algorithm demonstrated by Issartel et al. (2007) (Appendix A). A brief derivation for the estimation of an unknown point source (i.e. location and intensity) from the renormalization inversion is described in Appendix B.

## 2.3 Identification of point source

Consider a point source of continuous release at a position $\mathbf{x_o} = (x_o, y_o)$ and with the intensity $q_o$. The point source is thus expressed as a function of the preceding parameters: $\mathbf{s}(\mathbf{x}) = q_o \delta(\mathbf{x} - \mathbf{x_o})$. The relationship between the source and the measurements (Eq. ) becomes: $\boldsymbol{\mu} = q_o \mathbf{a}_w(\mathbf{x_o}) w(\mathbf{x_o}) + \boldsymbol{\epsilon}$. By replacing the measurement term in Eq. (4), one obtains:

$$\mathbf{s}_w = q_o w(\mathbf{x_o}) \mathbf{A}_w^T \mathbf{H}_w^{-1} \mathbf{a}_w(\mathbf{x_o}).$$

$\mathbf{s}_w$ reaches its maximum at position $\mathbf{x_o}$ as the renormalization criterion (Eq. ) is satisfied only at this position $\mathbf{x_o}$. Thus, $\mathbf{s}_w(\mathbf{x})$ at $\mathbf{x_o}$ becomes:

$$\mathbf{s}_w(\mathbf{x_o}) = q_o w(\mathbf{x_o}),$$

which estimates the source intensity $q_o = \mathbf{s}_w(\mathbf{x_o})/w(\mathbf{x_o})$.

## 3 The Combinatorial Optimization of a Monitoring Network

A predefined large network of $n$ sensors deployed in an area of interest is considered to determine an optimized network with smaller number of sensors, but with comparable information. For a given number of $p$ sensors such that $p < n$, one determines an array of $p$ sensors among $n$, which delivers maximum of the information. It is a combinatorial optimization problem that consists of choosing $p$ sensors among $n$, and thus constituting an optimal network. The optimal network will consist of $p$ sensors for which a defined cost function is minimum. The number of possible choices ${}^nC_p$ (number of combinations of $p$ among $n$) is very high, when an initial network is sufficiently instrumented ($n$ large) and $p$ is small with respect to $n$. As the number of combinations to be tested is very large, minimum of a cost function will be evaluated by a stochastic algorithm, viz. simulated annealing (SA).

### 3.1 Cost function

A cost function is defined (based on the renormalization theory) as a function that minimizes the quadratic distance between the observed and the simulated measurements according to the $\mathbf{H}_w$ norm (Issartel et al., 2012)., where $\mathbf{H}_w$ is the *Gram matrix* defined in a previous section 2.2. The quadratic distance between the real and the simulated concentration measurements according to the $\mathbf{H}_w$ norm is given by :

$$J = \|\boldsymbol{\mu} - \hat{\boldsymbol{\mu}}\|_{\mathbf{H}_w^{-1}}^2 = \frac{1}{2} \left[ (\boldsymbol{\mu} - \hat{\boldsymbol{\mu}})^T \mathbf{H}_w^{-1} (\boldsymbol{\mu} - \hat{\boldsymbol{\mu}}) \right]$$

When considering a point source, $\hat{\boldsymbol{\mu}}$ is written by $\hat{\boldsymbol{\mu}} = q_o \mathbf{a}_w(\mathbf{x}) w(\mathbf{x})$, where $q_o$ and $\mathbf{x}$ are respectively the intensity and the position of a point source. By replacing $\hat{\boldsymbol{\mu}}$ in Eq. , one obtains (Sharan et al., 2012; Issartel et al., 2012):

$$J = J(q_o, \mathbf{x}) = \frac{1}{2} \left[ (\boldsymbol{\mu} - q_o \mathbf{a}_w(\mathbf{x}) w(\mathbf{x}))^T \mathbf{H}_w^{-1} (\boldsymbol{\mu} - q_o \mathbf{a}_w(\mathbf{x}) w(\mathbf{x})) \right]$$

For a fixed $\mathbf{x}$ in Eq. , $J$ reaches a strict local minimum if following two conditions are satisfied:

$$\frac{\partial J(q_o, \mathbf{x})}{\partial q_o} = 0$$

$$\frac{\partial^2 J(q_o, \mathbf{x})}{\partial q_o^2} > 0$$

For each fixed $\mathbf{x}$, the first condition (Eq. ) gives an estimate ($\tilde{q_0}$) of $q_0$ as: $\tilde{q_0} = \frac{\mathbf{a}_w^T(\mathbf{x}) \mathbf{H}_w^{-1} \boldsymbol{\mu}}{w(\mathbf{x})}$. The second condition (Eq. ) is always satisfied as $\frac{\partial^2 J(q_o, \mathbf{x})}{\partial q_o^2} = w^2(\mathbf{x}) > 0, \forall \mathbf{x}$

(Sharan et al., 2012). Corresponding to the estimate $\tilde{q_0}$ from the first condition (Eq. ) , the cost function $J$ from Eq. leads to the following expression (Issartel et al., 2012): A cost

function (say $J_s(\mathbf{x})$) to minimize is defined (Appendix C) as follows:

$$J_s(,\mathbf{x}) = \frac{\boldsymbol{\mu}^T \mathbf{H}_w^{-1} \boldsymbol{\mu}}{2} 1 - \frac{\mathbf{s}_w^2}{\boldsymbol{\mu}^T \mathbf{H}_w^{-1} \boldsymbol{\mu}} \tag{5}$$

where $\mathbf{s}_w$ is same as given in given by Eq. (4) and $\boldsymbol{\mu}^T \mathbf{H}_w^{-1} \boldsymbol{\mu}$ is a positive constant. Considering Eq. , it is obvious that the minimization of $J$ also corresponds to the

maximization of the term $\frac{\mathbf{s}_w^2}{\boldsymbol{\mu}^T \mathbf{H}_w^{-1} \boldsymbol{\mu}}$ or minimization of term $\left[ 1 - \frac{\mathbf{s}_w^2}{\boldsymbol{\mu}^T \mathbf{H}_w^{-1} \boldsymbol{\mu}} \right]$. Accordingly, the minimum value of the cost function $J$ in Eq leads to the following expression of

the cost function (say $J_s(\mathbf{x})$) to minimize:

$$J_s(\mathbf{x}) = 1 - \frac{\mathbf{s}_w^2}{\boldsymbol{\mu}^T \mathbf{H}_w^{-1} \boldsymbol{\mu}}$$

. A global minimum of the cost function $J_s(\mathbf{x})$ is evaluated by the SA algorithm.

## 3.2   Simulated Annealing (SA) algorithm for the Sensor's Network Optimization

The problem of optimization of a network is solved using the simulated annealing (SA) algorithm. The SA optimization algorithm is utilized here for the determination of the optimal networks by comparing its performance with the Genetic Algorithm

(GA)(Kouichi, 2017). These algorithms of different search technics (SA probabilistic and GA evolutionary) are evaluated based on the same cost function. The results showed that the optimal networks retained by the GA and the SA are quantitatively and qualitatively comparable (Kouichi, 2017). The SA has advantageous because it is relatively easy to implement and takes smaller computational time in comparison to GA. Both SA and GA optimization algorithms in the framework of this approach (based in the renormalization theory) has little influence on the estimation of the parameters of a source (Kouichi, 2017).

The SA is a random optimization technique based on an analogy with thermodynamics. The technique has been introduced to the computational physics over sixty years ago in the classic paper by Metropolis et al. (1953). The algorithm of simulated annealing is initiated by starting from an admissible network. At the subsequent steps, the system moves to another feasible network, according to a prescribed probability, or it remains in the current state. However, it is crucial to explain how this probability is calculated. The mobility of the random walk depends on a global parameter $T$ which is interpreted as 'temperature'.

The initial values of $T$ are large, allowing free exploration of large extents of the state space (this corresponds to the "melted state" in terms of the kinetic theory of matter). In the subsequent steps, the temperature is lowered allowing the algorithm to reach a local minimum.

For the SA, each network is considered as a state of a virtual physical system, and the objective function is interpreted as the internal energy of this system in a given state. According to statistical thermodynamics, the probability of a physical system

for being in a same state follows the Boltzmann distribution and depends on its internal energy and the temperature level. By analogy, the physical quantities (temperature, energy, etc.) become a pseudo-quantities. And during the minimization process,

the probabilistic treatment consists to accept a new network selected in the neighborhood of the current network following the same Boltzmann distribution and depending both on the cost difference between the new and the current networks and on the pseudo-temperature ('temperature'). To find the solution, the SA incorporates the 'temperature' into a minimization procedure. So at high 'temperature' (starting 'temperature'), the space of solution is widely explored, while at lower 'temperature' the exploration is restricted. The algorithm is stopped when the cold 'temperature' is reached. It is necessary to choose the law of decreasing 'temperature', called as cooling schedule. Different approaches to parameterize the SA are explored in Siarry (2016). Kirkpatrick et al. (1983) proposed an average probability to determine the initial (starting) 'temperature'. Nourani and Andresen (1998) compared the most used cooling schedules (exponential, logarithmic, and linear). The SA algorithm starts minimization of an objective function at annealing 'temperature' from a single stochastic point, then it searches for the minimal solutions by attempting all the points in search domain with respect to their value of the 'temperature'. The algorithm is depicted in a flow diagram in Figure 2 and a step by step implementation of the SA procedure for an optimized monitoring network in an urban environment is described as follows:

**Step 1. Parameters setting and initialization**

*Network parameters* ($n$ and $p$): $n$ is the number of possible locations of the sensors and $p$ is the optimal network number of sensors.

*Starting 'temperature'* ($T_0$): $T_0$ is also called the highest 'temperature'. It was determined from the Metropolis law: $T_0 = -\frac{\overline{(\Delta J_s)}}{log(P_0)}$, where $\overline{(\Delta J_s)}$ is an average of the difference of cost functions calculated for a large number of cases. $P_0$ is an acceptance probability and following the recommendations of Kirkpatrick et al. (1983), it was set to 0.8. Start iterations ($I_{tt} = 0$).

*Length of the bearing* ($L_{max}$): A length of the bearing is the number of iterations to be performed at each 'temperature' level. An equilibrium is reached for this number of iterations and any significant improvement of the cost function can be expected. No general rule is proposed to determine a suitable length. This number is often constant and proportional to the size of the problem.

*The 'temperature' decay factor* ($\theta$): The 'temperature' remains constant for $L_{max}$ iterations corresponding to each bearing. We used the exponential schedule due to its efficiency as denoted by Nourani and Andresen (1998). Then, the 'temperature' decreases law between two bearings varies as: $T_{b+1} = \theta T_b$, with $0 < \theta < 1$, where $b$ represents a bearing. So, it was retained a decay pattern by the bearings.

*The cold 'temperature'* ($T_{cold}$): $T_{cold}$ is often called the stopping 'temperature'. There is no clear rule to set this parameter. It is possible to stop calculations when no improvement in the cost function is observed during a large number of combinations. One can estimate this number and take into account the maximum length $L_{max}$ of each bearing, thus the cold 'temperature' can be expressed as a fraction of the starting 'temperature' $T_0$.

*Assigning the first best set of sensors,* $\mathbf{x}_{Best} \leftarrow \mathbf{x}_{rand}(p, n)$: $\mathbf{x}_{rand}(p, n)$ corresponds to a vector of $p$ sensors locations randomly

chosen among the *n* possible locations. A new solution is randomly explored. This vector is assigned to the first 'best' set of sensors.

**Step 2. Assigning a new set of sensors**

$\mathbf{x}_{new} \leftarrow \mathbf{x}_{rand}(p, n)$, where $\mathbf{x}_{rand}(p, n)$ corresponds to a vector of $p$ sensors locations randomly chosen among the *n* possible locations. This vector is assigned to a new set $(\mathbf{x}_{new})$ of the sensors.

**Step 3. Cost difference**

Given a sensor location $\mathbf{x}_{new}$, the cost function $J_s(\mathbf{x}_{new})$ is computed as follows:

- set $\boldsymbol{\mu}$ vector by using the measurements at the $\mathbf{x}_{new}$ locations,
- set rows of matrix $\mathbf{A}$ using the sensitivity at the $\mathbf{x}_{new}$ locations,
- determine $w(\mathbf{x})$, $\mathbf{H}_w$, and $\mathbf{a}_w$ iteratively using the algorithm in Eq. (A2),
- compute the source term $\mathbf{s}_w(\mathbf{x})$ using Eq. (4),
- compute the cost function $J_s(\mathbf{x}_{new})$ using Eq. (5).

$J_s(\mathbf{x}_{best})$ is computed like $J_s(\mathbf{x}_{new})$ using the same precedent steps. A cost difference is then calculated using $\Delta J_s = J_s(\mathbf{x}_{new}) - J_s(\mathbf{x}_{best})$. Increment the iterations $(I_{tt} \leftarrow I_{tt} + 1)$.

**Step 4. Test of sign of $\Delta J_s$**

If $\Delta J_s < 0$, the error associated with $\mathbf{x}_{new}$ is less than that with $\mathbf{x}_{best}$ and thus $\mathbf{x}_{new}$ will become the next 'best network' (*Step* 6). If this condition is not satisfied, the algorithm can jump out of a local minimum (*Step* 5).

**Step 5. Conditional jump**

When $\Delta J_s > 0$, the algorithm has ability to jump out any local minima if condition: $P_{01} \leq \exp(-\frac{\Delta J_s}{T})$ is satisfied, where $P_{01}$ is the acceptance probability (a random number between 0 and 1), and $T$ is the current annealing 'temperature'. It means that $\mathbf{x}_{new}$ will be the next 'best network' even if the associated error is greater than that of $\mathbf{x}_{best}$. If $P_{01} > \exp(-\frac{\Delta J_s}{T})$, go to *Step* 7.

**Step 6. Update $\mathbf{x}_{best}$**

In this step, $\mathbf{x}_{best}$ is updated by $\mathbf{x}_{new}$.

**Step 7. Maximum iteration check**

If the maximum number of iterations of a bearing $(L_{max})$ is reached, a state of equilibrium is then achieved for this 'temperature' and one can cool the actual 'temperature' (*Step* 8). If not, continue iterations (*Step* 2).

**Step 8. 'Temperature' cooling**

'Temperature' is cooled using the cooling schedule and iteration variable is reset to zero.

**Step 9. Cold 'temperature' test**

The cold 'temperature' ($T_{cold}$) is also known as the stopping 'temperature'. If this 'temperature' is reached, the algorithm is
stopped. When $T_{cold}$ is not reached, other 'temperature' bearing are performed using the cooling schedule.

**Step 10. Optimal network**

At this step, the last best network $\mathbf{x}_{best}$ is the optimal network. Source parameters are then estimated using the concentration
measurements and retroplumes only for sensors from the obtained optimal network as: (i) $\mathbf{x}_0$ is estimated at position of the
maximum of the source estimate function $\mathbf{s}_w(\mathbf{x})$, and (ii) the intensity $q_0$ is given by $q_0 = \mathbf{s}_w(\mathbf{x}_0)/w(\mathbf{x}_0)$.

In stochastic optimization algorithms, especially in the SA, it was observed that there is no guarantee for the convergence
of the algorithm with such a strong cooling (Cohn and Fielding, 1999; Abida et al., 2008). However, chances are that a near-
optimal network configuration can be reached. Due to this, one or more near-optimal networks can be obtained from this
methodology that satisfy the conditions of near overall optimum condition.

## 4    The Mock Urban Setting Test (MUST) Tracer Field Network

The MUST field experiment was conducted by the Defense Threat Reduction Agency (DTRA) in 2001. It was aimed to help
developing and validating the numerical models for flow and dispersion in an idealized urban environment. The experimental
design and observations are described in detail in Biltoft (2001) and Yee and Biltoft (2004). In this experiment, an urban
canopy was represented by a grid of 120 containers. These containers were arranged along 12 rows and 10 columns on the
army ground in the Utah desert, USA. Each container has dimensions of 2.54 m high, 12.2 m long and 2.42 m wide. The
spacing between the horizontal lines is 12.9 m, while the columns are separated by a distance of 7.9 m. The total area thus
formed is approximately $200 \times 200\,\mathrm{m}^2$. The experiment consists of 63 releases of a flammable gas (propylene $C_3H_6$) that is not
dangerous or harmful in quantities and could be released through the dissemination system into the open atmosphere (Biltoft,
2001). Different wind conditions (direction, speed, atmospheric stability) as well as different positions for gas emissions (inside
or outside the MUST urban canopy at different heights) were considered. These gas emissions were carried out under stable,
very stable, and neutral stability conditions. In this study, 20 trials in various atmospheric stability conditions are selected and
the meteorological variables are taken from an analysis of meteorological and micro-meteorological observations in Yee and
Biltoft (2004) (Table 1). It is noted that the errors related to meteorological data can affects the accuracy of the source term
estimation (Zhang et al., 2014, 2015), although this error is not considered in this study. In each trial, the gas was continuously
released for $\approx 15$ min, during which the concentration measurements were made. These concentration measurements were

carried out by 48 photoionization detectors (PIDs). 40 sensors were positioned on four horizontal lines at 1.6 m height (Figure 1) and 8 sensors were deployed in vertical direction at a tower located approximately in center of the MUST array.

## 5 CFD Modelling for Retroplumes in an Urban Environment

The flow-field in atmospheric dispersion models in geometrically complex urban or industrial environments cannot be considered as homogeneous throughout the computational domain. This is because the buildings and other structures in that region influence and divert the flow into unexpected directions. Consequently, the dispersion of a pollutant and computations of the adjoint functions (retroplumes) are affected by the flow-field induced by these structures in an urban region. Recently, Kumar et al. (2015a) utilized a CFD model to compute the flow-field and the forward dispersion in 20 trials of the MUST field experiment. In order to reconstruct an unknown continuous point source, the computed flow-field is then used to compute the retroplumes for all selected trials (Kumar et al., 2015b). A The CFD computations of the flow field presented in Kumar et al. (2015a) and retroplumes computed in Kumar et al. (2015b) are utilized in the proposed optimization methodology described in this study to obtain the optimal monitoring networks. In these studies, a CFD model fluidyn-PANACHE was utilized to calculate the flow-field, considering a subdomain of calculation (whose dimensions are $250 \times 225$ m$^2$ with a height of 100 m) that consists the MUST urban array created by the containers, sources, receptors, and other instruments in this experiment. This subdomain is embedded in a larger computational domain (dimensions of $800 \times 800$ m$^2$ with a height of 200 m) to ensure a smooth transition of the flow between the edges of the domain and the obstacles zone. This extension of the outer domain far from the main experimental site is essential to reduce effects of the inflow boundary conditions imposed at inlet of the outer domain. A more detailed description about the CFD model and its simulations for the MUST field experiment, e.g., boundary conditions, turbulence model, etc. are presented in Kumar et al. (2015a) and now briefly discussed in the Supplementary Information (SI). An unstructured mesh was generated in both domains with more refinement in the urbanized area in inner subdomain and at the positions of receptors, thus generating 2849276 meshes.

The simulations results with fluidyn-PANACHE in each MUST trial were obtained with inflow boundary conditions from vertical profiles of the wind ($U$), the turbulent kinetic energy ($k$) and its dissipation rate ($\epsilon$). These inflow profiles include: *(i) Wind profile*: Gryning et al. (2007) profiles in stable and neutral conditions and a profile based on the stability function by Beljaars and Holtslag (1991) in very stable conditions, *(ii) Temperature profile:* Monin-Obukhov similarity theory based logarithmic profiles, *(iii) Turbulence profiles:* $k$ and $\epsilon$ profiles are based on an approximate analytical solution of one-dimensional $k - \epsilon$ prognostic equations (Yang et al., 2009). The atmospheric stability effects in the CFD model fluidyn-PANACHE are included through the inflow boundary condition (via advection). fluidyn-PANACHE includes a Planetary Boundary Layer (PBL) model that serves as an interface between the meteorological observations and the boundary conditions required by the CFD solver. The observed turbulence parameters, e.g. (i) sensible heat flux ($Q_h$), the Obukhov length ($L$), (iii) surface friction velocity ($u_*$) and the temperature scale ($\theta_*$) were used to derive the vertical profiles of mean velocity and potential temperature. As an example, the wind velocity vectors around some containers for the trial 11 are shown in SI Figure S1.1. This figure shows the deviations in the wind speed and its direction due to the obstacles in an urban-like environment. It should be noted

that the MUST experiment took place under neutral to stable and strongly stable conditions. However, the only atmospheric stability effects included in the CFD model are through the specification of inflow boundary conditions. Atmospheric stability has a profound impact on dispersion and would thus influences the adjoint functions. However, as presented and discussed in our previous study (Kumar et al., 2015a), even with specification of the stability dependent inflow boundary conditions only,

the predicting forward concentrations from the CFD model are in good agreement with the measured concentrations for all 20 trials in different atmospheric stability conditions. However, at micro-scales also, small irregularities can break the repeated flow patterns found in a regular array of containers with identical shape (Qu et al., 2011). In addition, uncertainties associated with the thickness and the properties of the material of the container wall also affect flow pattern and the resulted concentrations and adjoint functions (Qu et al., 2011). Accordingly, the atmospheric stratification and stability effects should also be

included through surface cooling or heating in the CFD model and stability effects from inflow boundary conditions. Since the released gas propylene is heavier than the air and would behave as a dense gas, a buoyancy model was used to model the body force term in the Navier-Stokes equations. The buoyancy model is suitable for the dispersion of heavy gases where density difference in the vertical direction drives the body force.

In order to compute the retroplumes in each MUST trial, firstly the CFD simulations were performed to compute the con-

verged flow-field in computational domain, secondly the flow-field is reversed and used in the standard advection-diffusion equation to compute the adjoint functions $\mathbf{a}_i(\mathbf{x})$. In this computation of the retroplumes corresponding to each receptor in a selected trial, the advection-diffusion equation is solved by considering a receptor as a virtual point source with unit release rate at the height of that receptor. Also, the meteorological conditions remained invariant during the whole experimental period in a trial. The details about the retroplumes and the correlated theory of the duality verification (i.e. comparison of the concentra-

tions predicted with the forward (direct) model and the adjoint model) for all 20 trials of the MUST field experiment are given in Kumar et al. (2015b) and we have utilized the same retroplumes in this study for the optimization process. Since we are concerned to establish an optimized monitoring network in a domain that contains the MUST urban array, the retroplumes are computed in the inner subdomain only. Consequently all the computations for an optimized monitoring network were carried out in the inner subdomain only. The sensors in the optimized monitoring network are supposed to deploy on a fixed vertical

height above the ground surface. Accordingly, the retroplumes corresponding to only 40 receptors at 1.6 m height were utilized in computations for the optimized monitoring networks in the MUST urban environment.

## 6   Results and Discussion

The calculations were performed by coupling the SA algorithm to a deterministic renormalization inversion algorithm and the CFD adjoint fields to optimize the optimally reduce the size of an existing monitoring network in an urban-like environment

of the MUST field experiment. The network optimization process consists of finding the best set of sensors that leads to the lowest cost function. In this study, the validation is realized following two separated steps. The first step consists to form two optimal monitoring networks by using the presented optimization methodology which makes it possible to reduce the size of an original network of 40 sensors to approx. one-third (13 sensors) and one-fourth (10 sensors). The second step consists to

compare a posteriori ~~the~~ performance of the obtained ~~optimal networks to~~ reduced size optimal networks with the 'MUST ~~predefined~~ existing network' of 40 sensors at 1.6 m above the ground surface. In first step, ~~the comparison~~ a comparison (based on a cost function) with networks of the same size (e.g. 10 sensors) was performed implicitly during the optimization process. As the SA is an iterative algorithm, during the optimization process networks of same size are compared at each iteration and the ~~'best one'~~ 'best one' is retained. The networks have also been generated randomly like in Efthimiou et al. (2017); however, the search space of the problem is very large. In our case, the number of the compared networks is equivalent to the number of iterations (as an example for optimal network of 10 sensors $\sim 3 \times 10^4$ configurations are compared). Here, the comparison is based on a cost function and inspired from the renormalized data assimilation method. The cost function quantifies the quadratic distance between the observed and the simulated measurements. The 'optimal network' produces the 'best' description of the observations (i.e. corresponds to the minimal quadratic distance) and permits a posteriori to estimate the location and emission rate of an unknown continuous point source in an urban-like environment.

The size of the 'MUST predefined (original) network' is 40 sensors and the sizes of the optimized networks are fixed after performing a first optimization with the number of sensors from 4 to 16 (Kouichi, 2017). This first evaluation showed that for some trials, a small number of sensors could not allow to correctly reconstruct the source and divergences in the calculations have been noted. Accordingly, the source estimation obtained for different trials and network sizes show that, very often, networks of less than 8 sensors ~~can not correctly~~ may not characterize the source correctly. On the other hand, beyond 13 sensors, the source estimation is not significantly improved, and the associated errors were roughly constant (Kouichi, 2017). Therefore, in order to ensure an acceptable estimate of the source for all the trials, the sizes of the optimized network are fixed as 10 and 13 sensors ($1/4^{th}$ and $\sim 1/3^{rd}$ respectively of the original network of 40 sensors).

The optimization calculations were performed using Matlab on a computer with configuration "Intel® Core™ i7-4790 CPU @ 3.60 GHz and 16 GB RAM". The averaged computational time for optimization of one 10 sensors network was $\approx 2.5$ hrs and $\approx 8.5$ hrs for 13 sensors network. In computations, a value of parameter $T_0 = 10$ was fixed according to the scale of cost function and using the methodology described in *Step* 1 and $T_{cold} = 10^{-13}$ was used for both the optimal sensors networks. $\theta$ is a decay factor of the 'temperature' for an exponential cooling schedule that describes a procedure of the temperature decrease. The best cooling schedule is the exponential decay as demonstrated by Nourani and Andresen (1998); Cohn and Fielding (1999). $\theta$ was fixed as 0.9 following the recommendation in literature (Siarry, 2014). This value allows a sufficiently slow cooling in order to give more chance to the algorithm to explore a large search space and to avoid the local minima. $L_{max}$ is taken as 100 & 200 for 10 & 13 sensors networks, respectively, following the recommendation in Siarry (2014) and according to number of the possible combinations that increases with the number of sensors ($8.5 \times 10^8$ for 10 sensors and $1.2 \times 10^{10}$ for 13 sensors).

Figure 3 shows the optimal networks of 10 and 13 sensors respectively for three representative trials 5 (very stable), 11 (neutral), and 19 (stable) in the MUST urban array. These three trials correspond to one trial each in neutral, stable, and very stable atmospheric conditions during the release. The optimal monitoring networks of 10 and 13 sensors for all selected 20 MUST trials are shown in SI Figures S2.1&S2.2.

In order to analyze the performance of the optimal monitoring configurations of smaller sizes, the source reconstructions were performed to estimate the unknown location and the intensity of a continuous point release. These source reconstruction results were obtained using the information from the optimal monitoring networks formed by 10 and 13 sensors in each MUST trial. In this performance evaluation process, the retroplumes and the concentration measurements were utilized from the sensors corresponding to these optimal networks. The retroplumes were computed using CFD simulations, considering the dispersion in a complex terrain. The source reconstruction results from both the optimal monitoring networks were also compared with results computed from the initial MUST network formed by 40 sensors (Kumar et al., 2015b). As in practice, the number of measurements is limited, this comparison allowed concluding that in urban areas, the reduction of optimally reduction of a networks size is possible and does not degrade without degrading significantly its efficiency in for the source estimation.

Source estimation results from the different monitoring networks are shown in Table 2 for all 20 selected trials of the MUST experiment. These results are presented in terms of the location error $(E_l^p)$, which is an euclidean distance between the estimated and the true source location, and $E_q^p$, a ratio of the estimated to the true source intensity. The corresponding monitoring network is represented by a superscript $p$ (representing the number of sensors in an optimal network) on $E_l^p$ and $E_q^p$. In order to quantify the uncertainty, a 10% Gaussian noise was added at each measurements. Accordingly, 50 simulations for the source reconstruction were performed with these noise measurements using the optimal networks for each trial. The average and the standard deviation of $E_l^p$ and $E_q^p$ are calculated and the result results are also presented in Table 2.

For a given trial, a parameter *skeleton* represents the common sensors between two optimal networks of different sizes (with 10 and 13 sensors). These results exhibit that the SA algorithm coupled with renormalization inversion theory and CFD modeling approach has succeeded in proposing the good optimal monitoring networks to reduce the size of an existing larger network to estimate the unknown emissions with almost similar accuracy in an urban environment.

Figure 4 shows isopleths of the renormalized weight function (also called as the visibility function) and the normalized source estimate function $\mathbf{s}_w^n(\mathbf{x}) = \mathbf{s}_w(\mathbf{x})/\max(\mathbf{s}_w(\mathbf{x}))$ correspond to both optimal monitoring networks for three representative trials (e.g. 5, 11, and 19) of the MUST experiment. These isopleths for all selected 20 MUST trials are shown in SI Figures S3. As already discussed in the literature, the visibility function includes the natural information associated with a monitoring network for the source retrieval in a domain and physically interprets the extent of regions seen by the network (Issartel, 2005; Sharan et al., 2009). This function is independent of the effective values of the concentration measurements and depends only on geometry of the monitoring network. Hence, this leads to a priori information about the unknown source apparent to the monitoring network. A statistical parameter, factor of $g$ (FA$g$), for the source reconstruction results from each monitoring network is presented in Table 3, where FA$g$ represents the percentage no. of trials in which the source intensity is estimated within a factor of $g$ to the actual emission rates. The statistics calculated with original 40 sensors network show that the average location error for all 20 trials is 14.62 m, and in 75% of the trials, the intensity of the source is estimated within a factor of two to the actual emission rates. In 90% of the trials, intensity was estimated within a factor of three and within a factor of four in 95% trials (Table 3). If trial 2 is considered, large location errors (greater than 30 m) and the intensity values ranged between a factor of three to five, were observed (Table 2) independently of the number of sensors in the networks . If

we consider the trials 15, 16, & 20, it was noted from the numerical results that the larger location errors do not necessarily correspond to the high intensity errors (Table 2).

From distribution of the optimized sensors in networks in Figures 3 for trials 5, 11 & 19 and SI Figures S2.1&S2.2 for all selected trials, it was noted that a larger number of sensors are close to the source position in the optimal networks in most

of the trials. This tendency makes it possible for sensors from an optimal network to monitor the region where a source can be located and it can be explained in terms of the visibility function. The visibility function includes the natural information associated with a monitoring network for the source retrieval in a domain and physically interprets the extent of regions seen by the network (Issartel, 2005; Sharan et al., 2009). The visibility function is independent of the effective values of the concentration measurements and depends only on geometry of the monitoring network. Hence, this leads to a priori information about the unknown source apparent to the monitoring network. It was observed that the visibility functions have significant levels at the source positions (Figures 4 & S3).

The source reconstruction results from the optimal monitoring networks formed by 10 sensors have an averaged location error ($E_l^{10}$) of 19.20 m for all 20 trials in the MUST experiment (Tables 2&3). In most of the trials, the location and the intensity of a continuous point emission are estimated accurately and close to the true source parameters. The location error is minimum in trial 14 ($E_l^{10}$ = 5.50 m) and maximum in trial 2 ($E_l^{10}$ = 56.88 m) (Table 2). For this configuration of the optimal sensors network, the source intensity in 80% of the trials are estimated within a factor of two to their true release rates (Tables

2&3).

For all 20 trials, the averaged location error $E_l^{13}$ is 17.42 m for the optimal networks formed by 13 sensors, which is smaller than the averaged $E_l^{10}$ = 19.20 m obtained with 10 sensors (Tables 2&3). The location error is observed minimum in trial 5 ($E_l^{13}$ = 2.13 m) and maximum in trial 16 ($E_l^{13}$ = 63.04 m) (Table 2). For this optimal network, in 80% of the trials, the source intensity is estimated within a factor of two to the actual emission rates. It was noted from the evaluation results that the

increase in the number of sensors in a network has little influence on the accuracy of the estimated intensity (Tables 2&3).

In some trials, it was also noted that the distance of an estimated source to real source can decreases with a decrease in sensors number and are also increases with the number of sensors in some other cases. It is may be because the information added by a new sensor was not necessarily beneficial. As it is noticeable that in a particular meteorological condition (i.e. wind direction, speed and atmospheric stability), some of the sensors in a network may have little contribution to the STE. So, increasing the

number of the sensors may not always provide the best estimation because with addition of the more no. of sensors, we also add more model and measurements errors in the estimation process. These errors can affect the source estimation results in some trials. In some cases, it may also depend on sensitiveness of the added sensor's position in an extended optimal network to the source estimation. It is also noted that for a monitoring network, not only the number of sensors but also the sensors distribution form (or sensor position) affect the information captured from network.

In fact, both optimal networks for each trial show a diversity of structures independently of the number of sensors considered. For this, the *skeleton* was used to analyze the heterogeneity of the structures of different optimal networks. A skeleton with 7 sensors is considered as a strong common base for the networks. This is the case for trials 3,6,14,15&20 (Table 2). It is noted that the overall results obtained are comparable (little differences between the results obtained by the networks). For these networks, a strong common base leads to a near global optimum. If we consider networks with a weak common base,

the *skeleton* was formed of up to 3 sensors, particularly in trials 1 and 11. The performances do not systematically converge

independently of the size of networks. Thus, for trial 1, better results were obtained with a network formed by 13 sensors compared to that by 10 sensors. This result reflects that the algorithm with the network formed by 13 sensors, converges probably toward a near global optimum. For trial 11 also, it was noted that the performances obtained by the two networks are identical. This shows that the networks with different sensors configurations may lead to a near overall optimum. This result is in coherence with Kovalets et al. (2011) and Efthimiou et al. (2017). Considering a network of 10 sensors, they shown for the same experimental data that the best source reconstruction is possible with only 5% or 10% of the total network combinations, randomly selected.

Considering the networks of intermediate structures, with *skeletons* varying from 4 to 6 sensors, notedly for trials 2,4,5,7,8,9, 12,13,16,17,18,&19, no obvious trend is noticed. These results tend to show that for a given trial, one or more optimal networks can satisfy the conditions of a near overall optimum (to be minimized). The obtained optimal networks may have a more or less common structure (having a greater or lesser number of *skeleton*).

Moreover, uncertainties calculated for different network sizes do not show an obvious trend. Indeed, a general relationship between the number of samplers and the uncertainties is not obvious. One notice that changing size of the network (increasing or decreasing the number of sensors) can lead to the growth or diminution of the uncertainties in the source parameters estimation. As an example, for Trial 7 uncertainties grow while for Trial 17 uncertainties diminish (Table 2).

It should be noted that this study deals with the case of reducing number of sensors in order to obtain an optimal network from an existing large network. This optimization was carried out under the constraints of an existing network of the original 40 sensors in the MUST field experiment. If one compares the performances of the obtained optimal monitoring networks of smaller sizes with the initial (original) network of 40 sensors in MUST environment, both optimal networks provide satisfactory estimations of unknown source parameters. The 40 sensors network gives an averaged location error of 14.62 m for all trials and the release rate were estimated within a factor of two in 75% trials. However, reducing the number of sensors to $\sim 1/3^{rd}$ from the original 40 sensors, the 13 sensors optimal networks also give comparable source estimations performance with an averaged location error of 17.42 m. Even with the 13 sensors optimal networks, source intensities in 80% trials were accurately estimated within a factor of two. Similarly for 10 sensors optimal networks, the averaged location error (=19.20 m) is slightly larger than that obtained from 13 and 40 sensors networks. However, reducing the number of sensors to $1/4^{th}$ gives extra advantages in case of the limited available sensors for a network in emergency scenarios of an accidental or deliberated releases in complex urban environments.

It should also be noted that the optimization evaluation in this study is performed using the MUST set of measurements and this makes it more likely that the resulting sensor configuration performs well reconstructing the source (that "the same measurements shouldn't be used for the optimization and for the reconstructions"). However, this doesn't limit the application of the proposed methodology for some important practical applications like the accurate emissions estimation. In fact, this can be considered as a limitation of the used data for this application as for a complete process of the optimization and then the evaluation one requires a sufficiently long set of measurements so that the whole data can be divided into two parts: (i) first part for the designing an optimal sensor network and (ii) the second part for the evaluation of the designed optimal network. However, the durations of the releases in the MUST field experiment were not sufficiently

large to divide the whole data from a test release separately into two parts for designing the optimal sensor network and then its evaluation. However, in further evaluation of the resulting optimal sensor configuration, a different set of the concentration measurements can be constructed by adding some noise to the measurements. For a continuous release in steady atmospheric conditions, the average value of the steady concentration in a test release is not expected to deviate drastically from the mean values in each segment of the complete data. So this new set of the concentration measurements with added noise can partially fulfill the purpose of the evaluation of a designed optimal network. As shown in Table 2, the errors in the estimated source parameters are small even with the new sets of concentration measurements constructed by adding 10% Gaussian noise. This exercise shows that even if we have utilized a partially different set of the measurements for the evaluation of the optimal networks, the optimal networks have almost the same level of the source detection ability in an urban-like environment. However, a realistic data is required for further evaluation of the optimization methodology.

Although the MUST field experiment has been widely utilized for validation of the atmospheric dispersion models and the inversion methodologies for unknown source reconstruction in an urban-like environment, its experimental domain was only approx. 200 m × 200 m (with buildings represented by a grid of containers) and can be considered small for a real urban environment. Thus, it may not quite represent a real urban region in terms of scale, meteorological variability, or non-uniform terrain or roughness/canopy structure. However, the methodology presented here is general in nature to apply to a real urban environment also. The methodology involves the utilization of the CFD model which generally can include the effects of the urban geometry, meteorological variability, or non-uniform terrain or roughness/canopy structure in a real urban environment. It is also to note that, the optimal network design would also depend on diurnal and spatial variability in meteorological conditions which may increase or decrease the optimum number of sensors and also may change the 'best positions' to be instrumented by sensors.

## 7 Conclusions

This study describes an approach for the optimization of a optimally reducing the size of an existing monitoring network of the sensors in a geometrically complex urban environment. It is a matter of reducing the size of networks while retaining the capabilities of estimating an unknown source in an urban region. Given an urban-like environment of the MUST field experiment, the renormalization inversion method was chosen for the Source Term Estimation. It was coupled with the CFD model fluidyn-PANACHE for generation of the adjoint fields. Combinatorial optimization by the simulated annealing consisted in choosing a set of sensors which leads to an optimal monitoring network and allows an accurate unknown source estimation. This study extended an application of the renormalization data-assimilation theory for the definition of optimality criterion for the optimal network design to estimate an unknown continuous point release This study demonstrates how the renormalization inversion technique can be applied to optimally reducing the size of an existing large network of the concentration samplers for quantifying a continuous point source in an urban-like environment with almost the same level of source detection ability as the original network with larger number of samplers.

The numerical calculations were performed by coupling the simulated annealing stochastic algorithm to the renormalization inversion technique and the CFD modeling approach to optimize the optimally reduce the size of an existing monitoring network in urban-like environment of the MUST field experiment. The optimal networks were constructed to reduce size of the original networks (40 sensors) to approx. one-third (13 sensors) and to one-fourth (10 sensors). The 10 and 13 sensors optimal networks have estimated the averaged location errors of 19.20 m and 17.43 m, respectively, and have comparable source estimations performance with an averaged location error of 14.62 m from the original 40 sensors network. In 80% of the trials , the emission rates with trials with optimal networks of 10 and 13 sensorsnetworks were , the emission rates are estimated within a factor of two which to the actual release rates. These are also comparable with the factor of two source intensities to performance of the original 40-sensors network where in 75% trials with the original networkof the trials, the releases was estimated with a factor of two to the actual release.

It was shown that in most of the MUST trials, the number of sensors in optimal networks slightly influences the location error of an estimated source and this error tends to increase as the number of sensors decreases. In 20 MUST trials, an analysis of the networks formed by 10 & 13 sensors revealed the heterogeneity of their structures in an urban domain. It was observed that for some trials, optimal networks had a strong common structure. This tends to prove that a certain number of sensors have a primordial role in reconstructing an unknown source. It would reflect a fact that the disjoint sets of sensors can lead to the best estimate of an unknown source in an urban region. This opens enormous prospects for assessing the relative importance of each sensor in a source reconstruction process in an urban environment. Defining a global optimal network for all meteorological conditions is a complex problem, but of greater importance that one may want to pursue. This challenge consists to define an optimal static network able to reconstruct the sources in all varied meteorological conditions. This information can be of great importance to determine an optimal monitoring network by reducing the number of sensors for characterization of the unknown emissions in the complex urban or industrial environments.

*Data availability.* The authors received access to the MUST field experiment dataset from Dr. Marcel Koňig of Leibniz Institute for Tropospheric Research. The MUST database was officially available from the Defense Threat Reduction Agency (DTRA) at https://must-dpg.dpg.army.mil/.

## Appendix A: Weight function

Issartel et al. (2007) demonstrated that a weight function, which reduces the artifacts of the adjoint functions at the measurement points, must verify the following renormalization criterion:

$$\mathbf{a}_w^T(\mathbf{x})\mathbf{H}_w^{-1}\mathbf{a}_w(\mathbf{x}) \equiv 1 \tag{A1}$$

Following an iterative algorithm by Issartel et al. (2007), $w(\mathbf{x})$ is determined as:

$$w_0(\mathbf{x}) = 1, \quad \text{and} \quad w_{k+1}(\mathbf{x}) = w_k(\mathbf{x})\sqrt{\mathbf{a}_{wk}^T(\mathbf{x})\mathbf{H}_{wk}^{-1}\mathbf{a}_{wk}(\mathbf{x})} \tag{A2}$$

## Appendix B: Identification of point source

Consider a point source of continuous release at a position $\mathbf{x_o} = (x_o, y_o)$ and with the intensity $q_o$. The point source is thus expressed as a function of the preceding parameters: $\mathbf{s}(\mathbf{x}) = q_o\delta(\mathbf{x} - \mathbf{x_o})$. The relationship between the source and the measurements (Eq. (3)) becomes: $\boldsymbol{\mu} = q_o\mathbf{a}_w(\mathbf{x_o})w(\mathbf{x_o}) + \boldsymbol{\epsilon}$. By replacing the measurement term in Eq. (4), one obtains:

$$\mathbf{s_w} = q_o w(\mathbf{x_o})\mathbf{A}_w^\mathsf{T}\mathbf{H}_w^{-1}\mathbf{a}_w(\mathbf{x_o}). \tag{B1}$$

$\mathbf{s}_w$ reaches its maximum at position $\mathbf{x_o}$ as the renormalization criterion (Eq. (A1)) is satisfied only at this position $\mathbf{x_o}$. Thus, $\mathbf{s}_w(\mathbf{x})$ at $\mathbf{x_o}$ becomes:

$$\mathbf{s_w}(\mathbf{x_o}) = q_o w(\mathbf{x_o}), \tag{B2}$$

which estimates the source intensity $q_o = \mathbf{s}_w(\mathbf{x_o})/w(\mathbf{x_o})$.

## Appendix C: Derivation of the Cost function

A cost function is defined (based on the renormalization theory) as a function that minimizes the quadratic distance between the observed and the simulated measurements according to the $\mathbf{H}_w$ norm (Issartel et al., 2012). $\mathbf{H}_w$ is the *Gram matrix* defined in section 2.2. The quadratic distance between the real and the simulated concentration measurements according to the $\mathbf{H}_w$ norm is given by :

$$J = \|\boldsymbol{\mu} - \hat{\boldsymbol{\mu}}\|^2_{\mathbf{H}_w^{-1}} = \frac{1}{2}\left[(\boldsymbol{\mu} - \hat{\boldsymbol{\mu}})^\mathsf{T}\mathbf{H}_w^{-1}(\boldsymbol{\mu} - \hat{\boldsymbol{\mu}})\right] \tag{C1}$$

When considering a point source, $\hat{\boldsymbol{\mu}}$ is written by $\hat{\boldsymbol{\mu}} = q_o\mathbf{a}_w(\mathbf{x})w(\mathbf{x})$, where $q_o$ and $\mathbf{x}$ are respectively the intensity and the position of a point source. By replacing $\hat{\boldsymbol{\mu}}$ in Eq. (C1), one obtains (Sharan et al., 2012; Issartel et al., 2012):

$$J = J(q_o, \mathbf{x}) = \frac{1}{2}\left[(\boldsymbol{\mu} - q_o\mathbf{a}_w(\mathbf{x})w(\mathbf{x}))^\mathsf{T}\mathbf{H}_w^{-1}(\boldsymbol{\mu} - q_o\mathbf{a}_w(\mathbf{x})w(\mathbf{x}))\right] \tag{C2}$$

For a fixed $\mathbf{x}$ in Eq. (C2), $J$ reaches a strict local minimum if following two conditions are satisfied:

$$\frac{\partial J(q_o, \mathbf{x})}{\partial q_o} = 0 \tag{C3}$$

$$\frac{\partial^2 J(q_o, \mathbf{x})}{\partial q_o^2} > 0 \tag{C4}$$

For each fixed $\mathbf{x}$, the first condition (Eq. (C3)) gives an estimate ($\tilde{q}_0$) of $q_0$ as: $\tilde{q}_0 = \frac{\mathbf{a}_w^T(\mathbf{x})\mathbf{H}_w^{-1}\boldsymbol{\mu}}{w(\mathbf{x})}$. The second condition (Eq. (C4)) is always satisfied as $\frac{\partial^2 J(q_o, \mathbf{x})}{\partial q_o^2} = w^2(\mathbf{x}) > 0, \forall\mathbf{x}$ (Sharan et al., 2012). Corresponding to the estimate $\tilde{q}_0$ from the first condition (Eq. (C3)), the cost function $J$ from Eq. (C2) leads to the following expression (Issartel et al., 2012):

$$J(\tilde{q_0}, \mathbf{x}) = \frac{\boldsymbol{\mu}^\mathsf{T}\mathbf{H}_w^{-1}\boldsymbol{\mu}}{2}\left[1 - \frac{\mathbf{s}_w^2}{\boldsymbol{\mu}^\mathsf{T}\mathbf{H}_w^{-1}\boldsymbol{\mu}}\right] \tag{C5}$$

where $\mathbf{s}_w$ is same as given in Eq. (4) and $\boldsymbol{\mu}^T \mathbf{H}_w^{-1} \boldsymbol{\mu}$ is a positive constant. Considering Eq. (C5), it is obvious that the minimization of $J$ also corresponds to the maximization of the term $\frac{\mathbf{s}_w^2}{\boldsymbol{\mu}^T \mathbf{H}_w^{-1} \boldsymbol{\mu}}$ or minimization of term $\left[ 1 - \frac{\mathbf{s}_w^2}{\boldsymbol{\mu}^T \mathbf{H}_w^{-1} \boldsymbol{\mu}} \right]$. Accordingly, the minimum value of the cost function $J$ in Eq (C5) leads to the following expression of the cost function (say $J_s(\mathbf{x})$) to minimize:

$$J_s(\mathbf{x}) = 1 - \frac{\mathbf{s}_w^2}{\boldsymbol{\mu}^T \mathbf{H}_w^{-1} \boldsymbol{\mu}} \tag{C6}$$

A global minimum of the cost function $J_s(\mathbf{x})$ is evaluated by the SA algorithm.

*Acknowledgements.* The authors would like to thank Dr. Marcel Koňig of Leibniz Institute for Tropospheric Research and the Defense Threat Reduction Agency (DTRA) for providing access to the MUST field experiment dataset. The authors gratefully acknowledge Fluidyn France for use of the CFD model fluidyn-PANACHE. We also thank to Dr. Claude Souprayen from Fluidyn France for useful discussions.

10    Finally we thank to the reviewers Dr. J. Pudykiewicz, Dr. G.C. Efthimiou, one anonymous reviewer, and the topical editor Dr. Ignacio Pisso for their detailed and technical comments that helped to improve this study.

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

**Table 1.** The values of the meteorological (wind speed ($S_{04}$), wind direction ($\alpha_{04}$) at 4 m level of mast S), turbulence (the Obukhov length ($L$), friction velocity ($u_*$), turbulent kinetic energy ($k$) at 4 m level of tower T), and source parameters (source height ($z_s$), release duration ($t_s$), release rate ($q_s$)) in 20 selected trials of the MUST field experiment (Biltoft, 2001; Yee and Biltoft, 2004). Here, Trial Nos. 1-20 are assigned for just continuation and simplicity and these are not correspond to the same assigned trial no. for a given Trial name in the MUST experiment.

| Trial No. | Trial Name (JJJhhmm) | $q_s$ (l/min) | $t_s$ (min) | $z_s$ (m) | $S_{04}$ (m/s) | $\alpha_{04}$ (deg) | $u_*$ (m/s) | $L$ (m) | $k$ ($m^2s^{-2}$) |
|---|---|---|---|---|---|---|---|---|---|
| 1 | 2640138 | 175 | 21 | 0.15 | 2.35 | 17 | 0.26 | 91 | 0.359 |
| 2 | 2640246 | 200 | 15 | 0.15 | 2.01 | 30 | 0.25 | 62 | 0.306 |
| 3 | 2671852 | 200 | 22 | 0.15 | 3.06 | -49 | 0.32 | 330 | 0.436 |
| 4 | 2671934 | 200 | 15 | 1.8 | 1.63 | -48 | 0.08 | 5.8 | 0.148 |
| 5 | 2672033 | 200 | 15 | 1.8 | 2.69 | -26 | 0.17 | 4.8 | 0.251 |
| 6 | 2672101 | 200 | 14 | 0.15 | 1.89 | -10 | 0.16 | 7.7 | 0.218 |
| 7 | 2672150 | 200 | 16 | 0.15 | 2.30 | 36 | 0.35 | 150 | 0.409 |
| 8 | 2672213 | 200 | 15 | 1.8 | 2.68 | 30 | 0.35 | 150 | 0.428 |
| 9 | 2672235 | 200 | 15 | 2.6 | 2.32 | 36 | 0.26 | 48 | 0.387 |
| 10 | 2672303 | 200 | 19 | 1.8 | 2.56 | 17 | 0.25 | 74 | 0.367 |
| 11 | 2681829 | 225 | 15 | 1.8 | 7.93 | -41 | 1.10 | 28000 | 1.46 |
| 12 | 2681849 | 225 | 16 | 0.15 | 7.26 | -50 | 0.76 | 2500 | 0.877 |
| 13 | 2682256 | 225 | 15 | 0.15 | 5.02 | -42 | 0.66 | 240 | 0.877 |
| 14 | 2682320 | 225 | 15 | 2.6 | 4.55 | -39 | 0.50 | 170 | 0.718 |
| 15 | 2682353 | 225 | 15 | 5.2 | 4.49 | -47 | 0.44 | 120 | 0.727 |
| 16 | 2692054 | 225 | 22 | 1.3 | 3.34 | 39 | 0.36 | 170 | 0.362 |
| 17 | 2692131 | 225 | 17 | 1.3 | 4.00 | 39 | 0.42 | 220 | 0.582 |
| 18 | 2692157 | 225 | 15 | 2.6 | 2.98 | 43 | 0.39 | 130 | 0.505 |
| 19 | 2692223 | 225 | 15 | 1.3 | 2.63 | 26 | 0.35 | 120 | 0.484 |
| 20 | 2692250 | 225 | 17 | 1.3 | 3.38 | 36 | 0.37 | 130 | 0.537 |

**Table 2.** Source estimation results from the different monitoring networks for each selected trial of the MUST field experiment. $E_l^p$ and $E_q^p$ respectively denote the location error (m) and ratio of the estimated to true source intensity with the corresponding monitoring network. Here, the superscript $p$ on $E_l^p$ & $E_q^p$ represents the no. of sensors in an optimal network. *Skeleton* refers to the number of sensors common to the optimal networks of 10 and 13 sensors for a given MUST trial.

| Run No. | $E_l^{40}$ (m) | $E_l^{13}$ (m) | $E_l^{10}$ (m) | $E_q^{40}$ | $E_q^{13}$ | $E_q^{10}$ | *Skeleton* sensors |
|---|---|---|---|---|---|---|---|
| 1 | 3.3±1.3 | 19.60±12.13 | 33.76±5.30 | 0.92±0.08 | 1.04±0.23 | 1.24±0.22 | 3 |
| 2 | 42.9±23.8 | 31.91±8.80 | 56.88±9.51 | 4.01±1.57 | 3.21±0.41 | 5.12±3.63 | 4 |
| 3 | 10.8±1.6 | 9.01±2.47 | 9.01±3.02 | 1.17±0.27 | 0.71±0.16 | 0.71±0.16 | 7 |
| 4 | 22.8±7.7 | 18.07±1.84 | 18.07±2.61 | 0.27±0.35 | 0.83±0.21 | 0.83±0.26 | 6 |
| 5 | 21.9±2.1 | 2.13±2.54 | 11.56±4.21 | 0.57±0.07 | 0.95±0.05 | 0.67±0.05 | 6 |
| 6 | 5.0±1.6 | 6.96±0.19 | 6.96±0.00 | 2.14±0.60 | 1.04±0.06 | 1.04±0.04 | 7 |
| 7 | 12.4±9.1 | 18.85±9.08 | 12.99±1.67 | 0.41±0.49 | 3.11±0.51 | 1.06±0.07 | 4 |
| 8 | 15.8±12.1 | 12.86±1.28 | 15.79±1.05 | 2.22±0.90 | 1.32±0.34 | 1.76±0.11 | 6 |
| 9 | 7.7±1.2 | 8.20±0.35 | 8.08±0.00 | 1.37±0.07 | 3.06±0.17 | 7.55±0.39 | 5 |
| 10 | 8.8±3.0 | 8.00±4.57 | 8.00±5.68 | 1.08±0.19 | 1.08±0.77 | 1.08±1.07 | 8 |
| 11 | 19.8±5.0 | 17.19±12.00 | 17.19±7.06 | 1.67±0.12 | 1.62±0.40 | 1.62±0.26 | 3 |
| 12 | 7.4±6.6 | 5.43±11.69 | 10.22±9.10 | 0.95±0.06 | 0.85±0.28 | 0.20±0.04 | 4 |
| 13 | 7.7±0.6 | 8.63±4.36 | 8.63±3.86 | 0.97±0.07 | 0.78±0.18 | 0.78±2.05 | 4 |
| 14 | 2.2±1.9 | 5.50±2.98 | 5.50±3.88 | 1.42±0.17 | 0.88±0.24 | 0.88±0.40 | 7 |
| 15 | 1.1±1.0 | 30.23±2.14 | 37.98±0.72 | 1.88±0.09 | 0.57±0.07 | 0.17±0.01 | 7 |
| 16 | 26.7±4.9 | 63.04±6.84 | 29.80±9.86 | 1.70±0.06 | 0.29±0.06 | 0.67±0.23 | 5 |
| 17 | 7.0±1.9 | 14.07±2.78 | 23.05±10.44 | 0.90±0.05 | 1.10±0.04 | 1.52±0.16 | 6 |
| 18 | 14.3±11.0 | 12.83±4.18 | 12.83±4.61 | 1.15±0.46 | 1.15±0.16 | 1.15±0.21 | 6 |
| 19 | 22.3±6.4 | 10.77±4.25 | 13.46±4.8 | 1.76±0.16 | 0.99±0.20 | 0.83±0.25 | 6 |
| 20 | 32.5±1.8 | 45.23±1.78 | 44.29±0.31 | 0.83±0.04 | 1.68±0.06 | 1.56±0.06 | 7 |

**Table 3.** Statistics for the source reconstruction results from each monitoring network. Here, $E_l^p$ is the averaged location error for all 20 trials corresponding to each network. FA$g$ represents the percentage number of trials in which the source intensity is estimated within a factor of $g$.

| Sensors ($p$) | $E_l^p$ (m) | FA4 | FA3 | FA2 |
|---|---|---|---|---|
| 40 | 14.62 | 95 | 90 | 75 |
| 13 | 17.42 | 100 | 80 | 80 |
| 10 | 19.20 | 80 | 80 | 80 |

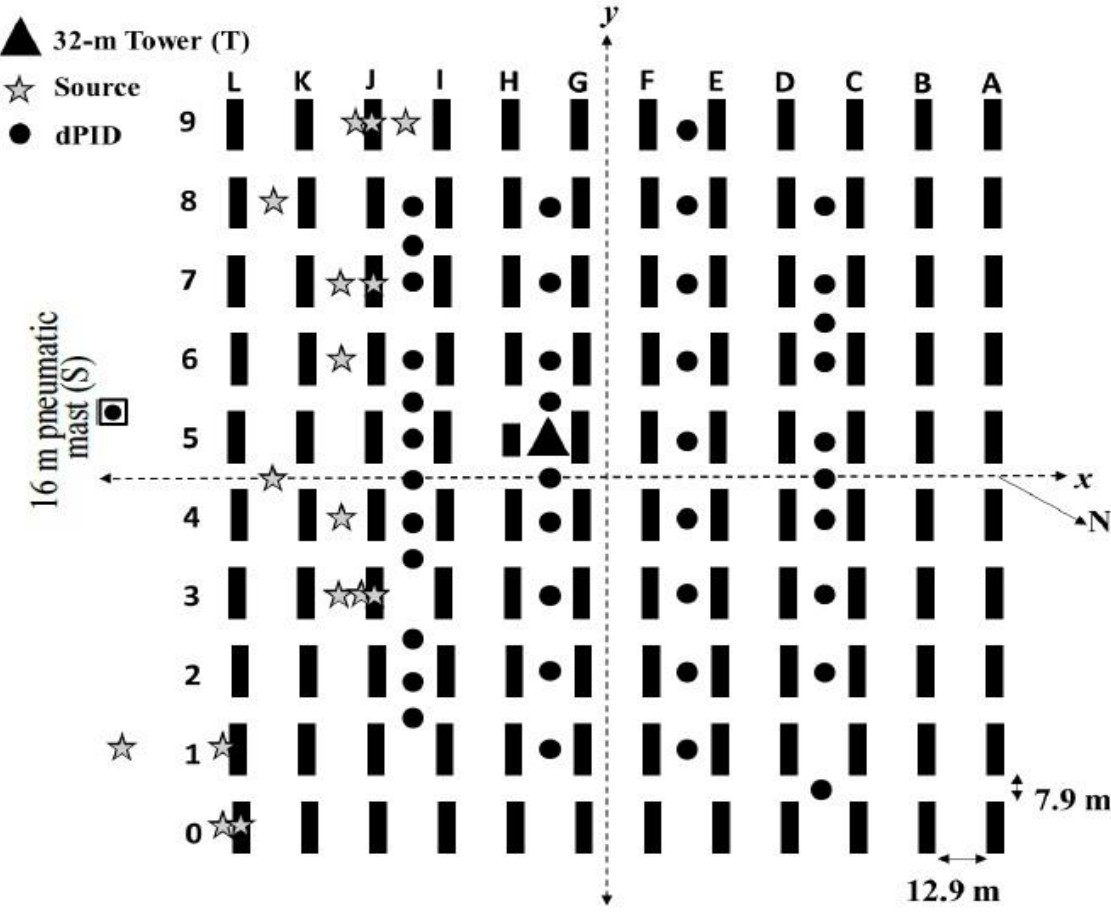

**Figure 1.** A schematic diagram of the MUST geometry showing 120 containers and source (stars) and receptors (black filled circles) locations. In a given trial - only one source was operational.

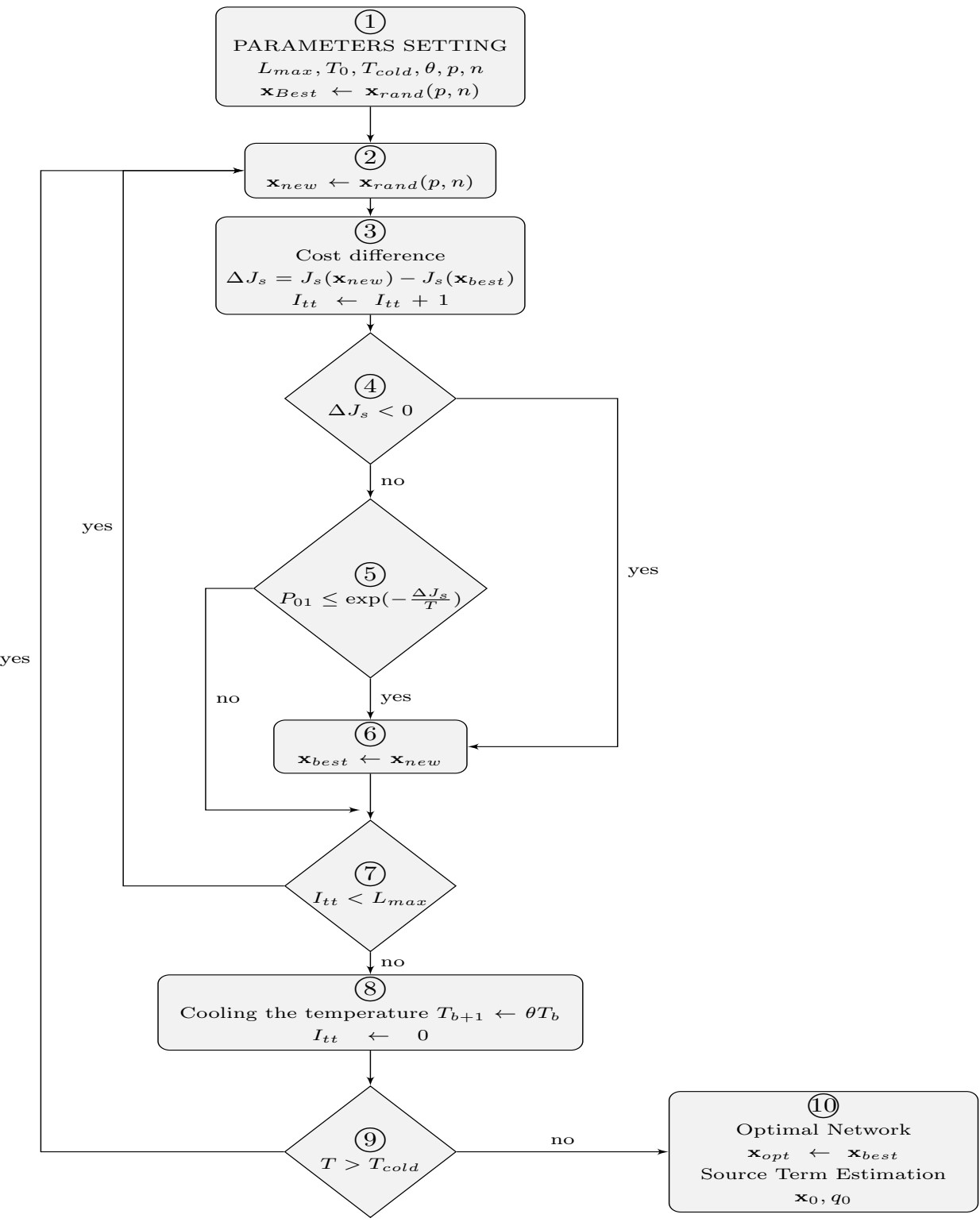

**Figure 2.** Flow diagram of the Simulated Annealing procedures to determine an optimized monitoring network.

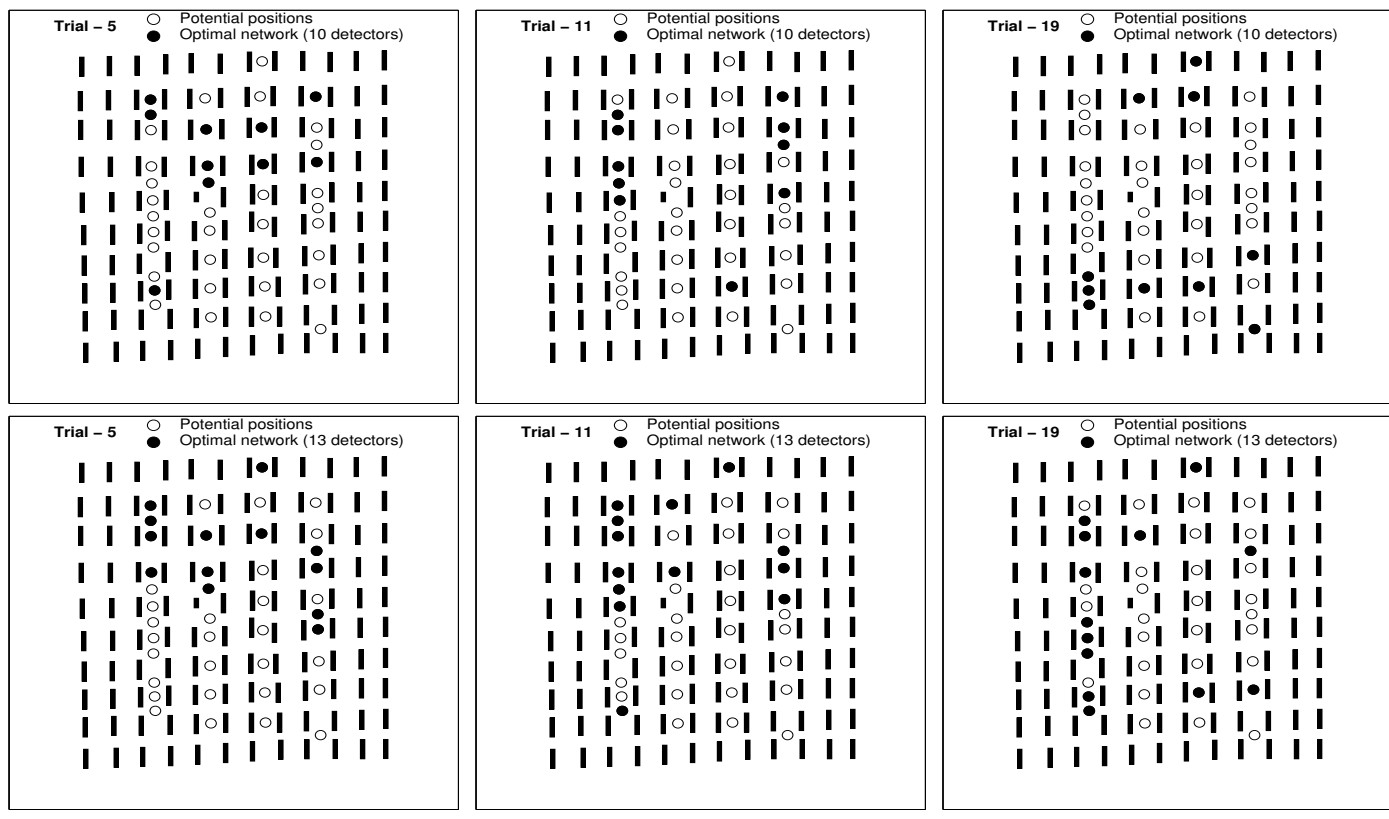

**Figure 3.** The optimal networks of 10 (first row) and 13 sensors (second row) respectively for trials 5 (very stable), 11 (neutral), and 19 (stable). Blank and filled black circles respectively represent the all (40) potential positions and the optimal positions of sensors.

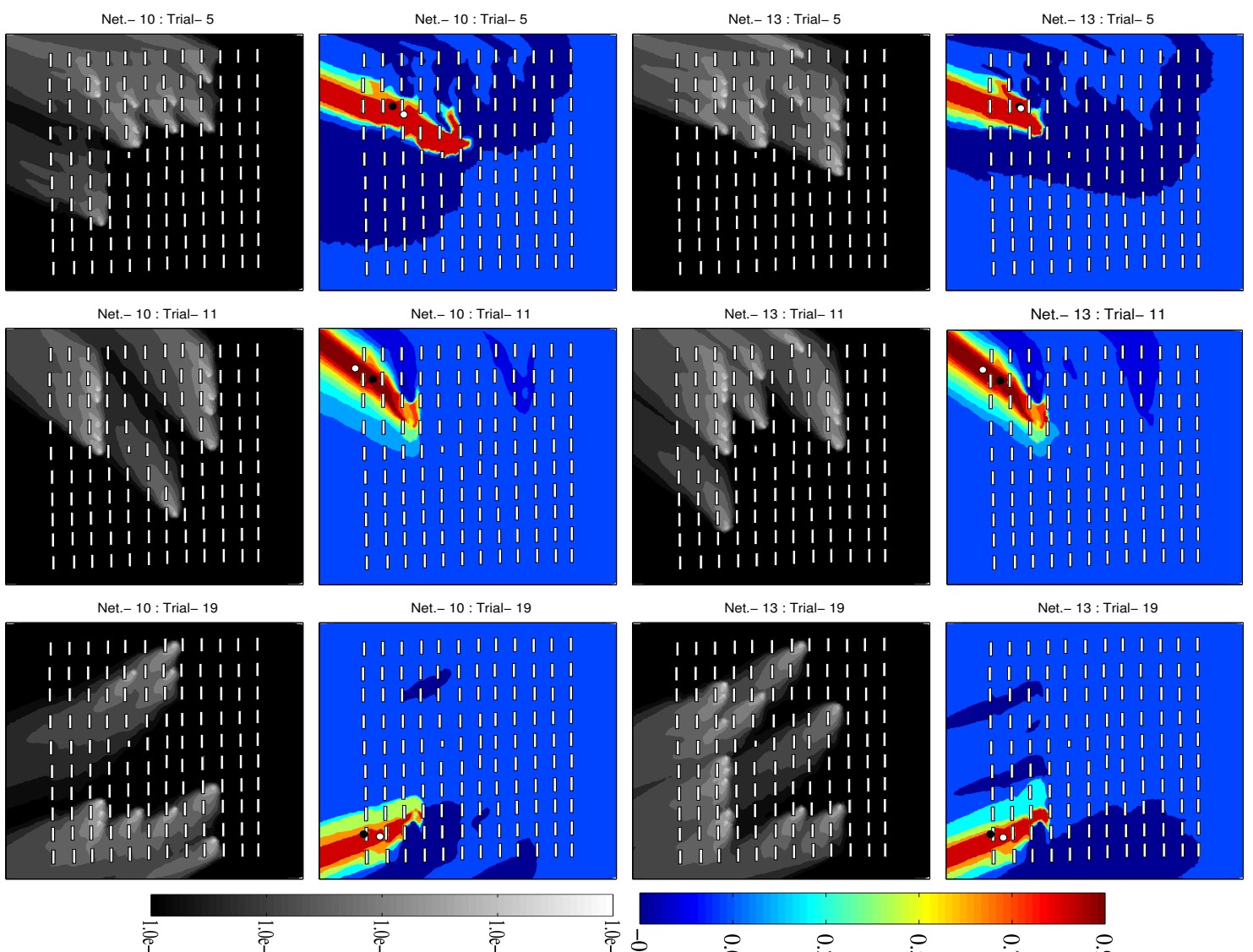

**Figure 4.** Isopleths of the renormalized weight function $(w(\mathbf{x}))$ (gray colored in first and third columns) and the normalized source estimate function $(\mathbf{s}_w^n(\mathbf{x}) = \mathbf{s}_w(\mathbf{x})/\max(\mathbf{s}_w(\mathbf{x})))$ (colored in second and fourth columns) for both optimal networks of 10 and 13 sensors respectively for trials 5 (very stable), 11 (neutral), and 19 (stable). The black and white filled circles respectively represent the true and estimated source locations.