# Peer review of "An Optimization for Reducing the Size of an Existing Urban-like Monitoring Network for Retrieving an Unknown Point Source Emission"

_Geoscientific Model Development, 2018_

## Referee Comment (RC1) · Dr Efthimiou (Referee) · 19 Mar 2018

The idea of the paper is to achieve the best result with as less as possible information. This idea is very innovative and I support any new effort. The application is the atmospheric dispersion in urban areas. The goal is to find the source when we know the flow field and the real concentration measurements.

I have one major comment/question.

When authors try to validate this approach they compare results of source inversion (distance to true source etc) with 'optimal network' of 10 sensors with the results obtained by the full network (40 sensors). Why don't they directly compare results of 'optimal' network of 10 sensors with the results of other networks of 10 sensors? Of

course, there are too many of such networks. But by application of the combination same procedure as we did in Kovalets et al (2011) and Efthimiou et al (2017) they at least could prove that their 'optimal network' yields the results within say best 5 or 10% of the results that could be achieved with 10 sensors.

1) I.V. Kovalets, S. Andronopoulos, A.G. Venetsanos, J.G. Bartzis, Identification of strength and location of stationary point source of atmospheric pollutant in urban conditions using computational fluid dynamics model, Math Comput Simulat, 82 (2011) 244-257.

2) G.C. Efthimiou, I.V. Kovalets, A. Venetsanos, S. Andronopoulos, C.D. Argyropoulos, K. Kakosimos, An optimized inverse modelling method for determining the location and strength of a point source releasing airborne material in urban environment, Atmos. Environ., 170 (2017) 118-129.

---

## Short Comment (SC1) · 20 Mar 2018

Dear authors,

as explained in
https://www.geoscientific-model-development.net/about/manuscript_types.html                ,
model description paper must include the name and version number of the published
method in the title. Therefore, name and version your method in the title upon revision
of the article.

Secondly, the algorithm described in the article needs to be published. Either as supplement to the paper are in a permanent archive providing a DOI and the DOI needs
to be cited in the code availability section. Please add the code and /or the respective

information latest upon revision of your article.

Best Astrid Kerkweg

---

## Short Comment (SC2) · 2 Apr 2018

We thank Dr. Efthimiou for reviewing our paper and for the positive feedback. We appreciate his questions:

1. Why we don't directly compare results of 'optimal' network of 10 sensors with the results of other networks of 10 sensors?

2. Why we compare results of source inversion (distance to true source, etc.) with 'optimal network' of 10 sensors with the results obtained by the full network (40 sensors)?

Our responses to the specifics questions are listed below:

1. The comparison with networks of the same size (10 sensors for example), is per-

[Figure]

formed implicitly during the optimization process. The Simulated Annealing used in this study compares at each iteration two networks (of the same size) and retains the 'best one'. The networks are generated randomly as in Kovalets et al (2011) and Efthimiou et al (2017) and the comparison is based on the cost function named Normalized Errors 'Js' and inspired from the renormalized data assimilation method. This cost function quantifies the quadratic distance between the observed and the simulated measurements according to the normalized Gram matrix 'Hw'. The challenge was to design the networks without using a priori the parameters of the real source and without considering an acceptance criteria of networks quality (rH $\leq$ 15 m, rV $\leq$ 2.5 m, $\delta$q $\leq$ 4) as performed in Kovalets et al (2011) and Efthimiou et al (2017). Based on 'Js', the 'optimal network' produce the 'best' description of the observations (i.e. corresponds to the minimal quadratic distance between the observed and the simulated measurements) and permits a posteriori to reconstruct its origin. In the attached figures 1, 2 & 3 is presented the evolutions of the cost functions for trials 5, 11 & 19 during the optimization process. Since the search space is quite large so about 3E+04 networks of 10 sensors are compared.

2. The results obtained by the optimal networks of 10 and 13 sensors are compared with the original network of 40 sensors for two reasons: As in practice the number of measurements is very limited, this comparison allows us to conclude that in urban areas source reconstruction can be conducted with networks of limited number of sensors and to confirm that the reduction of networks size don't degrade significantly its efficiency. For more details, the choice of the number of sensors (10 and 13) is fixed after observing that an acceptable estimation of the source in majorities of the trials was enabled by using minimum 8 sensors. Also by using more than 13 sensors optimal networks, the errors in source parameters estimation are stable and does not improve significantly, Kouichi, (2017). For this reason, the optimization were constructed and evaluated for sizes 10 and 13 (1/4th and about 1/3rd of the original network of 40 sensors).

1) I.V. Kovalets, S. Andronopoulos, A.G. Venetsanos, J.G. Bartzis, Identification of strength and location of stationary point source of atmospheric pollutant in urban conditions using computational fluid dynamics model, Math Comput Simulat, 82 (2011) 244-257.

2) G.C. Efthimiou, I.V. Kovalets, A. Venetsanos, S. Andronopoulos, C.D. Argyropoulos, K. Kakosimos, An optimized inverse modelling method for determining the location and strength of a point source releasing airborne material in urban environment, Atmos. Environ., 170 (2017) 118-129

3) Kouichi, H. (2017), Sensors networks optimization for the characterization of atmospheric releases source, Theses, Université Paris Saclay, France. https://hal.archives-ouvertes.fr/tel-01593834v2

**Fig. 1.** Figure 1: Evolution of the cost function for trial 5

**Fig. 2.** Figure 2: Evolution of the cost function for trial 11

[Figure]

**Fig. 3.** Figure 3: Evolution of the cost function for trial 19

[Figure]

---

## Short Comment (SC3) · 2 Apr 2018

Dear Editor Dr. Kerkweg,

We thank you for your comment. Your remarks will be taken into account latest upon revision of the article.

Best regards,

Hamza Kouichi.

---

## Referee Comment (RC2) · Anonymous Referee #2 · 4 Apr 2018

In this paper, the authors formulate a simulated annealing algorithm with a renormalization inversion algorithm coupled to a CDF flow and dispersion model and apply it to the Mock Urban Setting Test (MUST) tracer field experiment (which simulates an 'urban-like' environment). The aim of the work is to demonstrate how the inversion technique presented can be useful in optimally placing a smaller number of concentration samplers for quantifying a continuous point source with almost the same level of source detection ability as the original larger number of samplers.

The paper is well written, but in my opinion requires a major revision. My comments are as follows:

Main comments:

[Figure]

1) The MUST experiments took place under neutral to stable and strongly stable conditions. However, the CFD model used is for neutral conditions and does not include the effects of atmospheric stability over the urban area (the only stability effects included are through the specification of inflow boundary conditions). Atmospheric stability has a profound impact on dispersion and would thus influence the adjoint functions. The authors should discuss the consequences of its neglect on the results and the errors it introduces.

2) I have reservations about the usefulness of the methodology presented in real-world urban environments. The title of the paper states 'urban monitoring network' but there are no real urban configurations used. The MUST experimental domain was only 200 m x 200 m (with buildings represented by a grid of containers) which cannot quite represent an urban area in terms of scale, meteorological variability, or non-uniform terrain or roughness/canopy structure. So in a way the present study does not explore any aspects that are specific to urban environments. The authors should discuss this, particularly how their methodology could be applied and its limitations in real-world urban cases. Following on, the title of the paper should say 'urban-like' or something similar instead of 'urban'.

3) There were a total of 40 concentration samplers. In their optimisation, the authors arbitrarily fixed the number of samplers to 13 and 10 and then determined the optimum positions of these reduced number of samplers from the original 40 samplers. A better question to answer would have been "what is the minimum number of samplers required and what their positions are in order to quantify the source with a given degree of confidence or accuracy?"

The present optimisation is based on fixed meteorological conditions in a trial. In a real situation, the network design would also depend on diurnal and spatial variability in meteorological conditions (e.g. wind direction) which may increase or decrease the optimum number of sites. This, however, is not in the scope of the present study. Perhaps as a future study, the authors may consider using data from full scale field

measurements such as Salt Lake City Urban 2000 experiment.

4) Dense gas effects are included. How are they taken into account (or inverted) in the backward (i.e. retro plume) dispersion calculation for adjoint functions?

5) What is the uncertainty in the source estimation results in Table 2? Is the approach capable of providing uncertainty estimates (like the Bayesian one)?

6) How does the uncertainty in the results in Table 2 change as the number of samplers is changed? Have you included model and measurement uncertainties in the methodology?

7) Section 2.3: Is there a sensitivity of the source estimation / optimisation to how the weight function is selected? Could there be any other choices of the weight function?

8) Did you specify any a priori bounds on the estimated source position and source emission rate? If yes, what were they?

9) What is the advantage of the present technique compared to, say, the Bayesian approach which also provides probability associated with the solution?

10) Page 3, line 15: 'The Gaussian models are unable to capture. . .' While this may be generally true, a well formulated Gaussian plume model can describe idealised urban dispersion (e.g. Huq and Franzese, BLM, 147, 102-121, 2013).

11) Section 5: Was the CFD model validated using the MUST data for its ability to simulate the measured concentrations?

12) Source position was calculated. Does it include the source height too? Was source height a free parameter or a fixed one?

Other comments

13) Page 2, line 14: What is 'an NP-hard problem'?

14) Page 2, line 35: 'probabilities' should be 'probability'.

15) Page 3, line 8: 'required' should be 'require'.

16) Page 3, line 10: 'the continuous' should be 'continuous'.

17) Page 3, line 23: 'was' should be 'is'.

18) Is the optimisation methodology presented only valid for a single source?

19) Page 7, line 3: The term temperature should be put in quotes as this is not a real temperature in the present context.

20) Page 9, line 2: 'stopped' should be 'is stopped'.

21) Figures 1 and 3: Why some of the 40 samplers locations do not coincide in these figures?

22) Is the code for simulated annealing algorithm with the renormalization inversion technique available?

---

## Short Comment (SC4) · 4 Apr 2018

The manuscript highlights an interesting and challenging problem related to the optimization of sensor networks in the context of a point source reconstruction. In general, the optimization of monitoring network consists of two important issues: (i) reducing the number of receptors and (ii) finding an optimal design of the arrangement of the monitoring network. Here, the study deals only with selecting a reduced set of number of receptors among an already established monitoring network, which is very limited form of a real problem.

The authors have already published the inversion methodology and simulated annealing algorithm (SA) with its application to wind tunnel experiment in Kuichi et al. (2016).

The present study shows a similar application in an urban terrain field experiment by using a CFD model. It does not involve any new development in the model or inversion / optimization algorithm. The presentation of the results is classically similar to a point source reconstruction study which do not highlight any significant contribution related to optimal networking. The application of renormalized inversion and SA methodology in optimizing receptors are associated with several issues which are not clarified in the text. Besides, there are several examples of uncleared and overstated sentences, misinterpretation of mathematics, poor description of results and methodology. Overall, it needs to be justified that what is the significant outcome of this study and how their approach of determining optimal network, which is biased towards measurements, is justified in a general framework?.

Following are the major comments related to the manuscript:

General Comments:

1. In network optimization problem, finding an optimal rearrangement of a set of receptors and then, their evaluation for source estimation are two independent set of problems. The determination of optimal rearrangement should be performed independent of knowledge of measurements and it must contain available maximum information in the domain regarding observability of emissions. The second problem regarding evaluation of source retrieval should be carried out as a next step to validate the efficiency of such networks in the presence of random model-measurement errors. In this study, the two set of problems are mixed and arrangement of network is determined given the knowledge of measurements which is a biased choice of receptors. In addition, the study does not discuss any criterion which could quantify the information in a particular design or impact of model-measurement errors on the chosen network.

2. Through out the text, authors have mentioned the keyword "optimal network". A big question here, is how to prove that a particular design is optimal?. This requires rather mathematical or statistical arguments / proofs to support the fact that a design
is optimal. This can not be shown by showing source retrieval which is nothing but just the estimation of 3 parameters (location, (x, y) and strength q).

3. A big limitation of this approach is the subjectivity and biasness in the methodology and their dependence on the measurements. The optimality of the reduced set of receptors is shown based on the accuracy of the point source retrieval which is not relevant. The good retrieval results with presumed lesser number of receptors are not surprising since their chosen cost function depends on the measurement's values which always force the SA algorithm to choose the receptor locations with good model-measurement accuracy. They completely ignored the fact that their network design criterion should be independent and prior to the knowledge of measurements, which is one of the big limitations.

To be precise, the optimization methodology utilizes the same cost function for both the tasks: (i) Identifying a reduced set of receptors and (ii) retrieving the point source parameters. The cost function utilizes the actual measurements and measures the deviations between measured and predicted concentrations at the chosen set of receptors. The iterative SA algorithm tries to minimize this cost function, which means it selects the receptors with good model-measurement accuracy, i.e. which are closer to the measurements. This will eventually results in good retrieval depending on the model error. This clarifies the fact that the choice of receptors is always subjective to the model-measurement accuracy and will vary in case of perturbation in the model-measurement variables. Thus, this is a poor approach and always biased towards model-measurement accuracy which do not signify the objective of optimization of receptors. The optimization of receptors should have performed independent and prior to the knowledge of measurements, which is not done here.

4. The study do not provide any insights / discussion on systems observability while remaining ill-posed, quantification of information gain or loss during optimization of the network, statistical or mathematical criterion leading to network optimization and sensitivity of the network design with respect to the perturbation in the model-measurement

variables. Also, the study do not mention any optimality criterion, design of experiment or information theory criterion.

5. Another issue with the methodology is that the SA algorithm may not converge always to the same set of reduced receptors. More often, there is high probability that the reduced set will change in every repeated simulation since the number of possible combinations are really high. In this case, how do you guarantee the optimality of design?. Also, the authors never compared between various such different sets corresponding to same trials as how they are varying or what are the differentiation between them. It seems that the authors just choose the arrangement with least reconstruction error which is not logical.

6. If the objective was simply to have a reduced set of network where model-measurement errors are minimum (which is done here), why authors just did not select those locations where model predictions are matching with measurements ?. This could be done simply by comparing model predictions and measurements instead of a massive SA computation. Based on the proposed approach, this can not be called an optimization of the monitoring network.

7. The proposed approach also raises questions regarding the efficiency of the network in case of perturbed model-measurement fields/variables. Also, the retrieved parameters are highly sensitive toward the design of their network which raise further questions regarding the efficiency of the chosen network. The optimized choice of network will always be subjective with respect to the wind variability, model, model errors and measurements. In trials, where model does not perform well, the error will always be high, for example see in trials 2. This will raise the issue of failure of their monitoring networks in identifying correctly the emissions in case of large model errors. This is why the arrangement of the receptors vary in all the trials, even when in some trials, the wind conditions are approximately similar.

8. The study also discusses about weights which they, later, referred as visibility func-
tions highlighting prior informations regarding emissions. However, authors never mention "why they could not determine a criterion based on just visibility weight functions"? Which could be far relevant and independent to the measurements.

9. It is not clear why they could not find a common optimal network which could work in all the trials for point source retrieval ?. The original network of 40 receptors was already working in all the trials irrespective of model errors and varying meteorological conditions. It is useless to propose different configurations based on different meteorological conditions since meteorology can never be constant in a real scenario. The different configurations for different trials again highlight subjectivity of their approach. Thus, the study do not bring any significant outcome regarding the optimization of receptors.

10. How do you describe physical features and efficiency or quality of the reduced configuration?. This is never mentioned in the results and discussion. The discussion mainly involves only source retrieval.

11. Why their optimization always results in finding most of the sensors (5-6 detectors in the reduced configuration) close to the source location? It was never explained in the text. Does your optimized choice of receptors depends significantly on the receptors closer to the source location?. If yes, then what is use of optimizing since you will never know the source in accidental scenarios?

12. Why signal to noise ratio is not shown for all the reduced configurations? and it should be compared with the original network?

13. The authors have simply described the errors in retrieving the location and intensity of the source. The responsible reasons behind it were never explained?. This shows that authors are just interpreting the retrieval rather than really analyzing the results.

14. What is the role of weight function in your reduced configuration?. Does it have any effect on the systems observability and how it do affect your retrieval?

15. Why did you describe vector s on the figures of the source retrieval?. While it seems that you are retrieving source parameters in a weighted least-squares framework?. It was never explained in the results that what is the impact of reducing the receptors on the retrieved general source vector?.

16. A general choice of using a weight matrix in a least-squares methodology is measurement error covariance matrix. Authors did not justify how could they utilize matrix Hw as an alternative to measurement covariance matrix?. In addition, the Hw matrix is not a diagonal matrix which means that using Hw as a weight matrix will induce unphysical correlations among receptors which could be false as well. Did you analyze their impact on source retrieval?, If not, then why and how could you use them directly? Perhaps, you could assume an unity matrix. If not, why?

17. Another issue is with the presentation of the methodology. The study begins by posing an under-determined inverse problem of estimating state of emissions while their objective was to optimize a reduced set configuration for point source retrieval which is an overdetermined inverse problem. Why authors did not begin by directly posing the over-determined problem of point source retrieval?. Why they have presented unnecessary details regarding more general inverse problem of estimation emission state if it was not their objective?. The presented details were already published by several researchers in the literature. Further, authors again formulate the point source retrieval problem in a weighted least-squares sense. Why?. Why two different methods were presented for the same problem?.

18. Why do you need to compute a general vector s of state of emissions?. The objective was just to estimate point source parameters which could be estimated with the weighted / non-weighted least-squares method?. Please clarify?.

19. Again, in the results, figures highlights distribution of weights and vector s which was never related to their monitoring network optimization. Their presentation confuses the overall objective of the study. Do you propose an optimal design for point

source retrieval or a general source retrieval?. The figures related to weights are never explained as why they were needed? or what information do they provide related to the monitoring arrangement?. The given explanation is just copy-paste from previous papers of the authors.

20. Why authors did not compare the weights in comparison to their weights corresponding to the original receptors?.

21. Another issue is also related to the common base network among the 10 and 13 sensors network. Why their strong base network involves only 7 receptors? In general, the 10 sensors networks should be a subset of the 13 sensor network, if not then why?. Please clarify?. It is also surprising that in some trials the common base network involves only 3 sensors. This is unusual having so much variation in having common base sensors among 10 and 13 sensor network in trials. The authors should provide the reasoning behind?, not just mentioning the results.

Specific comments:

1. Abstract, Page 1, line 7. The sentence "The optimal networks in the MUST urban regions enabled ...". Rephrase the sentence. How could an optimal network enable a reduction?. I would like to mention again that the reduced set of receptors were never proved optimal.

2. Abstract, Page 1, line 11. The sentence "This study presents first application of the renormalization data assimilation approach for the optimal network design ….." is overstated and wrong. I could not find where and how did you apply renormalize data assimilation for optimal network design. Renormalize assimilation is only for retrieving the source parameters. I do not see in the text, how it could retrieve the reduced set of receptors. Also, you have interpreted a least-squares framework without justifying their inherent equivalence and choice of parameters with respect to the renormalization. Why?.

3. Page 1, line 18, the sentence "However, pre-deployment of these limited number of sensors . . ...". Please clarify, how could a pre-deployment of sensors helps to achieve maximum information from set of noisy concentration measurements. The objective of pre-deployment of sensors is to have maximum a priori information regarding state of emission and to correctly capture the data while extracting and utilizing information from the data is the final task of data fusion techniques.

4. Page 2, line 2. The sentence "detection of an unknown continuous point source's parameters ..." is wrong. How could you detect point source parameters? They are rather retrieved or estimated.

5. Page 2, line 12. See the sentence "The establishment of an optimal network requires...". Why do you think that for an optimal network it requires availability of concentration measurements?. Please justify? Measurements may be required for the evaluation or validation but for establishment a network can be made with the meteorology and dispersion model.

6. Page 2, line 20, See the sentence "This approach includes the geometric and flow complexity inherent . . ...". I do not think if there is any inverse approach which includes such things for optimization process. The flow variables are accounted through the model, perhaps in the inverse approach in the form of sensitivities which is also derived from adjoint model. All the STE approach can include such information from model.

7. Page 2, line 26. What is "regularized norm square". I do not think Sharan et al., 2012 have included such terms.

8. Page 3, lines 3-5. Issartel, 2005, Sharan et al., 2009, 2012 and Kumar et al., 2015b, do not discuss any iterative algorithm to minimize the difference between observed and modeled concentration.

9. Page 3, line 8, "does not required". Please correct the sentence.

10. Page 3, lines 16-17, Kumar et al. (2015b, 2016) do not provide any extension to

renormalized inversion. It was just an application with CFD model.

11.  Page 3, lines 22-29.  The authors defined the objective to determine optimal network but never achieved.  In line 23, "A methodology was proposed .......".  If the objective was to better characterize the source, why one need to reduce the number of receptors.  The reduction of receptors simply refers to the reduction of information regarding the observability of state of emissions.  In line 26, "For this purpose ...., but with comparable information".  Where do you show in the manuscript that the reduced information is comparable to the original network.  In line 27, "This work explores with two requirements ......", What does it mean, there is no movement of sensors.  You have performed only selection of sensors.

12.  Page 4, line 2, please correct "concentrations measurements".

13.  Page 4, line 4, please correct "an horizontal plane".

14.  Page 4, lines 8-10, The sentence "This study deals with linear relationship, as except from the nonlinear chemical reactions ....." is wrong. Most of these process are nonlinear in your case due to complex flow structures.

15.  Page 4, line 26, citing Kumar et al., 2016 is not appropriate. Please cite appropriate reference.

16.  Page 4, lines 28-30, I understand that the inverse solution from Eq. (2) will lead to peaks at the grid cells coinciding with the measurement cells. This is obvious since the sensitivity matrix has peaks at the measurement cells which are reflected in the inverse solution. However, the difficult part to understand is "why do you call these large values of sensitivities at measurement cells an artificial information? Or a virtual/unphysical reality?". The peak at measurement cells is obvious since the concentration is always maximum at source location and in adjoint computations, you have replaced your measurement cells as source. Please clarify.

17.  Page 5, line 5, Why do you think that normalizing peaks in the sensitivity matrix

with weights will cure the peak problem. I mean, even if you normalize peaks (infinitely large value) by some nonzero weight values, it will not change anything.

18. Page 5, line 9, citing Kumar et al., 2016 is not appropriate. Please cite appropriate reference.

19. Page 6, line 3, "but with comparable information". It is never done in the study.

20. Page 6, line 4, "delivers maximum of the information". It is never shown in the study.

21. Page 6, line 11, why cost function is defined according to Hw norm. Please clarify?.

22. Page 6, line 12. Check sentence "As the cost function is convex, its minimum value must correspond to the maximum intensity of the source". WHY?. Please clarify?. The intensity of source here is q. Check your mathematical expressions. You will obtain an estimate of q for each location vector x. In such case, the maximum value of q in the domain may go to infinity in case where weights w are very small or zero. It is not necessary that the maximum value of q will occur at the minimum of the cost function or at the source location.

23. Page 6, line 15. This is not clear how you can utilize matrix $H\_w$ in place of measurement error covariance matrix in Eq. (9). This is not obvious. Justify?.

24. Page 6, line 16, why two notations q and q0 (section 2.4) are utilized for the same representation?

25. Page 6, line 19, "conditions of maximum intensity….". It seems that authors have difficulties in understanding mathematics. Why equating first order derivative to zero will give maximum intensity?

26. Page 6, line 25, The expression for equation (13) is wrong. Please correct it.

27. Page 6, line 26. The mathematical expression is wrong.

28. Page 6, line 27. How do you guarantee the global minimum in SA algorithm?.

29. Page 7, line 16, "average of the difference of the cost functions calculated for a large number of cases". How did you compute it?

30. Page 7, line 19, "An equilibrium is reached …..". This means that the SA algorithm stops when cost function becomes constant. Then how do you prove global minimum?

31. In combinatorial optimization problem, it is not necessary that SA will provide the same solution or same set of receptors as the converged solution. In this case, how did you choose the solution?.

32. Page 8, step 2 and step 3 show that the choice of receptors depends on the measurements. Thus, there is no optimality in network. The authors have simply chosen the reduced set of receptors based on good retrieval results which is biased.

33. Page 9, lines 8-11. These are not clear. What do you mean by near-optimal network?. What are the conditions for "near overall optimum condition".

34. Page 11, line 11, "The network optimization process …..". This shows that the choice of receptors are biased towards the model-measurement accuracy.

35. Why the configuration of networks should vary based on meteorological stability conditions.

36. Page 12, lines 8-9. "These results exhibit that the SA …..". How it prove optimality of monitoring network?

37. Page 12, line 19. "were observed independently of the number of sensors". Why?. Please clarify?.

38. Page 12, line 20, "larger location errors do not ….". Why?. Please clarify?

39. Page 12, line 23, "a large number of sensors are close to the source positions ..". Why so?. Please justify?.

40. Page 12, line 29, "visibility functions have significant levels …..". What does this mean?.

41. Page 13, line 6, "increase in the number of sensors in a network has little influence on the accuracy…." Why? Please clarify?

42. Page 13, lines 8-10, "In some trials, it was also noted …...not necessary benefial." How could you justify this?. In fact, how did you evaluate if information added by a sensor is fruitful or not? If just based on accuracy of source retrieval, then it is illogical? If an information added by a new sensor is not beneficial then why it should increase the location error?. Also, how location error may decrease with decreasing the number of sensors? You can say this just because you are looking at source retrieval estimates. However, reducing the receptors will make the source retrieval unstable and more sensitive to the noise.

43. Page 13, line 17, What do you mean by "… diversity of structures independently of the number of sensors." You computation is based on fixed number of sensors and there is no discussion of diversity. However, in the following explanation, this is not understandable that why you have different number of common networks in different trials?. It seems that optimized choice of receptors is not really optimized. Otherwise, why in some trials (1, 11), only 3 sensors are found as a common base?. Also, even having 7 receptors as common in 10 and 13 sensor arrangement can not be called a strong common base. There is no explanation why all the 10 sensors are not subset of 13 sensor network. If, it was really optimized that it must have been. If not so, then please explain why?

44. Page 13, line 22, "The performances do not systematically converge independently to the size of the networks". What does it mean and why it does not converge?. Further, it is mentioned that in trial 1, 13 sensor network leads better performance than a 10 sensors network and algorithm leading to near global optimum is contained in the 13 sensor network. This really proves that fact that you choice of receptors is biased by

your source retrieval which is not the objective of an optimal network.

45. Page 13, line 27. Why there is no common trend in skeleton network observed in several trials?. There must be at least with similar flow conditions. If not, justify?

46. Page 13, line 29, "optimal networks can satisfy conditions of a near overall optimum (to be minimized)". What are the near overall optimum conditions?. If you are referring "minimization of cost function". This is a wrong approach.

References:

1. H. Kouichi, G. Turbelin, P. Ngae, A. A. Feiz, E. Barbosa & A. Chpoun (2016), Optimization of sensor networks for the estimation of atmospheric pollutants sources, WIT Transactions on Ecology and The Environment, Vol 207, doi:10.2495/AIR160021.

---

## Short Comment (SC5) · 6 Apr 2018

The subject of this paper is challenging and very timely; certainly, we would like to know how to monitor the spread of pollutants in the urban milieu as efficiently and accurately as possible. In order to accomplish this task, the Authors derive a method based on the combination of optimization techniques, inverse tracers transport modelling and Computational Fluid Dynamics.

The subject is very difficult and there are very few papers addressing the problem in a comprehensive manner; for this reason it is justified to consider publication of this study.

My main comments are related to the necessity of introducing some clarifications to

the presentation.

The Authors attempt to analyze two canonical problems: - Identification of the unknown source - Optimization of the measuring network These two problems are mutually exclusive. Furthermore, they have different cost functionals defined on different vector spaces and, consequently, the set of control parameters is not the same for each case. This important distinction is overlooked in the paper and it is advisable to modify the text by precisely defining the functionals and the control variables.

The problem of optimization of the network is solved using the simulated annealing algorithm. The technique has been introduced to the computational physics over sixty years ago in the classic paper: Metropolis, N.; Rosenbluth, A. W.; Rosenbluth, M.; Teller, A. H.; and Teller, E. "Equation of State Calculations by Fast Computing Machines." J. Chem. Phys. 21, 1087-1092, 1953.

Despite that the original formulation is rooted in the basic principles of physics, the reviewed paper, concerned with the network optimization, is missing the physical interpretation of the Simulated Annealing.

The description of the technique can read as follows:

The algorithm of simulated annealing is initiated by starting from an admissible network. At the subsequent steps, the system moves to another feasible network, according to a prescribed probability, or it remains in the current state. It is crucial to explain how this probability is calculated.

The mobility of the random walk depends on a global parameter T which is interpreted as temperature. The initial values of T are large, allowing free exploration of large extents of the state space (this corresponds to the "melted state" in terms of the kinetic theory of matter).

In the subsequent steps, the temperature is lowered allowing the algorithm to reach a local minimum.

[Figure]

The main characteristic of SA is relatively fast convergence but, unfortunately, it is not possible to prove that the minimum of the cost functional is global. There are several others stochastic minimization methods which can be explored; it is possible that they are potentially more applicable in the context of the monitoring of air pollutants.

My recommendation is to revise the paper. The clarification of the dichotomy between source estimation and network optimization is particularly important. It is also advisable that a description of the simulated annealing algorithm is included.

Please see for example http://katrinaeg.com/simulated-annealing.html

The problem of selection of the initial admissible network and the role of stratification should be discussed. It is well known that the flow around and above complicated structures is characterized by a complex topology. After some analysis of the literature, I'm convinced that the solution of the network optimization depends strongly on the flow Froude number.

The relevant information on the flow in the vicinity of a structure is discussed in the literature, please see for example https://link.springer.com/article/10.1007/s10652-016-9470-3 It would be interesting to present some figures describing both wind and potential temperature fields from the CFD model used in the study.

---

## Short Comment (SC6) · 8 May 2018

**Response to the Reviewer #2**

In this paper, the authors formulate a simulated annealing algorithm with a renormalization inversion algorithm coupled to a CDF flow and dispersion model and apply it to the Mock Urban Setting Test (MUST) tracer field experiment (which simulates an 'urban-like' environment). The aim of the work is to demonstrate how the inversion technique presented can be useful in optimally placing a smaller number of concentration samplers for quantifying a continuous point source with almost the same level of source detection ability as the original larger number of samplers. The paper is well written, but in my opinion requires a major revision. My comments are as follows:

*The authors are grateful to the reviewer's remarks and thanking him/her for reviewing the manuscript. In light of the reviewer's suggestions, the manuscript is revised. Reviewer's questions and remarks are repeated below (in red color) and our responses follow each question (in italic).*

**Main comments**

**1)** The MUST experiments took place under neutral to stable and strongly stable conditions. However, the CFD model used is for neutral conditions and does not include the effects of atmospheric stability over the urban area (the only stability effects included are through the specification of inflow boundary conditions). Atmospheric stability has a profound impact on dispersion and would thus influence the adjoint functions. The authors should discuss the consequences of its neglect on the results and the errors it introduces.

**Reply**: *We agree with the reviewer's remark that the atmospheric stability effects in the CFD model fluidyn-PANACHE were included through the inflow boundary conditions. The used version of fluidyn-PANACHE was not capable of incorporating atmospheric stratification through surface coolingor heating and whatever stability effects are included through the inflow boundary conditions. The fluidyn-PANACHE includes a Planetary Boundary Layer (PBL) model that serves as the interface between the meteorological observations and the boundary conditions required by the CFD solver. The PBL model is composed of two parts: (i) a micro-meteorological model that computes fundamental physical characteristics of the PBL from routine meteorological observations, and (ii) a boundary layer model for prescribing the vertical profiles of wind speed, temperature, and turbulence. However, as discussed in our previous study(Kumar et al., 2015a), even with specification of the stability dependent inflow boundary conditions only, the predicting concentrations from the CFD model are in goodagreement with the measured concentrationsin the MUST experimentfor all 20 trials in different atmospheric stability conditions. This may be due to that the scale and the urban geometry of the MUST field experiment are not large enoughfor the requirement to resolve theatmospheric stratification through surface coolingor heating.And the stability effects included through the inflow boundary conditions were enough to include the stability effects on the concentrations and adjoint functions at such a small scale urban-like*

*environment of the MUST field experiment.However, at microscales also, small irregularities can break the repeated flow patterns found in a regular array of containers with identical shape (Qu et al, 2011). In addition, uncertainties associated with the thickness and the properties of the material of the container wall also affect flow pattern and the resulted concentrations and adjoint functions (Qu et al, 2011). Also, for a real urban environment at the larger scales, the atmospheric stability will have a profound impact on dispersion and would thus influence the adjoint functions. Andthe stability effects through the specification of inflow boundary conditionsonly may not be appropriate for those environments. In these scenarios, the CFD model should be capable of incorporating the atmospheric stratification through surface coolingor heating in real urban environments.*

*A brief discussion about these is now included in the revised manuscript.*

**2)** I have reservations about the usefulness of the methodology presented in real-world urbanenvironments. The title of the paper states 'urban monitoring network' but there are no real urban configurations used. The MUST experimental domain was only 200 m x 200 m (with buildings represented by a grid of containers) which cannot quite represent an urban area in terms of scale, meteorological variability, or non-uniform terrain or roughness/canopy structure. So in a way the present study does not explore any aspects that are specific to urban environments. The authors should discuss this, particularly how their methodology could be applied and its limitations in real-world urban cases. Following on, the title of the paper should say 'urban-like' or something similar instead of 'urban'.

**Reply**: *We agree with the referee's remarksthat the Mock Urban Setting Test (MUST) tracer field experiment was performed in an urban-like environment and cannot quite represent an urban area in terms of scale, meteorological variability, or non-uniform terrain or roughness/canopy structure. However, the MUST field experiment has been widely utilized for the validation of the atmospheric dispersion models in urban-like environment. Although, as mentioned by the reviewer also, there are several limitations to utilize this experiment; but, the methodology presented here is general in nature to apply for a real urban environment also. The methodology involves the utilization of the CFD model which generally can include the effects of the urban geometry, meteorological variability, or non-uniform terrain or roughness/canopy structure in a real urban environment. Therefore the title that will appear on the revised version changed accordingly "Optimization of an Urban-like Monitoring Network for Retrieving an Unknown Point Source Emission". Also, the limitations of the present methodology, for its application in real-world urban cases are discussed in the revised version.*

**3)** There were a total of 40 concentration samplers. In their optimisation, the authors arbitrarily fixed the number of samplers to 13 and 10 and then determined the optimum positions of these reduced number of samplers from the original 40 samplers. A better question to answer would have been "what is the minimum number of samplers required and what their positions are in order to quantify the source with a given degree of confidence or accuracy?"

**Reply**: *The numbers of the sensors were not fixed arbitrary. The trends of location error ($E_l$) and ratio of the estimated to true source intensity ($E_q$) with the number of sensors from 4 to 16 are performed and the results are presented in Kouichi (2017). As already mentioned in the manuscript, the number of sensors in the optimized networks were reduced to 1/3ʳᵈ (13 sensors) and 1/4ᵗʰ (10 sensors) of the total number of sensors (40) originally deployed because for some cases a small number of sensors could not allow to be correctly reconstruct the source and divergences of the calculations have been noted. As an example for Trial 14, reconstructing the source by using a small number on sensors is not appropriate since 4, 5, or 6 sensors are not enough ($E_l > 100$ m and $\log(E_q) > 10E+17$). Also, after a certain number of sensors in the network, the source term estimation is not improved significantly (see Figure 1). Thus, selecting 10 (1/4ᵗʰ) and 13 (1/3ʳᵈ) number of sensors in the optimal networks ensures an acceptable estimate of the source for all the trials. These points are more clearly discussed in the revised manuscript.*

[Figure]

Figure 1: Errors in the estimation of the source (a) position and (b) intensity in Trial 14. Here p is the number of sensors

The present optimisation is based on fixed meteorological conditions in a trial. In a real situation, the network design would also depend on diurnal and spatial variability in meteorological conditions (e.g. wind direction) which may increase or decrease the optimum number of sites. This, however, is not in the scope of the present study. Perhaps as a future study, the authors may consider using data from full scale field measurements such as Salt Lake City Urban 2000 experiment.

**Reply**: *As the problem is complex, in this first study each meteorological situation is assumed as stationary and described by wind speed and direction and stability class. However, we agree with the referee, the network design would also depend on diurnal and spatial variability in meteorological conditions which may increase or decrease the optimum number of sites and also may change the 'best positions' to be instrumented by sensors. Indeed, we envisage as continuity of this work, to study the effect of the variability of the meteorological conditions. As suggested by the reviewer, we would like to utilize and validate the present*

*methodology by using the data from a full scale field measurements such as Salt Lake City Urban 2000 experiment in a future study.*

**4)** Dense gas effects are included. How are they taken into account (or inverted) in the backward (i.e. retro plume) dispersion calculation for adjoint functions?

**Reply**: *Since the released tracer gas $C_3H_6$ in MUST field experiment is heavier than the air, a buoyancy model is used to model the body force term in the Navier-Stokes equations. The buoyancy model is suitable for the dispersion of heavy gases where density difference in the vertical direction drives the body force. Many attempts have been made in literature to use CFD in simulating the dispersion of a negatively buoyant gas using a two-equation k-ε turbulence model (Sklavounos and Rigas, 2004; Tauseef et al., 2011, etc.). The fluidyn-PANACHE implementation of the k-ε model is derived from the standard high-Reynolds number (Re) form with corrections for buoyancy and compressibility (Launder, 2004; Hanjalic, 2005). The k-ε model computes the length and time scales from the local turbulence characteristics. Thus, it can model the turbulent flows subjected to both mechanical shear (obstacles, terrain undulations, canopy) as well as buoyancy (stability and buoyant/heavy gas plumes). For more information, The fluidyn-PANACHE is a three-dimensional (3-D) diagnostic model that solves Reynolds-averaged forms of the Navier-Stokes dynamics equations along with the equations describing conservation of tracer concentration, mass, and energy in the atmosphere (Fluidyn-PANACHE, 2010). As already mentioned in the manuscript, a detailed description of the fluidyn-PANACHE and its evaluation for the forward dispersion with the MUST field experiment was presented in our earlier paper (Kumar et al., 2015).*

**5)** What is the uncertainty in the source estimation results in Table 2? Is the approach capable of providing uncertainty estimates (like the Bayesian one)?

**Reply**: *With the present method, at this moment we cannot derive uncertainties like the Bayesian methods. However,we calculated posterior uncertainties on the source parameters estimation due to the measurements errors. In order to quantify the uncertainty, a 10 % Gaussian noise was added at each measurements. Using the optimal networks 50 simulations for source characterization are performed for each trial. The average and the standard deviation of $E_q$ and $E_l$ are calculated and the result are present in the Table 1 below. For the optimal networks, there is not an obvious trend and the uncertainties are in the same order of magnitude compared to the original network (40 sensors). The Table 2 of the actual version of the manuscript will be replaced by the following table 1 accordingly.*

| Run No. | $E_l^{40}$ (m) | $E_l^{13}$ (m) | $E_l^{10}$ (m) | $E_q^{40}$ | $E_q^{13}$ | $E_q^{10}$ | Skeleton Sensors |
|---|---|---|---|---|---|---|---|
| 1 | 3.3 ±1.3 | 19.6 ±12.13 | 33.76 ±5.30 | 0.92 ±0.08 | 1.04 ±0.23 | 1.24 ±0.22 | 3 |
| 2 | 42.9 ±23.8 | 31.91 ±8.80 | 56.88 ±9.51 | 4.01 ±1.57 | 3.21 ±0.41 | 5.12 ±3.63 | 4 |
| 3 | 10.8 ±1.6 | 9.01 ±2.47 | 9.01 ±3.02 | 1.17 ±0.27 | 0.71 ±0.16 | 0.71 ±0.16 | 7 |
| 4 | 22.8 ±7.7 | 18.07 ±1.84 | 18.07 ±2.61 | 0.27 ±0.35 | 0.83 ±0.21 | 0.83 ±0.26 | 6 |
| 5 | 21.9 ±2.1 | 2.13 ±2.54 | 11.56 ±4.21 | 0.57 ±0.07 | 0.95 ±0.05 | 0.67 ±0.05 | 6 |
| 6 | 5.0 ±1.6 | 6.96 ±0.19 | 6.96 ±0.00 | 2.14 ±0.60 | 1.04 ±0.06 | 1.04 ±0.04 | 7 |
| 7 | 12.4 ±9.1 | 18.85 ±9.08 | 12.99 ±1.67 | 0.41 ±0.49 | 3.11 ±0.51 | 1.06 ±0.07 | 4 |
| 8 | 15.8 ±12.1 | 12.86 ±1.28 | 15.79 ±1.05 | 2.22 ±0.90 | 1.32 ±0.34 | 1.76 ±0.11 | 6 |
| 9 | 7.7 ±1.2 | 8.20 ±0.35 | 8.08 ±0.00 | 1.37 ±0.07 | 3.06 ±0.17 | 7.55 ±0.39 | 5 |
| 10 | 8.8 ±3.0 | 8.00 ±4.57 | 8.00 ±5.68 | 1.08 ±0.19 | 1.08 ±0.77 | 1.08 ±1.07 | 8 |
| 11 | 19.8 ±5.0 | 17.19 ±12.00 | 17.19 ±7.06 | 1.67 ±0.12 | 1.62 ±0.40 | 1.62 ±0.26 | 3 |
| 12 | 7.4 ±6.6 | 5.43 ±11.69 | 10.22 ±9.10 | 0.95 ±0.06 | 0.85 ±0.28 | 0.2 ±0.04 | 4 |
| 13 | 7.7 ±0.6 | 8.63 ±4.36 | 8.63 ±3.86 | 0.97 ±0.07 | 0.78 ±0.18 | 0.78 ± 2.05 | 4 |
| 14 | 2.2 ±1.9 | 5.50 ±2.98 | 5.50 ±3.88 | 1.42 ±0.17 | 0.88 ±0.24 | 0.88 ±0.40 | 7 |
| 15 | 1.1 ±1.0 | 30.23 ±2.14 | 37.98 ±0.72 | 1.88 ±0.09 | 0.57 ±0.07 | 0.17 ±0.01 | 7 |
| 16 | 26.7 ±4.9 | 63.04 ±6.84 | 29.80 ±9.86 | 1.70 ±0.06 | 0.29 ±0.06 | 0.67 ±0.23 | 5 |
| 17 | 7.0 ±1.9 | 14.07 ±2.78 | 23.05 ±10.44 | 0.90 ±0.05 | 1.10 ±0.04 | 1.52 ±0.16 | 6 |
| 18 | 14.3 ±11.0 | 12.83 ±4.18 | 12.83 ±4.61 | 1.15 ±0.46 | 1.15 ±0.16 | 1.15 ±0.21 | 6 |
| 19 | 22.3 ±6.4 | 10.77 ±4.25 | 13.46 ±4.8 | 1.76 ±0.16 | 0.99 ±0.20 | 0.83 ±0.25 | 6 |
| 20 | 32.5 ±1.8 | 45.23 ±1.78 | 44.29 ±0.31 | 0.83 ±0.04 | 1.68 ±0.06 | 1.56 ±0.06 | 7 |

Table1. Source estimation results from the different monitoring networks for each selected trial of the MUST field experiment

**6)** How does the uncertainty in the results in Table 2 change as the number of samplers is changed? Have you included model and measurement uncertainties in the methodology?

**Reply**: *A general relationship between the number of samplers and the uncertainties is not obvious. We noticed that changing the size of network (increasing or decreasing the number of sensors) can lead to the growth or diminution of the uncertainties in the source parameters estimation. As example in Table 1, for Trial#7 uncertainties grow while for Trial#17 uncertainties diminish.*

*Accordingly to the answers of questions 5 and 6, results and interpretations of the effect of measurements errors on the source parameters estimation are included in the revised text.*

**7)** Section 2.3: Is there a sensitivity of the source estimation / optimisation to how the weight function is selected? Could there be any other choices of the weight function?

**Reply**: *The weight function is selected to minimize the information retrieved from the observations thus avoiding inversion artefacts close to the detectors positions. This optimal renormalizing function denoted $\phi(x)$ is unique as demonstrated by Issartel (2004). However, the sensitivity of the source estimation is essentially due to the information provided by each vector of measurements.*

**8)** Did you specify any a priori bounds on the estimated source position and source emission rate? If yes, what were they?

**Reply:** *In this study, we do not require to specify any a priori bounds on the estimated source position and source emission rate in the renormalization inversion technique.*

**9)** What is the advantage of the present technique compared to, say, the Bayesian approach which also provides probability associated with the solution?

**Reply**: *The technique used in this study does not require a priori information about the source (i.e. location and intensity) or about the measurements (i.e. knowledge of the observation-Error Covariance Matrices). The renormalization is an operational method compatible with upstream offline preparation for network implementation and compatible with rapid implementation for the monitoring operation phase for local-scale applications around sensitive sites. Also this method can be used to estimate a point or distributed source which can expand the cases studied.*

**10)** Page 3, line 15: 'The Gaussian models are unable to capture. . .' While this may be generally true, a well formulated Gaussian plume model can describe idealised urban dispersion (e.g. Huq and Franzese, BLM, 147, 102-121, 2013).

**Reply**: *Corrected accordingly as 'In general, the Gaussian models are unable to capture. . .'*

**11)** Section 5: Was the CFD model validated using the MUST data for its ability to simulate the measured concentrations?

**Reply**: *As already mentioned in the manuscript, the ability of the CFD model to simulate the measured concentrations using the MUST data and the prediction errors of the forward simulations used in this study were discussed in our previous study (Kumar et al., 2015).*

**12)** Source position was calculated. Does it include the source height too? Was source height a free parameter or a fixed one?

**Reply**: *The source height was not calculated in this study. The computations were carried out in 2-dimensional domain on a horizontal plane corresponds to an altitude of known source height Hs.Accordingly, the vertical dimension was eliminated in the formulations and the computations. Consequently, the adjoint functions were chosen as steady state retroplumes on the horizontal cross-section area passing through a plane $z = Hs$. The assumptions with respectto the vertical structure of the problem is useful to estimate the ground level sources or the emission sourcesalong a horizontal cross-section area passing through a fixed vertical level. However, in this study, the problem of vertical structure (i.e. height of a source) in three-dimensional space of an urban area is not addressed. In reality, an altitude of a release (i.e. source height) is also not known and required to estimate along with the projected release location on the ground surface and the release rate (Kumar et al., 2016).We envisage to include the height of the source in a future study.*

**Other comments**

13) Page 2, line 14: What is 'an NP-hard problem'?

**Reply**: *The problem of sensors network optimisationis NP-Hard (i.e. Non-deterministic Polynomial-time hardness) as shown by (Ko et al., 1995), which means that it is difficult for an exhaustive search algorithm to solve all instances of the problem because it's need a considerable time.*

14) Page 2, line 35: 'probabilities' should be 'probability'.

**Reply**: *Corrected. We have now carefully revised the manuscript to eliminate possible linguistic errors.*

15) Page 3, line 8: 'required' should be 'require'.

**Reply**: *Corrected.*

16) Page 3, line 10: 'the continuous' should be 'continuous'.

**Reply**: *Corrected.*

17) Page 3, line 23: 'was' should be 'is'.

**Reply**: *Corrected.*

18) Is the optimisation methodology presented only valid for a single source?

**Reply**: *In this study, the presented optimization methodology is only valid for a single source. Nevertheless, it is possible to consider the optimization for multiple sources. We envisage that evaluation in the future.*

19) Page 7, line 3: The term temperature should be put in quotes as this is not a real temperature in the present context.

**Reply**: *Changed accordingly.*

20) Page 9, line 2: 'stopped' should be 'is stopped'.

**Reply**: *Corrected.*

21) Figures 1 and 3: Why some of the 40 samplers locations do not coincide in these figures?

**Reply:** *In the schematization of the MUST experiment, the position of the tenth detector of the fourth row was slightly shifted from its true position. Figure 1 in actual manuscript version is adjusted accordingly as shown in figure 2 below.*

[Figure]

Figure 2: Schematization of the MUST experiment: (a) correct version (b) adjusted version

22) Is the code for simulated annealing algorithm with the renormalization inversion technique available?

**Reply:** Yes *the codes are available for one trial as example.*

**References**

*Fluidyn-PANACHE, 2010. User Manual. FLUIDYN France / TRANSOFT International, version 4.0.7 Edition.*

*Hanjalic, K., 2005. Turbulence and transport phenomena: Modelling and simulation. In: Turbulence Modeling and Simulation (TMS) Workshop. Technische Universitat Darmstadt.*

*Issartel, J.-P. Emergence of a tracer source from air concentration measurements, a new strategy for linear assimilation. Atmospheric Chem. Phys. 5, 249–273 (2005). URL* http://www.*atmos-chem-phys.net/5/249/2005/. DOI 10.5194/acp-5-249-2005.*

*Ko, C.-W., Lee, J., Queyranne, M., 1995 An exact algorithm for maximum entropy sampling. Operational Research, 43, 684–691. URL https://doi.org/10.1287/opre.43.4.684.*

*Kumar, P., Feiz, A.-A., Ngae, P., Singh, S. K., Issartel, J.-P., 2015. CFD simulation of short-range plume dispersion from a point release in an urban like environment. Atmospheric Environment 122, 645 – 656. URL http://www.sciencedirect.com/science/article/pii/S1352231015304465*

*Kumar, P., Singh, S. K., Feiz, A.-A., Ngae, P., 2016. An urban scale inverse modelling for retrieving unknown elevated emissions with building-resolving simulations. Atmospheric environment, 140, 135-146. URL https://doi.org/10.1016/j.atmosenv.2016.05.050*

*Launder, B., 2004. Turbulence modelling of buoyancy-affected flows. In: Singapore Turbulence Colloquium.*

*Qu, Y., Milliez, M., Musson-Genon, L., &Carissimo, B. (2011). Micrometeorological modeling of radiative and convective effects with a building-resolving code. Journal of applied meteorology and climatology, 50(8), 1713-1724.*

*Sklavounos, S., Rigas, F., 2004. Validation of turbulence models in heavy gas dispersion over obstacles. Journal of Hazardous Materials 108 (1), 9 – 20. URL http://www.sciencedirect.com/science/article/pii/S0304389404000494*

*Tauseef, S., Rashtchian, D., Abbasi, S., 2011. CFD-based simulation of dense gas dispersion in presence of obstacles. Journal of Loss Prevention in the Process Industries 24 (4), 371 – 376. URL http://www.sciencedirect.com/science/article/pii/S0950423011000222*

---

## Short Comment (SC7) · 12 May 2018

**Response to the Reviewer #1**

The subject of this paper is challenging and very timely; certainly, we would like to know how to monitor the spread of pollutants in the urban milieu as efficiently and accurately as possible. In order to accomplish this task, the Authors derive a method based on the combination of optimization techniques, inverse tracers transport modelling and Computational Fluid Dynamics. The subject is very difficult and there are very few papers addressing the problem in a comprehensive manner; for this reason it is justified to consider publication of this study.

*We would like to thank Pr. Pudykiewicz for his careful consideration of this manuscript and for his helpful and insightful comments. We have carefully considered his comments and worked to include them in the revised version of the manuscript according to the proposed suggestions.*

*Please find below the responses to his comments.*

1) 'The Authors attempt to analyze two canonical problems: - Identification of the unknown source - Optimization of the measuring network. These two problems are mutually exclusive. Furthermore, they have different cost functionals defined on different vector spaces and, consequently, the set of control parameters is not the same for each case. This important distinction is overlooked in the paper and it is advisable to modify the text by precisely defining the functionals and the control variables.'

**Reply**: *In this study two canonical problems are considered:*

*a) Identification of the unknown source: the source term estimation STE is studied in the framework of a parametric approach using the renormalization technic. Here the challenge is to determine the parameters of the source (intensity and position) using any measurements vector (in practice the number of measurements is limited). Based on retroplumes (using sensors locations and CFD model in backward mode), we first determine the optimal renormalized Gram matrix $H_w$, for which an optimal weight function is required. This optimal weight function that verify the renormalization conditions minimize the information retrieved from the observations thus avoiding inversion artefacts close to the detectors positions. As the renormalization is a data assimilation method, the cost function to minimize is defined as the quadratic distance between the observed and the simulated measurements according to the $H_w$ norm.*

*b) Optimization of the measuring network: here, the optimization consists of selecting the best positions to be instrumented by the sensors among potential locations. This choice is operated in a space of search constituted of all possible networks (of a specific size) and based on a cost function that describe quantitatively the quality of the networks. The optimal network has the lowest quadratic distance between the observed and the simulated measurements according to the $H_w$ norm. This optimal (or near-optimal) network is obtained using the*

*Simulated Annealing (SA) algorithm. The data here are the measurements and the according sensors locations.*

*These canonical problems are coupled at each iteration during the searching process.*

*The text of the revised manuscript is accordingly modified to clarify this important point.*

2) 'The problem of optimization of the network is solved using the simulated annealing algorithm. The technique has been introduced to the computational physics over sixty years ago in the classic paper: Metropolis, N.; Rosenbluth, A. W.; Rosenbluth, M.; Teller, A. H.; and Teller, E. "Equation of State Calculations by Fast Computing Machines." J. Chem. Phys. 21, 1087-1092, 1953. Despite that the original formulation is rooted in the basic principles of physics, the reviewed paper, concerned with the network optimization, is missing the physical interpretation of the Simulated Annealing. The description of the technique can read as follows: The algorithm of simulated annealing is initiated by starting from an admissible network. At the subsequent steps, the system moves to another feasible network, according to a prescribed probability, or it remains in the current state. It is crucial to explain how this probability is calculated. The mobility of the random walk depends on a global parameter T which is interpreted as temperature. The initial values of T are large, allowing free exploration of large extents of the state space (this corresponds to the "melted state" in terms of the kinetic theory of matter). In the subsequent steps, the temperature is lowered allowing the algorithm to reach a local minimum.'

**Reply**: *The SA algorithm is a random optimization technique based on an analogy with thermodynamics. For the SA, each network of p sensors is considered as a state of a virtual physical system, and the objective function is interpreted as the internal energy of this system in a given state. According to statistical thermodynamics the probability of a physical system to being in a state β follows the Boltzmann distribution $P_\beta = \frac{1}{Z} exp(\frac{-\Delta E_\beta}{K_B T})$ , where Z is the partition function, $E_\beta$ is the internal energy, T is the temperature at the state β and $K_B$ is the constant of Boltzmann. By analogy, the physical quantities (temperature, energy, etc.) become pseudo-quantities and during the minimization process, the probabilistic treatment consists to accept a new network selected in the neighborhood of the current network following the probability $P = exp(\frac{-\Delta J}{T})$. Where ΔJ is the cost difference between the new and the current configurations. At high temperature, the SA performs a coarse search of the space of global states, avoids local minima and finds a good minimum. As the temperature is lowered, the search becomes fine in the neighborhood of the already determined minimum and the SA reaches a better minimum.*

*We include this physical interpretation of the Simulated Annealing in the revised manuscript.*

3) The main characteristic of SA is relatively fast convergence but, unfortunately, it is not possible to prove that the minimum of the cost functional is global. There are several others stochastic minimization methods which can be explored; it is possible that they are potentially *more applicable in the context of the monitoring of air pollutants.*

**Reply***: It is clear that for all the metaheuristic algorithms (such as the SA) it is not possible to prove that the minimum of the cost functional is global. This question is crucial for us, for this reason we plan in the future to study the degree of confidence on the 'optimal networks'. Nevertheless, before retaining SA as an optimization technique we tested and compared the results obtained by Genetic Algorithm (GA) and SA based on the normalized error cost function (Kouichi, 2017).These algorithms of different search technics (SA probabilistic & GA evolutionary) are evaluated for the reconstruction of a source in a wind tunnel (DYCE experiment (Lepley et al., 2011). The optimization consisted in selecting the best positions for sensors implantation among 27 potential positions scattered in the Wind Tunnel. The results show that the optimal networks retained by the GA and the SA are quantitatively (figure 1) and qualitatively (figure 2) comparable. The errors in source parameters estimation by using the optimal networks of 3 to 13 sensors are presented in figure 1 below. The SA has advantageous because it is relatively easy to implement and takes smaller computational time in comparison to GA. Both SA and GA optimization algorithms in the framework of our approach (based in the renormalization theory) has little influence on estimation of the parameters of a source.*

[Figure]

Figure 1: Error of source parameters estimation for (a) SA (b) GA in the DYCE wind tunnel experiment. Here m is the number of sensors, $E_l$ and $E_q$ respectively denote the location error (m) and the ratio of the estimated to true source intensity

[Figure]

Figure 2: Optimal networks (m = 3, 6, 9 and 12) obtained by (a) Simulated Annealing SA and (b) Genetic Algorithm GA

4) The problem of selection of the initial admissible network and the role of stratification should be discussed. It is well known that the flow around and above complicated structures is characterized by a complex topology. After some analysis of the literature, I'm convinced that the solution of the network optimization depends strongly on the flow Froude number. The relevant information on the flow in the vicinity of a structure is discussed in the literature, please see for example https://link.springer.com/article/10.1007/s10652-016- 9470-3 It would be interesting to present some figures describing both wind and potential temperature fields from the CFD model used in the study.

**Reply**: *The initial admissible network is selected following the trends of location error ($E_l$) and ratio of the estimated to true source intensity ($E_q$) with the number of sensors from 4 to 16 are performed and the results are presented in Kouichi (2017). As already mentioned in the manuscript, the number of sensors in the optimized networks were reduced to $1/3^{rd}$ (13 sensors) and $1/4^{th}$ (10 sensors) of the total number of sensors (40) originally deployed because for some cases a small number of sensors could not allow to be correctly reconstruct the source and divergences of the calculations have been noted. As an example for Trial 14, reconstructing the source by using a small number on sensors is not appropriate since 4, 5, or 6 sensors are not enough ($E_l > 100$ m and $log(E_q) > 10E+17$). Also, after a certain number of sensors in the network, the source term estimation is not improved significantly (see Figure 3). Thus, selecting 10 ($1/4^{th}$) and 13 ($1/3^{rd}$) number of sensors in the optimal networks ensures an acceptable estimate of the source for all the trials. These points are more clearly discussed in the revised manuscript.*

[Figure]

Figure 3: Errors in the estimation of the source (a) position and (b) intensity in Trial 14. Here p is the number of sensors

*The atmospheric stability effects in the CFD model fluidyn-PANACHE were included through the inflow boundary conditions. We had already analyzed the importance of the proper inflow boundary conditions for wind and turbulence variables on forward and backward atmospheric dispersion in an urban area (Kumar et al., 2015). Accordingly, Gryning et al. (2007) wind profile and an approximate analytical solution of the one-dimensional k-ε prognostic equation (Yang et al., 2009) for the turbulence profiles were used for inflow*

*boundary. Gryning et al. (2007) wind profile is composed of the three different length scales in surface, middle, and upper layers of the atmospheric boundary layer (ABL), and is applicable in the entire ABL. It was also noted that Gryning's wind profile is not applicable in the trials (number 4, 5, and 6 of the MUST field experiment) of very stable atmospheric conditions. Thus, a wind profile based on the similarity function proposed by Beljaars and Holtslag (1991) were used in these trials. Monin-Obukhov similarity theory-based logarithmic temperature profile was used to describe its vertical variation in neutral and stable conditions in the MUST field experiment. Since the coefficients in approximated analytical profiles of k and ε are estimated by fitting the observed values of k, the turbulence profiles follow the actual representation of k in each trial of the MUST experiment (Kumar et al., 2015).*

*More generally, in pollutant dispersion problems, when a proper level of turbulence intensity is important at the upwind side of the obstacles, the commonly used techniques are based on setting up simplified forms of inlet TKE (Santos et al., 2009), dynamical recycling (Tomas et al., 2015) or smooth inflow with generic downwind roughness elements (Tomas et al., 2016). Such conditions mostly affect the intensity of vertical mixing and the rate of boundary layer growth, decisive factors in determining concentrations of pollutants emitted within the urban canopy (Korycki et al., 2016). We think that these effects are beyond the scope of this work and could be further explored for the future. These discussions are now included in the revised manuscript,*

*We also present in the revised manuscript figures describing wind fields from the CFD model for the trials 4, 11 and 19. As example, in the figure 4 below is showed the wind velocity vectors around some containers for the trial 11.*

[Figure]

Figure 4: Wind velocity vectors for the trial 11

**References**

*Beljaars, A., Holtslag, A., 1991. Flux parameterization over land surfaces for atmospheric models. J. Appl. Meteorol., 30 (3), 327-341, http://dx.doi.org/10.1175/1520-0450(1991)030<0327:FPOLSF>2.0.CO;2*

*Gryning, S.-E., Batchvarova, E., Brmmer Jr., B., Gensen, H., Larsen, S., 2007. On theextension of the wind profile over homogeneous terrain beyond the surfaceboundary layer. Bound. Layer Meteorol. 124 (2), 251-268. http://dx.doi.org/10.1007/s10546-007-9166-9*

*Korycki, M., Łobocki, L., Wyszogrodzki, A., 2016. Numerical simulation of stratified flow around a tall building of a complex shape Environ Fluid Mech, 16, 1143–1171, DOI 10.1007/s10652-016-9470-3*

*Kouichi, H., Jul. 2017. Sensors networks optimization for the characterization of atmospheric releases source. Thesis, Université Paris Saclay, France. URL https://hal.archives-ouvertes.fr/tel-01593834*

*Kumar, P., Feiz, A.-A., Ngae, P., Singh, S. K., Issartel, J.-P., 2015. CFD simulation of short-range plume dispersion from a point release in an urban like environment. Atmospheric Environment 122, 645 – 656.
URL http://www.sciencedirect.com/science/article/pii/S1352231015304465*

*Lepley, J. J., Lloyd, D., Robins, A., Wilks, A., Rudd, A., Belcher, S., (2011). Dynamic sensor deployment for the monitoring of chemical releases in urban environments (DYCE), Proc. of SPIE, 8018(12), 1-11, doi:10.1117/12.883373.*

*Santos, J.M., Reis, N.C., Goulart, E.V., Mavroidis, I., 2009. Numerical simulation of flow and dispersionaround an isolated cubical building: the effect of the atmospheric stratification. Atmos Environ43:5484–5492. doi:10.1016/j.atmosenv.2009.07.020*

*Tomas, J.M., Pourquie, M.J.B.M., Jonker, H.J.J., 2015. The influence of an obstacle on flow and pollutantdispersion in neutral and stable boundary layers. Atmos Environ 113:236–246, https://doi.org/10.1016/j.atmosenv.2015.05.016*

*Tomas, J.M., Pourquie, M.J.B.M., Jonker, H.J.J., 2016. Stable stratification effects on flow and pollutant dispersion in boundary layers entering a generic urban environment. Bound Layer Meteorol 159(2):221–239, https://doi.org/10.1007/s10546-015-0124-7*

*Yang, Y., Gu, M., Chen, S., Jin, X., 2009. New inflow boundary conditions formodelling the neutral equilibrium atmospheric boundary layer in computationalwind engineering. J. Wind Eng. Industrial Aerodyn. 97 (2), 88-95.
http://www.sciencedirect.com/science/article/pii/S0167610508001815.*

---

## Short Comment (SC8) · 31 May 2018

**Response to the Short Comment SC4**

*We thank our former colleague Dr. Sarvesh Singh in the LMEE Laboratory with whom we have also published many research papers as a coauthor, for his detailed reading of our paper. We are grateful to him, despite the fact he is perhaps a bit too personally involved, reading and commenting so abundantly our work, pointing out the weakness of some difficult explanations given too quickly. He has a good practice in inverse modeling for the air pollutant source reconstruction in flat and homogeneous terrains. However, we understand that this multidisciplinary study which requires wide knowledge, not only of the inverse problem, but also of the CFD modelling, optimization and experience in engineering, can lead to confusion and misunderstanding. Below we tried to answer (in italic form) to all of his comments (in red color), also many of his comments were repeated several times asking same question again and again.*

**Comment:** The manuscript highlights an interesting and challenging problem related to the optimization of sensor networks in the context of a point source reconstruction. In general, the optimization of monitoring network consists of two important issues: (i) reducing the number of receptors and (ii) finding an optimal design of the arrangement of the monitoring network. Here, the study deals only with selecting a reduced set of number of receptors among an already established monitoring network, which is very limited form of a real problem.

**Reply:** *The optimization of monitoring network doesn't consist in general two important issues: reducing the number of receptors and finding an optimal design of the arrangement of the monitoring network, this affirmation is very simplistic. Fundamentally, the optimization of monitoring networks problems may concern:*

*1. First deployment: This case is the more complex and consists in defining different interesting areas (monitored area, vulnerable area, danger area, the potential locations, etc.) before optimizing the network. This problem cannot be dealt in general, because the studied zones change according to the situation (industrial zone, target of an aggression from the external, etc.). Once the search space is defined (i.e. the candidate locations for the sensors implantation), the problem can concern finding the best configuration of a minimal number of sensors (similar to problem 3 below) or the best spatial arrangement of a predetermined number of detectors (in some cases for example the protection against eventual terrorist attack, **a limited number of sensors is not important** and the security must be guaranteed by the maximum means).*

*2. Updating an existing network: This problem consists in changing the sensors positions in the interesting area for the specific needs (such as an important variation of the meteorological situations after long time the network is designed) without changing the size of the networks (i.e. number of sensors).*

*3. Reducing the size of an existing network: The challenge here is to determine the optimal size of network and the best locations for the sensors implantation. Here the original network is considered as the search space.*

*4. Increasing the size of an existing network: This problem consists in determining the best positions to add to a set of sensors already placed on a site. The number of detectors to add can be prefixed or included in the optimization problem (i.e. must be determine).*

*The detailed problems are independent and each one of them have its own requirements*.

*It is also important to know that the optimization of sensors networks depend on the network type:*

*a. Mobile network deployed only on emergency: Here the detectors are rapidly deployed specifically for collecting the information (i.e. measurements) to be used for a specific need (neutralize the source, refurbishment an installations on industrial site, etc.). In this case, the meteorological conditions (as wind speed and direction, etc.) can be known in real time from the available observations or from numerical weather forecasting models and can be assumed as stationary. The optimization in this case can be performed in real time if the interesting area is not complex and the calculation can be conducted quickly in a very short time (using Gaussian model and an optimization algorithm for example). If the domain is complex (i.e. contains several obstacles), CFD model must be used to include the effect of the obstacles, the optimal locations to be instrumented by the sensors must be determined in upstream off-line.*

*b. Permanent mobile network: Here the vulnerable area is monitored permanently by detectors embarked in mobile systems (such as drones or robots). The optimization in this case consists in determining the best locations following the situation (meteorological conditions, presence of a danger, detection of an accident, etc.).*

*c. Permanent static network: Here the vulnerable area is monitored permanently by a fixed network that must be efficient regardless of the meteorological situations. The optimal design consists in finding the best arrangement of the detectors (the number can be minimal or prefixed).*

***As a conclusion, the optimization of monitoring network in an urban environment is a complex problem that must be deal with proper use of the inverse modeling, CFD modeling, and optimization techniques. The present study deals with the above cases (3) and (a). As also pointed out by Dr. Pudykiewicz, one of the reviewers of this paper and a distinguished scientist in this field, the subject of this study is challenging, very timely, and very difficult and there are very few papers addressing the problem in a comprehensive manner. In order to accomplish this task, we derived a method based on the combination of optimization techniques, inverse tracers transport modeling and Computational Fluid Dynamics. So the remark of Dr. Singh that the study is very limited form of a real problem is not true.***

*For more detail we have cited below some studies that can help to understand the challenges and the methods that can be used in the context of our work:*

- *(Chen et al 2012): Optimization of water quality monitoring network in a large river by combining measurements, a numerical model and matter-element analyses.*
- *(Ainslie et al 2009): Application of an entropy-based Bayesian optimization technique to the redesign of an existing monitoring network for single air pollutants.*
- *(Mofarrah & Husain, 2010): A holistic approach for optimal design of air quality monitoring network expansion in an urban area.*
- *(Lepley et al 2011): Dynamic sensor deployment for the monitoring of chemical releases in urban environment.*

- *(Le et al, 2003): Designing networks for monitoring multivariate environmental fields using data with monotone pattern.*
- *(Jiang et al, 2007): Optimization of mobile radioactivity monitoring networks.*

**Comment:** The authors have already published the inversion methodology and simulated annealing algorithm (SA) with its application to wind tunnel experiment in Kuichi et al. (2016). The present study shows a similar application in an urban terrain field experiment by using a CFD model.

**Reply:** *'' Kuichi et al. (2016)'', do you mean Kouichi et al. (2016)?*

*This comment of Dr. Singh is not completely true and also misleading as in the study of Kouichi et al. (2016), which is in fact a conference paper, the dispersion experiment of the gas and the modeling study were performed in very idealized conditions in a wind tunnel (standard deviation adjusted specifically for the wind tunnel, wind speed fixed according to the best results, measurements far from the boundary layer, dispersion in a homogeneous space without obstacles, etc.). All these aspects simplify the complexity of the optimal design problem and don't lead to fine analysis of the locations importance on the reconstruction of the source parameters. The current study of the networks optimization in view of the source reconstruction in urban domain  is a very complex and challenging problem and the renormalization inversion method has never been the subject of this study, this clearly justifies the originality of this work.  With proper citations of our earlier works, in this study, we never claimed to present the inversion methodology. In fact, the earlier source reconstruction results were presented for comparison purposes only with proper citation of that work. In this study, we derived a methodology for designing the optimal monitoring networks **in an urban like environment** based on the combination of optimization techniques, inverse tracers transport modeling and Computational Fluid Dynamics. Dr. Singh comment that this is just an application in an urban terrain field experiment by using a CFD model, cannot be justified as Dr. Pudykiewicz also pointed out that this study is challenging and very difficult and there are very few papers addressing the problem in a comprehensive manner. In fact, the complexity of the problem increases manifold for urban environments where simple analytical or Gaussian models have limitations and cannot be apply for such complex environments.  This study presents a method for designing the optimal monitoring networks in an urban like environment.*

**Comment:** It does not involve any new development in the model or inversion / optimization algorithm.

 **Reply***: Fundamentally an optimization study does not require development in the used methods or algorithms but it is ensured by three essential phases:*

*1. Choice of an objective functions: (also known as cost functions), in our case this function is the optimality criterion which describes the quality of a network and which is in agreement with the defined problem (mobile or static network, for reconstruction of a source or for other need, etc.).*

*2. The problem statement: this consists in defining how the optimization problem is approached (i.e. discrete or continuous search areas, definitions of spatial zones, etc.).*

*3. The choice of the optimization algorithm: which is generally in coherence with the phase 2 (Determinist optimization, Hard optimization, etc.)*

*The complexity of the optimization problems is in the well definition of these three phases specially the choice of the optimality criterion and also in the implementation and the exploitation of the retained techniques. In this study, we never claimed any development in the inversion or optimizations algorithms as these techniques were already available in the literature. We derived a methodology for designing the optimal monitoring networks in an urban like environment which is based on the combination of optimization techniques, inverse tracers transport modeling and Computational Fluid Dynamics. Which attests to a new development for optimally designing the sensors monitoring network in an urban like environment.*

**Comment:** The presentation of the results is classically similar to a point source reconstruction study which do not highlight any significant contribution related to optimal networking.

**Reply:** *In the present study, the obtained optimal networks were analyzed qualitatively and quantitatively for all the trials of MUST field experiment. The dispersion of the sensors in the urban like environment was critically analyzed according to the source position and the meteorological situation. A fine analysis is performed to highlight the common structures (Skeleton) in the optimal networks. Also, a posteriori study is realized in order to evaluate the performance of the optimal networks. For this the errors in source parameters estimation are compared with the errors obtained from the original network which leads to the important conclusions in networks size reduction in the framework of source reconstruction in an urban environment. As the applicability of the obtained monitoring networks is validated and analyzed by estimating the source parameters from the concentration measurements from the optimal networks, it is obvious to present and analyzed the source reconstruction results and compare these with from previous study. We do not agree with Dr. Singh's point that this study does not highlight any significant contribution related to optimal networking. In fact, using the proposed methodology, we were accurately able to estimate the source parameters using the measurements only from $1/4^{th}$ and $1/3^{rd}$ sensors with approximately similar accuracy compare to the network of original number of sensors. This is significant contribution that reduces the number of sensors in an urban like environment and without compromising the ability of the network with minimal number of sensors to estimate the source.*

**Comment:** The application of renormalized inversion and SA methodology in optimizing receptors are associated with several issues which are not clarified in the text.

**Reply:** *As mentioned in the abstract and in the text the renormalized inversion method, the SA and the CFD model were coupled to obtain the optimized network of the receptors and the related issues were partially presented when the methodology of the networks optimization was presented for an urban like environment. We have more clarified these issues in the updated version of the manuscript.*

**Comment:** Besides, there are several examples of uncleared and overstated sentences, misinterpretation of mathematics, poor description of results and methodology.

**Reply:** *Unfortunately we cannot answer to a not given examples. Can you please indicate the ''uncleared and overstated sentences'' and the ''misinterpretation of mathematics''? However, we will carefully read again and correct the text in updated manuscript for such examples (if any).*

**Comment:** Overall, it needs to be justified that what is the significant outcome of this study and how their approach of determining optimal network, which is biased towards measurements, is justified in a general framework?.

**Reply:** *As clarified before (see the first reply) this study is carried out in the framework of the cases (3) and (a). We do not study all the envisaged cases. It is clear that the problem of sensors network optimization is not trivial and need to be solved according to a given configuration. We remind that in this study we provide answer to the specific operational need for witch it is necessary to deploy sensors in emergency situations where the meteorological conditions are known in real time and some information about the measurements are available. We explain more: for example, in an industrial site the area where an eventual source can be located is known. Consequently, the source is roughly localized and the optimized network is deployed to refine the estimation of the source position which helps to repair the installation. A second concrete case concerns the estimation of the intensity of hazardous accidental release. This data (i.e. intensity of the source) is primordial for following and/or for predicting the evolution of the accidental plume. In such specific case, it is judicious to be based on the scenarios which justify the use of concentrations in optimal design. For more clarification we cite some example of works based on scenarios (i.e. a priori informations are available) where the measurements were used in the optimal design of the monitoring networks:*

- *(Ma et al 2013): Comparison and improvements of optimization methods for gas emission source identification.*
- *(Mason et Bohlin, 1995): Network optimization of a radionuclide monitoring system for the comprehensive nuclear test ban treaty.*
- *(Berry et al, 2006-a): A facility location approach to sensor placement optimization.*
- *(Watson J-P et al, 2004): A multiple-objective analysis of sensor placement optimization in water networks.*
- *(Krause et al, 2008) Efficient Sensor Placement Optimization for Securing Large Water Distribution Networks.*
- *(Hamel et al, 2010): Sensor Placement for Urban Homeland Security Applications.*
- *(Abida et Bocquet, 2009): Targeting of observations for accidental atmospheric release monitoring.*

*Also we can find in the literature some works of sensors networks optimization based on error cost function (error objective function) similar to the optimality criterion that we proposed in our study:*

- *(Abida et al, 2008): Design of a network over France in case of a radiological accidental release.*
- *(Saunier et al, 2009): Model reduction via principal component truncation for the optimal design of atmospheric monitoring networks.*
- *(Jiang et al, 2007): Optimization of mobile radioactivity monitoring networks.*

**This study shows that it is possible to reconstruct a source of atmospheric emissions with a limited number of concentration measurements and presents a methodology for selecting the 'best' sensors positions basing on an optimality criterion and by coupling an optimization algorithm an inversion method and a CFD model that include the complexity of an urban domain. This study presents a practical method for managing realistic situations. In an area of interest it is not possible to place the sensors anywhere. This study present an investigation on the measurements vector used in the inverse problems**.

**General Comments:**

**Comment 1**. In network optimization problem, finding an optimal rearrangement of a set of receptors and then, their evaluation for source estimation are two independent set of problems. The determination of optimal rearrangement should be performed independent of knowledge of measurements and it must contain available maximum information in the domain regarding observability of emissions. The second problem regarding evaluation of source retrieval should be carried out as a next step to validate the efficiency of such networks in the presence of random model-measurement errors. In this study, the two set of problems are mixed and arrangement of network is determined given the knowledge of measurements which is a biased choice of receptors. In addition, the study does not discuss any criterion which could quantify the information in a particular design or impact of model-measurement errors on the chosen network.

**Reply:** *We agree with Dr. Singh's remarks that in a network optimization problem, finding an optimal rearrangement of a set of receptors and then, their evaluation for source estimation are two independent set of problems. In fact, we also followed the same procedure. The network optimization problem was independently presented and performed before any evaluation by estimating the source parameters using the measurements from the sensors from the obtained optimal network. However, since the optimization methodology utilizes some concepts (not the source estimation part) from the renormalization inversion methodology, we presented it after the source estimation methodology. As also explained in the manuscript and more clearly in the updated version, the network optimization problem is completely independent from the source estimation evaluation. It is very clearly explained in the flow diagram in Figure 2 and shows that source was estimated only when we obtained the optimal monitoring network. In this work, the first step consist to find the best configuration of a limited set of sensors using the meteorological data, the sensors positions on the instrumented area, a CFD technique and the concentration observations. The second step consists to evaluate a posteriori the performance of the optimal networks in comparison with the original network used in MUST field experiment.*

*The problem of optimal design of sensors networks for source reconstruction can be performed (i) without a priori information or (ii) with a priori information of the source (such us its position, intensity, etc.) and the observations (i.e. pollutant concentrations) can be used in the optimization process see as examples (Ma et al, 2014).*
*In a case when the source is considered completely unknown (as example in terrorist attack) the challenge is to design a network able to reconstruct an eventual source regardless of its position and intensity. Thus a specific cost function could be defined in order to assure the optimal design and the concept of information can be used. The PhD thesis of Kouichi (2017) and Kouichi et al. (2016) was inspired by this concept for defining the entropic criterion based on the renormalization method in order to estimate the best arrangement of the sensors for source characterization regardless its intensity and position. This work is already a subject of a publication that we realized.*
*In a case when some knowledge about the source are available we remind as example on industrial sites the 'danger zone' were the source can be located is completely known (storage tank of hazardous products, network of pipelines, etc.) for this reason works of optimization can be conducted based on scenarios for witch a priori information on the source was utilized (other example in specific situation of accidental release the position of the source is known (observed in the site) and the need is to estimate its intensity in order to estimate the dispersed quantities of a hazardous agent this can help to predict the evolution*

*of the accidental plume or to determine the contaminated area or to estimate the quantities inhaled by the personals exposed on site, etc.*

*Concentrations measurements can routinely be available from an already existing large monitoring network. In this study we utilized these concentration measurements to reduce the size of a large network. The updated version of the manuscript also discusses a posteriori error analysis of the source reconstruction based on the random measurements errors from an obtained monitoring network.*

**Comment 2**. Throughout the text, authors have mentioned the keyword "optimal network". A big question here, is how to prove that a particular design is optimal?. This requires rather mathematical or statistical arguments / proofs to support the fact that a design is optimal. This can not be shown by showing source retrieval which is nothing but just the estimation of 3 parameters (location, (x, y) and strength q).

**Reply:** *Throughout the text we mentioned that there is no guarantee in the convergence of the SA and we confirmed (based on the adequate bibliographical references) that the obtained network can be the optimal or the near optimal one. This complex combinatorial optimization approach retained a big attention in the literature and the SA is selected following the recommendations from more than one works of networks optimization in the framework of the atmospheric dispersion context (Abida et al, 2008; Abida et Bocquet, 2009; Jiang et al, 2007; etc.). Nevertheless before utilizing the probabilistic algorithm SA, we tested its performance in comparison with the Genetic Algorithm GA of evolutionary research technic (Kouichi, 2017). Concerning the statistical study after the achievement of the optimization we plan to perform this investigation as continuity of this first study.*

**Comment 3**. A big limitation of this approach is the subjectivity and biasness in the methodology and their dependence on the measurements. The optimality of the reduced set of receptors is shown based on the accuracy of the point source retrieval which is not relevant. The good retrieval results with presumed lesser number of receptors are not surprising since their chosen cost function depends on the measurement's values which always force the SA algorithm to choose the receptor locations with good model measurement accuracy. They completely ignored the fact that their network design criterion should be independent and prior to the knowledge of measurements, which is one of the big limitations. To be precise, the optimization methodology utilizes the same cost function for both the tasks: (i) Identifying a reduced set of receptors and (ii) retrieving the point source parameters. The cost function utilizes the actual measurements and measures the deviations between measured and predicted concentrations at the chosen set of receptors. The iterative SA algorithm tries to minimize this cost function, which means it selects the receptors with good model-measurement accuracy, i.e. which are closer to the measurements. This will eventually results in good retrieval depending on the model error. This clarifies the fact that the choice of receptors is always subjective to the model-measurement accuracy and will vary in case of perturbation in the model measurement variables. Thus, this is a poor approach and always biased towards model-measurement accuracy which do not signify the objective of optimization of receptors. The optimization of receptors should have performed independent and prior to the knowledge of measurements, which is not done here.

**Reply:** *This comment is similar to the comment n°1 and some evoked points in the introduction so we conserve the same responses. We hope that the clarification that we presented before help for best understanding the aim and the approach of this study.*

**Comment 4**. The study do not provide any insights / discussion on systems observability while remaining ill-posed, quantification of information gain or loss during optimization of the network, statistical or mathematical criterion leading to network optimization and sensitivity of the network design with respect to the perturbation in the model-measurement variables. Also, the study do not mention any optimality criterion, design of experiment or information theory criterion.

**Reply:** *The sensitivity of the network design with respect to the perturbation in the model-measurement variables is studied a posteriori and discussed in the revised version according to the recommendation of the Referee#2. Concerning the optimality criterion and the information theory we remind as we mentioned before (reply for comment 1) that the entropic criterion is defined by Kouichi (2017) and is the subject of another publication. We hope that the clarification that we presented before, specially the framework and the challenge of this study is now clear (mobile networks, with a priori information, etc.)*

**Comment 5**. Another issue with the methodology is that the SA algorithm may not converge always to the same set of reduced receptors. More often, there is high probability that the reduced set will change in every repeated simulation since the number of possible combinations are really high. In this case, how do you guarantee the optimality of design?. Also, the authors never compared between various such different sets corresponding to same trials as how they are varying or what are the differentiation between them. It seems that the authors just choose the arrangement with least reconstruction error which is not logical.

**Reply:** *This comment is similar to the comment n°2 and some evoked points in the introduction so we conserve the same responses that we clarified before.*

**Comment 6**. If the objective was simply to have a reduced set of network where model measurement errors are minimum (which is done here), why authors just did not select those locations where model predictions are matching with measurements?. This could be done simply by comparing model predictions and measurements instead of a massive SA computation. Based on the proposed approach, this can not be called an optimization of the monitoring network.

**Reply:** *Selecting the locations where model predictions are matching with measurements doesn't guarantee a 'best' estimation of the source parameters (inverse problem and direct problem are completely different). Effectively, this is one among the important results of this study. As we used a data assimilation approach, the best network is obtained for the best reproduction of the observations (i.e. correspond to minimal quadratic distance between the modeled and the measured concentrations). Also, as stated by Dr. Singh about the network based on the matching of direct model prediction with measurements, the direct model also required the knowledge of the exact source parameters and this information may not be available in general. However, in this study, we utilized only the concentration measurements and not the source parameters to obtain the monitoring networks.*

**Comment 7**. The proposed approach also raises questions regarding the efficiency of the network in case of perturbed model-measurement fields/variables. Also, the retrieved parameters are highly sensitive toward the design of their network which raise further questions regarding the efficiency of the chosen network. The optimized choice of network will always be subjective with respect to the wind variability, model, model errors and measurements. In trials, where model does not perform well, the error will always be high, for

example see in trials 2. This will raise the issue of failure of their monitoring networks in identifying correctly the emissions in case of large model errors. This is why the arrangement of the receptors vary in all the trials, even when in some trials, the wind conditions are approximately similar.

**Reply:** *This study presents our first attempt of sensors networks optimization for the reconstruction of releases source in urban domains. The door still open for continuity in order to integrate the effect of meteorological conditions variability or to integrate the influence of the models errors. Nevertheless, following the recommendation of the Referee#2, the effect of random measurement errors is now integrated into the analyses of the optimal networks performance. Some limitations of this work are also included in the updated version of the manuscript. It is not always true as Dr. Singh commented that in trials, where model does not perform well, the error will always be high, for example see in trials 2. In trial 2, the predicted concentrations from direct model were in good agreement with the observations (NMSE = 0.17, Correlation coefficient = 0.95, Index of Agreement = 0.97) (Kumar et al., 2015a). Even by utilizing the concentration measurements from all 40 sensors in source estimation, the retrieval error was large in this trial 2 (Kumar et al., 2015b). Also, all problems of the optimizing the network, e.g., networks without using concentration measurements, one single network for all meteorological conditions, etc cannot be deal in a single study. The optimal networks dealing with some of these problems were partially presented in the PhD thesis of Kouichi (2017) and will be presented in separate publications.*

**Comment 8**. The study also discusses about weights which they, later, referred as visibility functions highlighting prior informations regarding emissions. However, authors never mention "why they could not determine a criterion based on just visibility weight functions"? Which could be far relevant and independent to the measurements.

**Reply:** *The method that we described to assure the optimization is not unique, it is possible to use the visibility as optimality criterion and it is a different approach. In any case, we mentioned that the renormalized errors criterion is the unique cost function that can be used to assure the design in such problem configuration. The optimization approach only based on the visibility weight function was also performed as another research problem in the PhD thesis of Kouichi (2017). Dr. Singh is well aware of this study and corresponding partial results as he was also present in the final PhD viva presentation of Hamza Kouichi.*

**Comment 9**. It is not clear why they could not find a common optimal network which could work in all the trials for point source retrieval?. The original network of 40 receptors was already working in all the trials irrespective of model errors and varying meteorological conditions. It is useless to propose different configurations based on different meteorological conditions since meteorology can never be constant in a real scenario. The different configurations for different trials again highlight subjectivity of their approach. Thus, the study do not bring any significant outcome regarding the optimization of receptors.

**Reply:** *As we clarified before (see first reply in page 2 / case of permanent static network), finding a common optimal network which could work in all the trials is a different optimization problem, also an adequate optimality criterion (entropic criterion extended for several meteorological situations) is defined by Kouichi (2017). Dr. Singh is well aware of this study progress and corresponding partial results as he was also present in the final PhD viva presentation of Dr. Hamza Kouichi. We remind, this study is specific to emergency situations where the meteorological conditions can be known in real time and don't varying*

*significantly (it is assumed as stationary because the problem of optimization in urban environment is very complex and this is our first tentative in this framework).*

**Comment 10**. How do you describe physical features and efficiency or quality of the reduced configuration?. This is never mentioned in the results and discussion. The discussion mainly involves only source retrieval.

**Reply:** *We analyzed qualitatively (structures of the optimal networks in the instrumented area) and quantitatively (errors in source reconstruction) and the results showed that no trend is obvious thus proves that the problem of sensors networks optimization in urban environment is not trivial also the reduction of an original network achieved successfully and the performance for source reconstruction is maintained.*

**Comment 11**. Why their optimization always results in finding most of the sensors (5-6 detectors in the reduced configuration) close to the source location? It was never explained in the text. Does your optimized choice of receptors depends significantly on the receptors closer to the source location?. If yes, then what is use of optimizing since you will never know the source in accidental scenarios?

**Reply:** *Most of the sensors are selected by the SA close to the source location and this tendency is logical because these sensors make the area around the real source well visible from the network, nevertheless this doesn't guarantee a correct reconstruction of the source. As examined by Kouichi (2017), in some cases a limited number of sensors close to the source are not enough also adding sensors to a 'key configuration' don't improve the precision in source parameters estimation thus prove that the reduction of the number of sensors is justified. We remind that this study is for a specific need where a priori information (i.e. the measurements) is available and the network are optimized to be deployed in emergency situations.*

**Comment 12**. Why signal to noise ratio is not shown for all the reduced configurations? and it should be compared with the original network?

**Reply:** *Can you please clarify what do you mean by 'signal to noise ratio' in this context?*

**Comment 13**. The authors have simply described the errors in retrieving the location and intensity of the source. The responsible reasons behind it were never explained?. This shows that authors are just interpreting the retrieval rather than really analyzing the results.

**Reply:** *This comment is similar to the comment n°10 so we conserve the same responses.*

**Comment 14**. What is the role of weight function in your reduced configuration?. Does it have any effect on the systems observability and how it do affect your retrieval?

**Reply:** *Fundamentally, the role of the weight function in the renormalized data assimilation is minimizing the over interpretation of the observation, concerning the optimal networks no evident trend is relieved this confirm the complexity of such problem.*

**Comment 15**. Why did you describe vectors on the figures of the source retrieval?. While it seems that you are retrieving source parameters in a weighted least-squares framework?. It

was never explained in the results that what is the impact of reducing the receptors on the retrieved general source vector?.

**Reply:** *The source vector **s** is not described separately but analyzed with the visibility field obtained by the optimal networks and for each trial of the MUST experiment. The goal is to assure a qualitative examination and to validate the fact that the optimal network covers correctly the source position. Also to confirm that in the monitored area only one punctual source is detected, in the figures by analyzing the level in the source vector we confirm that the maximum is unique so the estimated source after the reduction of the original network size is physically coherent.*

**Comment 16**. A general choice of using a weight matrix in a least-squares methodology is measurement error covariance matrix. Authors did not justify how could they utilize matrix Hw as an alternative to measurement covariance matrix?. In addition, the Hw matrix is not a diagonal matrix which means that using Hw as a weight matrix will induce unphysical correlations among receptors which could be false as well. Did you analyze their impact on source retrieval?, If not, then why and how could you use them directly? Perhaps, you could assume an unity matrix. If not, why?

**Reply:** *As explained by Issartel et al. (2012), classically, the least squares are weighted using the covariance matrix (Hw) of the measurement errors. However, in practice, this matrix cannot be determined for the prevailing part of these errors arising from the limited representativity of the dispersion model. Issartel et al. (2012) proposed an alternative weighting based on a matrix Hw introduced by Issartel et al. (2007), that is related to a unified approach of the parametric and assimilative inverse problems corresponding, respectively, to identification of the point of emission or estimation of the distributed emissions. The weighting was shown to optimize the resolution and numerical stability of the inversion (Issartel et al., 2012). The importance of the most common monitoring networks, with point detectors at various locations, is stressed as a misleading singular case. As discussed by Issartel et al. (2012), it is possible to understand a drawback of two classical choices of Hw as the identity matrix, associated with the ordinary norm, or as the diagonal covariance matrix of noise supposed to be uncorrelated in the various measurements. The justification to utilize the matrix Hw is proposed and given in Issartel et al. (2012). A brief discussion about the justification is presented in the revised version.*

**Comment 17**. Another issue is with the presentation of the methodology. The study begins by posing an under-determined inverse problem of estimating state of emissions while their objective was to optimize a reduced set configuration for point source retrieval which is an overdetermined inverse problem. Why authors did not begin by directly posing the over-determined problem of point source retrieval?. Why they have presented unnecessary details regarding more general inverse problem of estimation emission state if it was not their objective?. The presented details were already published by several researchers in the literature. Further, authors again formulate the point source retrieval problem in a weighted least-squares sense. Why?. Why two different methods were presented for the same problem?.

**Reply:** *Before presenting the optimization methodology a brief mathematical formulation of the renormalisation technique is presented for the simple reason that we cannot define the optimality criterion without presenting its origin and its physical signification. If we present directly the adequate cost function (i.e. normalized errors) this cannot be appropriate for readers that don't have any information about the renormalisation method. Concerning the*

*'detail' of point source estimation simply because for the quantitative analyses (i.e. performance in source parameters estimation) we use this method so it is logic to present this 'detail'. However, as suggested, this part is reduced subsequently in the updated manuscript.*

**Comment 18**. Why do you need to compute a general vector s of state of emissions?. The objective was just to estimate point source parameters which could be estimated with the weighted / non-weighted least-squares method?. Please clarify?.

**Reply:** *This comment is similar to the comment n°15 so we conserve the same responses.*

**Comment 19**. Again, in the results, figures highlights distribution of weights and vector s which was never related to their monitoring network optimization. Their presentation confuses the overall objective of the study. Do you propose an optimal design for point source retrieval or a general source retrieval?. The figures related to weights are never explained as why they were needed? or what information do they provide related to the monitoring arrangement?. The given explanation is just copy-paste from previous papers of the authors.

**Reply:** *This comment is similar to the comment n°15 & n°18 so we conserve the same responses. However, the figures related to the visibility function are discussed with respect to the corresponding monitoring networks in the updated version of the manuscript.*

**Comment 20**. Why authors did not compare the weights in comparison to their weights corresponding to the original receptors?.

**Reply:** *We compared the performance of the optimal networks in comparison to the original network this implicitly and indirectly inadequate the role of the weights for each network.*

**Comment 21**. Another issue is also related to the common base network among the 10 and 13 sensors network. Why their strong base network involves only 7 receptors? In general, the 10 sensors networks should be a subset of the 13 sensor network, if not then why?. Please clarify?. It is also surprising that in some trials the common base network involves only 3 sensors. This is unusual having so much variation in having common base sensors among 10 and 13 sensor network in trials. The authors should provide the reasoning behind?, not just mentioning the results.

**Reply:** *The analyses of the common structures (Skeleton) in the optimal networks confirm that the solution is not unique so more than one network can lead to a good estimation of the source parameters. This result is very important and is in coherence with the works of Kovalets et al. (2011) and Efthimiou et al. (2017) that confirmed for the same experimental data the best source reconstruction using 10 sensors is possible for 5 or 10% among a significative set of randomly networks.*

**Specific comments:**

1. Abstract, Page 1, line 7. The sentence "The optimal networks in the MUST urban regions enabled ...". Rephrase the sentence. How could an optimal network enable a reduction?. I would like to mention again that the reduced set of receptors were never proved optimal.

**Reply:** *The sentence is rephrased for more clarity in the updated version. However, other part of this comment is similar to the general comments n°2 & 5 and some evoked points in the introduction so we conserve the same responses that we clarified before.*

2. Abstract, Page 1, line 11. The sentence "This study presents first application of the renormalization data assimilation approach for the optimal network design : : :.." is overstated and wrong. I could not find where and how did you apply renormalize data assimilation for optimal network design. Renormalize assimilation is only for retrieving the source parameters. I do not see in the text, how it could retrieve the reduced set of receptors. Also, you have interpreted a least-squares framework without justifying their inherent equivalence and choice of parameters with respect to the renormalization. Why?.

**Reply:** *We modified this sentence in updated version, however, the late part of this comment is similar to more than one comments and some evoked points in the introduction so we conserve the same responses. We hope that after the clarification, the aim and the presented methodology are now clear. A brief discussion about the justification of the weighted least-squares with respect to the renormalization framework is presented in the revised version.*

3. Page 1, line 18, the sentence "However, pre-deployment of these limited number of sensors : : :..". Please clarify, how could a pre-deployment of sensors helps to achieve maximum information from set of noisy concentration measurements. The objective of pre-deployment of sensors is to have maximum a priori information regarding state of emission and to correctly capture the data while extracting and utilizing information from the data is the final task of data fusion techniques.

**Reply:** *This comment is similar to more than one comment and some evoked points in the introduction so we conserve the same responses.*

4. Page 2, line 2. The sentence "detection of an unknown continuous point source's parameters ..." is wrong. How could you detect point source parameters? They are rather retrieved or estimated.

**Reply:** *We agree with this comment and accordingly the sentence is modified.*

5. Page 2, line 12. See the sentence "The establishment of an optimal network requires...". Why do you think that for an optimal network it requires availability of concentration measurements?. Please justify? Measurements may be required for the evaluation or validation but for establishment a network can be made with the meteorology and dispersion model.

**Reply:** *This comment is similar to more than one comment and some evoked points in the introduction so we conserve the same responses. This point was explained earlier in detail.*

6. Page 2, line 20, See the sentence "This approach includes the geometric and flow complexity inherent : : :..". I do not think if there is any inverse approach which includes such things for optimization process. The flow variables are accounted through the model, perhaps in the inverse approach in the form of sensitivities which is also derived from adjoint model. All the STE approach can include such information from model.

**Reply:** *The approach term written here doesn't mean to refer the inversion or STE approach only. It refers to the whole methodological approach to optimize the network by coupling the optimization techniques, inverse tracers transport modeling and Computational Fluid Dynamics. It is modified to avoid the confusion.*

7. Page 2, line 26. What is "regularized norm square". I do not think Sharan et al., 2012 have included such terms.

**Reply:** *The sentence is modified.*

8. Page 3, lines 3-5. Issartel, 2005, Sharan et al., 2009, 2012 and Kumar et al., 2015b, do not discuss any iterative algorithm to minimize the difference between observed and modeled concentration.

**Reply:** *The sentence is modified.*

9. Page 3, line 8, "does not required". Please correct the sentence.

**Reply:** *Corrected.*

10. Page 3, lines 16-17, Kumar et al. (2015b, 2016) do not provide any extension to renormalized inversion. It was just an application with CFD model.

**Reply:** *Modified.*

11. Page 3, lines 22-29. The authors defined the objective to determine optimal network but never achieved. In line 23, "A methodology was proposed : : :: : :.". If the objective was to better characterize the source, why one need to reduce the number of receptors. The reduction of receptors simply refers to the reduction of information regarding the observability of state of emissions. In line 26, "For this purpose : : :., but with comparable information". Where do you show in the manuscript that the reduced information is comparable to the original network. In line 27, "This work explores with two requirements : : :...", What does it mean, there is no movement of sensors. You have performed only selection of sensors.

**Reply:** *This comment is similar to more than one comment and some evoked points in the introduction so we conserve the same responses.*

12. Page 4, line 2, please correct "concentrations measurements".

**Reply:** *Corrected.*

13. Page 4, line 4, please correct "an horizontal plane".

**Reply:** *Corrected.*

14. Page 4, lines 8-10, The sentence "This study deals with linear relationship, as except from the nonlinear chemical reactions : : :.." is wrong. Most of these process are nonlinear in your case due to complex flow structures.

**Reply:** *This statement was based on the cited reference and we will verify it again and remove/correct it accordingly.*

15. Page 4, line 26, citing Kumar et al., 2016 is not appropriate. Please cite appropriate reference.

**Reply:** *We will check and correct the appropriate reference if needed.*

16. Page 4, lines 28-30, I understand that the inverse solution from Eq. (2) will lead to peaks at the grid cells coinciding with the measurement cells. This is obvious since the sensitivity matrix has peaks at the measurement cells which are reflected in the inverse solution. However, the difficult part to understand is "why do you call these large values of sensitivities at measurement cells an artificial information? Or a virtual/unphysical reality?". The peak at measurement cells is obvious since the concentration is always maximum at source location and in adjoint computations, you have replaced your measurement cells as source. Please clarify.

**Reply:** *These points were already clarified in many papers of Dr. J.-P. Issartel (main developer of the Renormalization inversion theory) and also in some other papers of his coauthors including Dr. Singh. In fact, this was one of the main force to the development of the renormalization inversion technique for source estimations which Dr. Singh has also used a lot in his papers.*

17. Page 5, line 5, Why do you think that normalizing peaks in the sensitivity matrix with weights will cure the peak problem. I mean, even if you normalize peaks (infinitely large value) by some nonzero weight values, it will not change anything.

**Reply:** *We don't understand this comment can you please more clarify?*

18. Page 5, line 9, citing Kumar et al., 2016 is not appropriate. Please cite appropriate reference.

**Reply:** *We will check and correct the appropriate reference if needed*

19. Page 6, line 3, "but with comparable information". It is never done in the study.

**Reply:** *What do you mean by" It is never done in this study"? In evaluation of the optimized networks, we have compared the source estimation performances of the obtained optimal networks of reduced number of sensors with the original network of 40 sensors. The obtained optimal networks provided comparable estimates of the source parameters with the estimates obtained from the original network of 40 sensors.*

20. Page 6, line 4, "delivers maximum of the information". It is never shown in the study.

**Reply:** *It is statement that refers to define one of the objective of an optimized monitoring network to deliver maximum of the information in respect to the source estimation from reduced number of sensors in an optimized network. We do not think any wrong in this sentence.*

21. Page 6, line 11, why cost function is defined according to Hw norm. Please clarify?.

**Reply:** *Because our approach is based on the renormalized data assimilation method as an inverse technique. It was explained in several studies, originally discussed in a study by Issartel et al. (2012). It is more clarified in the updated version of the manuscript.*

22. Page 6, line 12. Check sentence "As the cost function is convex, its minimum value must correspond to the maximum intensity of the source". WHY?. Please clarify?. The intensity of source here is q. Check your mathematical expressions. You will obtain an estimate of q for each location vector x. In such case, the maximum value of q in the domain may go to infinity

in case where weights w are very small or zero. It is not necessary that the maximum value of q will occur at the minimum of the cost function or at the source location.

**Reply:** *We agree with Dr. Singh's remark about the cost function which was mistakenly stated that way. This sentence is removed/modified from/in the updated version of the manuscript as it doesn't affect the subsequent part of the methodology.*

23. Page 6, line 15. This is not clear how you can utilize matrix H_w in place of measurement error covariance matrix in Eq. (9). This is not obvious. Justify?.

**Reply:** *As it was explained in several studies, originally discussed in a study by Issartel et al. (2012), it is more clarified in the updated version of the manuscript.*

24. Page 6, line 16, why two notations q and q0 (section 2.4) are utilized for the same representation?

**Reply:** *Thank you for pointed it out. We have corrected it to the same symbol throughout the manuscript.*

25. Page 6, line 19, "conditions of maximum intensity: : :.". It seems that authors have difficulties in understanding mathematics. Why equating first order derivative to zero will give maximum intensity?

**Reply:** *The sentence is corrected. It is just a simple derivative test of a function to find its critical points that determines whether each point is a local maximum, a local minimum, or a saddle point. Here, equating the first order derivative to zero leads to an estimate of the source intensity.*

26. Page 6, line 25, The expression for equation (13) is wrong. Please correct it.

**Reply:** *Corrected.*

27. Page 6, line 26. The mathematical expression is wrong.

**Reply:** *It seems to be correct; however, we will verify it again and modify if needed.*

28. Page 6, line 27. How do you guarantee the global minimum in SA algorithm?.

**Reply:** *This comment is repeated again and similar to more than one comments and some evoked points in the introduction so we conserve the same responses.*

29. Page 7, line 16, "average of the difference of the cost functions calculated for a large number of cases". How did you compute it?

**Reply:** *This is for estimating the starting 'temperature', the procedure consists in generating a set of randomly networks of same size then in evaluating the quality of each network using the renormalization algorithm.*

30. Page 7, line 19, "An equilibrium is reached : : :.". This means that the SA algorithm stops when cost function becomes constant. Then how do you prove global minimum?

**Reply:** *This comment is similar to more than one comments and some evoked points in the introduction so we conserve the same responses.*

31. In combinatorial optimization problem, it is not necessary that SA will provide the same solution or same set of receptors as the converged solution. In this case, how did you choose the solution?.

**Reply:** *This comment is similar to more than one comments and some evoked points in the introduction so we conserve the same responses. All the details about the SA and haw it is employed in this context are available in Kouichi (2017).*

32. Page 8, step 2 and step 3 show that the choice of receptors depends on the measurements. Thus, there is no optimality in network. The authors have simply chosen the reduced set of receptors based on good retrieval results which is biased.

**Reply:** *The question is repeated again and the answers to these points are already given to his previous comments.*

33. Page 9, lines 8-11. These are not clear. What do you mean by near-optimal network?. What are the conditions for "near overall optimum condition".

**Reply:** *As explained before, all this details are available in the cited references in the literature.*

34. Page 11, line 11, "The network optimization process : : :..". This shows that the choice of receptors are biased towards the model-measurement accuracy.

**Reply:** *Again the same question is repeated and so we conserve the same responses as given earlier.*

35. Why the configuration of networks should vary based on meteorological stability conditions.

**Reply:** *This is evident because the arrangement of the sensors in the monitored area should be according to the wind direction for example. It is clear that we cannot place sensors in regions were no measurements will be collected. We remind that the objective of this study is not to optimize a permanent static network.*

36. Page 12, lines 8-9. "These results exhibit that the SA : : :..". How it prove optimality of monitoring network?

**Reply:** *The answer is already given to this question as this comment is similar to more than one comment and some points in the introduction.*

37. Page 12, line 19. "were observed independently of the number of sensors". Why?. Please clarify?.

**Reply:** *This is showed in Table 2. However, the sentence is modified to avoid any confusion.*

38. Page 12, line 20, "larger location errors do not : : :.". Why?. Please clarify?

**Reply:** *This is noted in the results.*

39. Page 12, line 23, "a large number of sensors are close to the source positions ..". Why so?. Please justify?.

**Reply:** *This is noted in the qualitative analyses. This point is already clarified in response to a similar earlier comment.*

40. Page 12, line 29, "visibility functions have significant levels : : :..". What does this mean?.

**Reply:** *This is noted in the qualitative analyses please see figures 4 & S2.*

41. Page 13, line 6, "increase in the number of sensors in a network has little influence on the accuracy: : :." Why? Please clarify?

**Reply:** *This is clear in the text because by adding sensors in a performance network can more add measurements and model errors in the methodology which may affect the estimation of the parameters of the source.*

42. Page 13, lines 8-10, "In some trials, it was also noted : : :..not necessary benefial." How could you justify this?. In fact, how did you evaluate if information added by a sensor is fruitful or not? If just based on accuracy of source retrieval, then it is illogical? If an information added by a new sensor is not beneficial then why it should increase the location error?. Also, how location error may decrease with decreasing the number of sensors? You can say this just because you are looking at source retrieval estimates. However, reducing the receptors will make the source retrieval unstable and more sensitive to the noise.

**Reply:** *We tried to explain these points in this section of the manuscript as it was observed in some trials. The distance of an estimated source to real source was observed to decreases with an increase in sensors number and also with the decrease number of sensors in some other cases. At this moment, we do not have a theoretical explanation for this behavior. However, it is also logical that increasing the number of the sensors after a number may not always provide the best estimation because with addition of the more no. of sensors, we also add more model and measurements errors in the estimation process. These errors may affect the source estimation results in some trials. As pointed out by Dr. Singh, reducing the receptors will make the source retrieval unstable and more sensitive to the noise, this problem may also required a theoretical justification of the minimum number of sensors needed for source estimation. And it is beyond the scope of this study. However, these limitations are now explained in the updated manuscript.*

43. Page 13, line 17, What do you mean by ": : : diversity of structures independently of the number of sensors." You computation is based on fixed number of sensors and there is no discussion of diversity. However, in the following explanation, this is not understandable that why you have different number of common networks in different trials?. It seems that optimized choice of receptors is not really optimized. Otherwise, why in some trials (1, 11), only 3 sensors are found as a common base?. Also, even having 7 receptors as common in 10 and 13 sensor arrangement can not be called a strong common base. There is no explanation why all the 10 sensors are not subset of 13 sensor network. If, it was really optimized that it must have been. If not so, then please explain why?

**Reply:** *This comment is similar to the major comments n°21 so we reserve the same responses that we clarified before.*

44. Page 13, line 22, "The performances do not systematically converge independently to the size of the networks". What does it mean and why it does not converge?. Further, it is

mentioned that in trial 1, 13 sensor network leads better performance than a 10 sensors network and algorithm leading to near global optimum is contained in the 13 sensor network. This really proves that fact that you choice of receptors is biased by your source retrieval which is not the objective of an optimal network.

**Reply:** *This comment is similar to more than one comment and some evoked points in the introduction so we conserve the same responses. However, we wish to point out again that the choice of the receptors were not determined based on the source estimation. These optimal sensor locations were independently estimated based on the measurements, adjoint functions, and SA algorithm. The source estimation is performed only to validate the performance of obtained optimal networks.*

45. Page 13, line 27. Why there is no common trend in skeleton network observed in several trials?. There must be at least with similar flow conditions. If not, justify?

**Reply:** *This comment is similar to more than one comment so we conserve the same responses.*

46. Page 13, line 29, "optimal networks can satisfy conditions of a near overall optimum (to be minimized)". What are the near overall optimum conditions?. If you are referring "minimization of cost function". This is a wrong approach.

**Reply:** This comment is similar to more than one comment. *Throughout the text we mentioned that there is no guarantee in the convergence of the SA and we confirmed (based on the adequate bibliographical references) that the obtained network can be the optimal or the near optimal one. This complex combinatorial optimization approach retained a big attention in the literature and the SA is selected following the recommendations from more than one works of networks optimization in the framework of the atmospheric dispersion context (Abida et al, 2008; Abida et Bocquet, 2009; Jiang et al, 2007; etc.). Nevertheless before utilizing the probabilistic algorithm SA, we tested its performance in comparison with the Genetic Algorithm GA of evolutionary research technic (Kouichi, 2017).*

**References**

*Abida, R., Bocquet, M., Vercauteren, N., Isnard, O., (2008). Design of a monitoring network over france in case of a radiological accidental release. Atmospheric Environment, 42, 5205–5219, doi: 10.1016/j.atmosenv.2008.02.065.*

*Abida, R, et Bocquet, M., (2009). Targeting of observations for accidental atmospheric release monitoring. Atmospheric Environment, 43, 6312–6327, doi: 10.1016/j.atmosenv.2009.09.029.*

*Ainslie, B., Reuten, C., Steyn, D. G., Le, N. D., & Zidek, J. V. (2009). Application of an entropy-based Bayesian optimization technique to the redesign of an existing monitoring network for single air pollutants. Journal of environmental management, 90(8), 2715-2729.*

*Berry, J., Hart, W., Phillips, C. A., and Watson, J. P., (2006-a). A facility location approach to sensor placement optimization. 8th Annual Symp.on Water Distribution Systems Analysis, Cincinnati, Ohio, Environmental and Water Resources Institute of ASCE (EWRI of ASCE), New York.*

Chen Q, Wu W, Blanckaert K, Ma J, Huang G. Optimization of water quality monitoring network in a large river by combining measurements, a numerical model and matter-element analyses. J Environ Manag, 2012;110: 116–124. doi: 10.1016/j.jenvman.2012.05.024 )

Hamel, D., Chwastek, M., Garcia, M., Farouk, B., Kam, M., Dandekar, K. R., (2010). SensorPlacement for Urban Homeland Security Applications. International Journal of Distributed Sensor Networks, vol. 2010, Article ID 859263, p15.

Issartel, J.-P., Sharan, M., and Modani, M.: An inversion technique to retrieve the source of a tracer with an application to synthetic satellite measurements, Proceedings of the Royal Society of London A: Mathematical, Physical and Engineering Sciences, 463, 2863–2886, https://doi.org/10.1098/rspa.2007.1877, http://rspa.royalsocietypublishing.org/content/463/2087/2863.abstract, 2007.

Issartel, J-P., Sharan, M., Singh, S.K., (2012). Identification of a point of release by use of optimally weighted least squares. Pure Appl. Geophys., 169(3), 467–482, doi:10.1007/s00024- 011-0381-4.

Kouichi, H. (2017). Sensors networks optimization for the characterization of atmospheric releases source, Theses, Université Paris Saclay, France. https://hal.archivesouvertes.fr/tel-01593834v2

Kouichi, H. Turbelin, G. Ngae, P. Feiz, A. A. Barbosa E. & A. Chpoun (2016), Optimization of sensor networks for the estimation of atmospheric pollutants sources, WIT Transactions on Ecology and The Environment, Vol 207, doi:10.2495/AIR160021.

Krause, A., Leskovec, J., Guestrin, C., VanBriesen, J., Faloutsos, C., (2008). Efficient Sensor Placement Optimization for Securing Large Water Distribution Networks. Journal of Water Resources Planning and Management (JWRPM).

Kumar, P., Feiz, A.-A., Ngae, P., Singh, S. K., and Issartel, J.-P.: CFD simulation of short-range plume dispersion from a point release in an urban like environment, Atmospheric Environment, 122, 645 – 656, https://doi.org/http://dx.doi.org/10.1016/j.atmosenv.2015.10.027, http://www.sciencedirect.com/science/article/pii/S1352231015304465, 2015a.

Kumar, P., Feiz, A.-A., Singh, S. K., Ngae, P., and Turbelin, G.: Reconstruction of an atmospheric tracer source in an urban-like environment, Journal of Geophysical Research: Atmospheres, 120, 12 589–12 604, https://doi.org/10.1002/2015JD024110, http://dx.doi.org/10.1002/ 2015JD024110, 2015b.

Lepley, J., Lloyd, D., Robins, A., Rudd, A., Wilks, A., 2011. Dynamic sensor deployment for the monitoring of chemical releases in urban environments (dyce). In: Chemical, Biological, Radiological, Nuclear, and Explosives (CBRNE) Sensing XII. Vol. 8018

Le, N. D., Sun, L., Zidek, J. V., (2003). Designing networks for monitoring multivariate environmental fields using data with monotone pattern. Tech. rep., Statistical and Applied Mathematical Sciences Institute, RTP, NC.

Mofarrah, A., & Husain, T. (2010). A holistic approach for optimal design of air quality monitoring network expansion in an urban area. Atmospheric Environment, 44(3), 432-440.

*Ma, D., Deng, J., Zhang, Z., 2013. Comparison and improvements of optimization methods for gas emission source identification. Atmospheric Environment 81 (Supplement C), 188 - 198.*

*Mason, L. R., et Bohlin, J. B., (1995). Network optimization of a radionuclide monitoring system for the comprehensive nuclear test ban treaty. Tech. rep., Pacific-Sierra Research Corporation, PSR Technical Report 2585.*

Saunier, O., Bocquet, M., Mathieu, A., Isnard, O., (2009). Model reduction via principal component truncation for the optimal design of atmospheric monitoring networks. Atmospheric Environment 43. 4940–4950.

*Watson, J.-P., Greenberg, H. J., Hart, W. E., (2004). A multiple-objective analysis of sensor placement optimization in water networks. Proc., World Water and Environmental Resources Conf., ASCE, Reston Va.*

---

## Author Comment (AC1) · 20 Jun 2018

**Response to the Reviewer #1**

The idea of the paper is to achieve the best result with as less as possible information. This idea is very innovative and I support any new effort. The application is the atmospheric dispersion in urban areas. The goal is to find the source when we know the flow field and the real concentration measurements.

I have one major comment/question.

When authors try to validate this approach they compare results of source inversion (distance to true source etc) with 'optimal network' of 10 sensors with the results obtained by the full network (40 sensors). Why don't they directly compare results of 'optimal' network of 10 sensors with the results of other networks of 10 sensors? Of course, there are too many of such networks. But by application of the combination same procedure as we did in Kovalets et al (2011) and Efthimiou et al (2017) they at least could prove that their 'optimal network' yields the results within say best 5 or 10% of the results that could be achieved with 10 sensors.

**Reply:** *We would like to thank Dr. Efthimiou for the positive feedback and we appreciate his comments/questions. We have carefully considered his comments and worked to include them in a revised version of the manuscript according to the proposed suggestions.*

*Dr. Efthimiou's above comment can be complied into two following comments and please find below the responses to these as follows:*

**Comment 1:** Why don't we directly compare results of the 'optimal' network of 10 sensors with the results of other networks of 10 sensors?

**Reply:** *The comparison with networks of the same size (10 sensors for example) is performed implicitly during the optimization process. The Simulated Annealing used in this study compares at each iteration two networks (of the same size) and retains the 'best one'. The networks are generated randomly like in Kovalets et al (2011) and Efthimiou et al (2017). Since the search space is quite large, the number of the compared networks is equivalent to the number of iterations. The comparison is based on a cost function named Normalized Errors Js and inspired from the renormalized data assimilation method. This cost function quantifies the quadratic distance between the observed and the simulated measurements according to the normalized Gram matrix $H_w$. The 'optimal network' produce the 'best' description of the observations (i.e. corresponds to the minimal quadratic distance) and permits a posteriori to reconstruct its origin. In figures 1, 2 & 3 is presented the evolution of the cost functions (trials 5, 11 & 19) during the optimization process. For these trials, ~ $3\times10^4$ networks of 10 sensors are compared. The challenge in our study is to design the networks without using a priori the parameters of the real source and without considering an acceptance level of networks quality (the solution is 'good' if it satisfies three fixed criteria of values rH ≤ 15 m, rV ≤ 2.5 m, δq ≤ 4 ) as performed in Kovalets et al (2011) and Efthimiou et al (2017). These points are more clearly discussed in the revised text.*

**Comment 2:** Why we compare results of source inversion (distance to the true source, etc.) with 'optimal network' of 10 sensors with the results obtained by the full network (40 sensors)?

**Reply:** *The results obtained by the optimal networks of 10 and 13 sensors are compared as a posteriori with the original network of 40 sensors used in MUST experiment and evaluated for source reconstruction by using the renormalization technique (Kumar et al, 2015b). As in practice, the number of measurements is limited, this comparison allowed concluding that in urban areas, the reduction of networks size is possible and does not degrade significantly its efficiency in source estimation. For more details, the choice of the size of the network (10 and 13) is fixed after observing that an acceptable estimation of the source in majorities of the trials was enabled by using minimum 8 sensors. Also by using more than 13 sensors optimal networks, the errors in source parameters estimation are stable and does not improve significantly (Kouichi, 2017). For this reason, the optimized networks were constructed and evaluated for sizes 10 and 13 (1/4th and ∼ 1/3rd of the original network of 40 sensors) with the original large network. These points are more clearly discussed in the revised text.*

**References**

1) I.V. Kovalets, S. Andronopoulos, A.G. Venetsanos, J.G. Bartzis, Identification of strength and location of stationary point source of atmospheric pollutant in urban conditions using computational fluid dynamics model, Math Comput Simulat, 82 (2011) 244-257.

2) G.C. Efthimiou, I.V. Kovalets, A. Venetsanos, S. Andronopoulos, C.D. Argyropoulos, K. Kakosimos, An optimized inverse modelling method for determining the location and strength of a point source releasing airborne material in urban environment, Atmos. Environ., 170 (2017) 118-129

3) Kouichi, H. (2017), Sensors networks optimization for the characterization of atmospheric releases source, Theses, Université Paris Saclay, France.

4) Kumar, P., Feiz, A.-A., Singh, S. K., Ngae, P., and Turbelin, G.: Reconstruction of an atmospheric tracer source in an urban-like environment, 15 Journal of Geophysical Research: Atmospheres, 120, 12 589–12 604, https://doi.org/10.1002/2015JD024110, http://dx.doi.org/10.1002/ 2015JD024110, 2015b.

---

## Author Comment (AC2) · 20 Jun 2018

**Response to the Reviewer #2**

In this paper, the authors formulate a simulated annealing algorithm with a renormalization inversion algorithm coupled to a CDF flow and dispersion model and apply it to the Mock Urban Setting Test (MUST) tracer field experiment (which simulates an 'urban-like' environment). The aim of the work is to demonstrate how the inversion technique presented can be useful in optimally placing a smaller number of concentration samplers for quantifying a continuous point source with almost the same level of source detection ability as the original larger number of samplers. The paper is well written, but in my opinion requires a major revision. My comments are as follows:

*The authors are grateful to the reviewer's remarks and thanking him/her for reviewing the manuscript. In light of the reviewer's all suggestions, the manuscript is revised. Reviewer's questions and remarks are repeated below (in red color) and our responses (in italic) follow each question.*

**Main comments**

**1)** The MUST experiments took place under neutral to stable and strongly stable conditions. However, the CFD model used is for neutral conditions and does not include the effects of atmospheric stability over the urban area (the only stability effects included are through the specification of inflow boundary conditions). Atmospheric stability has a profound impact on dispersion and would thus influence the adjoint functions. The authors should discuss the consequences of its neglect on the results and the errors it introduces.

**Reply**: *We agree with the reviewer's remark that the atmospheric stability effects in the CFD model fluidyn-PANACHE were included through the inflow boundary conditions. The used version of fluidyn-PANACHE was not capable of incorporating the atmospheric stratification through surface cooling or heating and whatever stability effects are included through the inflow boundary conditions. The fluidyn-PANACHE includes a Planetary Boundary Layer (PBL) model that serves as the interface between the meteorological observations and the boundary conditions required by the CFD solver. The PBL model is composed of two parts: (i) a micro-meteorological model that computes fundamental physical characteristics of the PBL from routine meteorological observations, and (ii) a boundary layer model for prescribing the vertical profiles of wind speed, temperature, and turbulence. However, as discussed in our previous study (Kumar et al., 2015a), even with the specification of the stability dependent inflow boundary conditions only, the predicting concentrations from the CFD model are in good agreement with the measured concentrations in the MUST experiment for all 20 trials in different atmospheric stability conditions. This may be due to that the scale and the urban geometry of the MUST field experiment are not large enough for the requirement to resolve the atmospheric stratification through surface cooling or heating. And the stability effects included through the inflow boundary conditions were enough to include the stability effects on the concentrations and adjoint functions at such a small scale*

*urban-like environment of the MUST field experiment. However, at microscales also, small irregularities can break the repeated flow patterns found in a regular array of containers with identical shape (Qu et al, 2011). In addition, uncertainties associated with the thickness and the properties of the material of the container wall also affect flow pattern and the resulted concentrations and adjoint functions (Qu et al, 2011). Also, for a real urban environment at the larger scales, the atmospheric stability will have a profound impact on dispersion and would thus influence the adjoint functions. And the stability effects through the specification of inflow boundary conditions only, may not be appropriate for those environments. In these scenarios, the CFD model should be capable of incorporating the atmospheric stratification through surface cooling or heating in real urban environments.*

*A brief discussion about these is now included in the revised manuscript.*

**2)** I have reservations about the usefulness of the methodology presented in real-world urban environments. The title of the paper states 'urban monitoring network' but there are no real urban configurations used. The MUST experimental domain was only 200 m x 200 m (with buildings represented by a grid of containers) which cannot quite represent an urban area in terms of scale, meteorological variability, or non-uniform terrain or roughness/canopy structure. So in a way the present study does not explore any aspects that are specific to urban environments. The authors should discuss this, particularly how their methodology could be applied and its limitations in real-world urban cases. Following on, the title of the paper should say 'urban-like' or something similar instead of 'urban'.

**Reply**: *We agree with the referee's remarks that the Mock Urban Setting Test (MUST) tracer field experiment was performed in an urban-like environment and cannot quite represent an urban area in terms of scale, meteorological variability, or non-uniform terrain or roughness/canopy structure. However, the MUST field experiment has been widely utilized for the validation of the atmospheric dispersion models in an urban-like environment. Although, as mentioned by the reviewer also, there are several limitations to utilize this experiment; but, the methodology presented here is general in nature to apply to a real urban environment also. The methodology involves the utilization of the CFD model which generally can include the effects of the urban geometry, meteorological variability, or non-uniform terrain or roughness/canopy structure in a real urban environment. Therefore the title that will appear on the revised version changed accordingly "Optimization of an Urban-like Monitoring Network for Retrieving an Unknown Point Source Emission". Also, the limitations of the present methodology, for its application in real-world urban cases are discussed in the revised version.*

**3)** There were a total of 40 concentration samplers. In their optimisation, the authors arbitrarily fixed the number of samplers to 13 and 10 and then determined the optimum positions of these reduced number of samplers from the original 40 samplers. A better question to answer would have been "what is the minimum number of samplers required and what their positions are in order to quantify the source with a given degree of confidence or accuracy?"

**Reply**: *The numbers of the sensors were not fixed arbitrarily. The trends of location error ($E_l$) and the ratio of the estimated to true source intensity ($E_q$) with the number of sensors from 4 to 16 are performed and the results are presented in Kouichi (2017). As already mentioned in the manuscript, the number of sensors in the optimized networks were reduced to $1/3^{rd}$ (13 sensors) and $1/4^{th}$ (10 sensors) of the total number of sensors (40) originally deployed because for some cases a small number of sensors could not allow to correctly reconstruct the source and divergences of the calculations have been noted. As an example for Trial 14, reconstructing the source by using a small number on sensors is not appropriate since 4, 5, or 6 sensors are not enough ($E_l > 100$ m and $\log(E_q) > 10E+17$). Also, after a certain number of sensors in the network, the source term estimation is not improved significantly (see Figure 1). Thus, selecting 10 ($1/4^{th}$) and 13 ($1/3^{rd}$) number of sensors in the optimal networks ensures an acceptable estimate of the source for all the trials. These points are more clearly discussed in the revised manuscript.*

[Figure]

Figure 1: Errors in the estimation of the source (a) position and (b) intensity in Trial 14. Here p is the number of sensors

The present optimisation is based on fixed meteorological conditions in a trial. In a real situation, the network design would also depend on diurnal and spatial variability in meteorological conditions (e.g. wind direction) which may increase or decrease the optimum number of sites. This, however, is not in the scope of the present study. Perhaps as a future study, the authors may consider using data from full scale field measurements such as Salt Lake City Urban 2000 experiment.

**Reply**: *As the problem is complex, in this first study each meteorological situation is assumed as stationary and described by wind speed and direction and stability class. However, we agree with the referee, the network design would also depend on diurnal and spatial variability in meteorological conditions which may increase or decrease the optimum number of sites and also may change the 'best positions' to be instrumented by sensors. Indeed, we envisage as continuity of this work, to study the effect of the variability of the meteorological conditions. As suggested by the reviewer, we would like to utilize and validate the present*

*methodology by using the data from full-scale field measurements such as Salt Lake City Urban 2000 experiment in a future study.*

**4)** Dense gas effects are included. How are they taken into account (or inverted) in the backward (i.e. retro plume) dispersion calculation for adjoint functions?

**Reply**: *Since the released tracer gas $C_3H_6$ in MUST field experiment is heavier than the air, a buoyancy model is used to model the body force term in the Navier-Stokes equations. The buoyancy model is suitable for the dispersion of heavy gases where density difference in the vertical direction drives the body force. Many attempts have been made in the literature to use CFD in simulating the dispersion of a negatively buoyant gas using a two-equation k-ε turbulence model (Sklavounos and Rigas, 2004; Tauseef et al., 2011, etc.). The fluidyn-PANACHE implementation of the k-ε model is derived from the standard high-Reynolds number (Re) form with corrections for buoyancy and compressibility (Launder, 2004; Hanjalic, 2005). The k-ε model computes the length and time scales from the local turbulence characteristics. Thus, it can model the turbulent flows subjected to both mechanical shear (obstacles, terrain undulations, canopy) as well as buoyancy (stability and buoyant/heavy gas plumes). For more information, the fluidyn-PANACHE is a three-dimensional (3-D) diagnostic model that solves Reynolds-averaged forms of the Navier-Stokes dynamics equations along with the equations describing conservation of tracer concentration, mass, and energy in the atmosphere (Fluidyn-PANACHE, 2010). As already mentioned in the manuscript, a detailed description of the fluidyn-PANACHE and its evaluation for the forward dispersion with the MUST field experiment was presented in our earlier paper (Kumar et al., 2015).*

**5)** What is the uncertainty in the source estimation results in Table 2? Is the approach capable of providing uncertainty estimates (like the Bayesian one)?

**Reply**: *With the present method, at this moment we cannot derive uncertainties like the Bayesian methods. However, we calculated posterior uncertainties on the source parameters estimation due to the measurements errors. In order to quantify the uncertainty, a 10% Gaussian noise was added at each measurement. Using the optimal networks 50 simulations for source characterization are performed for each trial. The average and the standard deviation of $E_q$ and $E_l$ are calculated and the results are present in Table 1 below. For the optimal networks, there is not an obvious trend and the uncertainties are in the same order of magnitude compared to the original network (40 sensors). The Table 2 of the actual version of the manuscript will be replaced by the following table 1 accordingly.*

| Run No. | $E_l^{40}$ (m) | $E_l^{13}$ (m) | $E_l^{10}$ (m) | $E_q^{40}$ | $E_q^{13}$ | $E_q^{10}$ | Skeleton Sensors |
|---|---|---|---|---|---|---|---|
| 1 | 3.3 ±1.3 | 19.6 ±12.13 | 33.76 ±5.30 | 0.92 ±0.08 | 1.04 ±0.23 | 1.24 ±0.22 | 3 |
| 2 | 42.9 ±23.8 | 31.91 ±8.80 | 56.88 ±9.51 | 4.01 ±1.57 | 3.21 ±0.41 | 5.12 ±3.63 | 4 |
| 3 | 10.8 ±1.6 | 9.01 ±2.47 | 9.01 ±3.02 | 1.17 ±0.27 | 0.71 ±0.16 | 0.71 ±0.16 | 7 |
| 4 | 22.8 ±7.7 | 18.07 ±1.84 | 18.07 ±2.61 | 0.27 ±0.35 | 0.83 ±0.21 | 0.83 ±0.26 | 6 |
| 5 | 21.9 ±2.1 | 2.13 ±2.54 | 11.56 ±4.21 | 0.57 ±0.07 | 0.95 ±0.05 | 0.67 ±0.05 | 6 |
| 6 | 5.0 ±1.6 | 6.96 ±0.19 | 6.96 ±0.00 | 2.14 ±0.60 | 1.04 ±0.06 | 1.04 ±0.04 | 7 |
| 7 | 12.4 ±9.1 | 18.85 ±9.08 | 12.99 ±1.67 | 0.41 ±0.49 | 3.11 ±0.51 | 1.06 ±0.07 | 4 |
| 8 | 15.8 ±12.1 | 12.86 ±1.28 | 15.79 ±1.05 | 2.22 ±0.90 | 1.32 ±0.34 | 1.76 ±0.11 | 6 |
| 9 | 7.7 ±1.2 | 8.20 ±0.35 | 8.08 ±0.00 | 1.37 ±0.07 | 3.06 ±0.17 | 7.55 ±0.39 | 5 |
| 10 | 8.8 ±3.0 | 8.00 ±4.57 | 8.00 ±5.68 | 1.08 ±0.19 | 1.08 ±0.77 | 1.08 ±1.07 | 8 |
| 11 | 19.8 ±5.0 | 17.19 ±12.00 | 17.19 ±7.06 | 1.67 ±0.12 | 1.62 ±0.40 | 1.62 ±0.26 | 3 |
| 12 | 7.4 ±6.6 | 5.43 ±11.69 | 10.22 ±9.10 | 0.95 ±0.06 | 0.85 ±0.28 | 0.2 ±0.04 | 4 |
| 13 | 7.7 ±0.6 | 8.63 ±4.36 | 8.63 ±3.86 | 0.97 ±0.07 | 0.78 ±0.18 | 0.78 ± 2.05 | 4 |
| 14 | 2.2 ±1.9 | 5.50 ±2.98 | 5.50 ±3.88 | 1.42 ±0.17 | 0.88 ±0.24 | 0.88 ±0.40 | 7 |
| 15 | 1.1 ±1.0 | 30.23 ±2.14 | 37.98 ±0.72 | 1.88 ±0.09 | 0.57 ±0.07 | 0.17 ±0.01 | 7 |
| 16 | 26.7 ±4.9 | 63.04 ±6.84 | 29.80 ±9.86 | 1.70 ±0.06 | 0.29 ±0.06 | 0.67 ±0.23 | 5 |
| 17 | 7.0 ±1.9 | 14.07 ±2.78 | 23.05 ±10.44 | 0.90 ±0.05 | 1.10 ±0.04 | 1.52 ±0.16 | 6 |
| 18 | 14.3 ±11.0 | 12.83 ±4.18 | 12.83 ±4.61 | 1.15 ±0.46 | 1.15 ±0.16 | 1.15 ±0.21 | 6 |
| 19 | 22.3 ±6.4 | 10.77 ±4.25 | 13.46 ±4.8 | 1.76 ±0.16 | 0.99 ±0.20 | 0.83 ±0.25 | 6 |
| 20 | 32.5 ±1.8 | 45.23 ±1.78 | 44.29 ±0.31 | 0.83 ±0.04 | 1.68 ±0.06 | 1.56 ±0.06 | 7 |

Table1. Source estimation results from the different monitoring networks for each selected trial of the MUST field experiment

**6)** How does the uncertainty in the results in Table 2 change as the number of samplers is changed? Have you included model and measurement uncertainties in the methodology?

**Reply**: *A general relationship between the number of samplers and the uncertainties is not obvious. We noticed that changing size of the network (increasing or decreasing the number of sensors) can lead to the growth or diminution of the uncertainties in the source parameters estimation. As an example in Table 1, for Trial#7 uncertainties grow while for Trial#17 uncertainties diminish.*

*Accordingly to the answers of questions 5 and 6, results and interpretations of the effect of measurements errors on the source parameters estimation are included in the revised text.*

**7)** Section 2.3: Is there a sensitivity of the source estimation / optimisation to how the weight function is selected? Could there be any other choices of the weight function?

**Reply**: *The weight function is selected to minimize the information retrieved from the observations thus avoiding inversion artifacts close to the detectors positions. This optimal renormalizing function denoted φ(x) is unique as demonstrated by Issartel (2004). However, the sensitivity of the source estimation is essentially due to the information provided by each vector of measurements.*

**8)** Did you specify any a priori bounds on the estimated source position and source emission rate? If yes, what were they?

**Reply:** *In this study, we do not require to specify any a priori bounds on the estimated source position and source emission rate in the renormalization inversion technique.*

**9)** What is the advantage of the present technique compared to, say, the Bayesian approach which also provides probability associated with the solution?

**Reply**: *The technique used in this study does not require a priori information about the source (i.e. location and intensity) or about the measurements (i.e. knowledge of the observation-Error Covariance Matrices). The renormalization is a deterministic inversion method compatible with upstream offline preparation for network implementation and compatible with rapid implementation for the monitoring operation phase for local-scale applications around sensitive sites. Also, this method can be used to estimate a point or distributed source which can expand the cases studied.*

**10)** Page 3, line 15: 'The Gaussian models are unable to capture. . .' While this may be generally true, a well formulated Gaussian plume model can describe idealised urban dispersion (e.g. Huq and Franzese, BLM, 147, 102-121, 2013).

**Reply**: *Corrected accordingly as 'In general, the Gaussian models are unable to capture. . .'*

**11)** Section 5: Was the CFD model validated using the MUST data for its ability to simulate the measured concentrations?

**Reply**: *As already mentioned in the manuscript, the ability of the CFD model to simulate the measured concentrations using the MUST data and the prediction errors of the forward simulations used in this study were discussed in our previous study (Kumar et al., 2015).*

**12)** Source position was calculated. Does it include the source height too? Was source height a free parameter or a fixed one?

**Reply**: *The source height was not calculated in this study. The computations were carried out in the 2-dimensional domain on a horizontal plane corresponds to an altitude of known source height Hs. Accordingly, the vertical dimension was eliminated in the formulations and the computations. Consequently, the adjoint functions were chosen as steady state retroplumes on the horizontal cross-section area passing through a plane $z = Hs$. The assumption with respect to the vertical structure of the problem is useful to estimate the ground level sources or the emission sources along a horizontal cross-section area passing through a fixed vertical level. However, in this study, the problem of vertical structure (i.e. the height of a source) in three-dimensional space of an urban area is not addressed. In reality, an altitude of a release (i.e. source height) is also not known and required to estimate along with the projected release location on the ground surface and the release rate (Kumar et al., 2016). We envisage to include the height of the source in a future study.*

**Other comments**

13) Page 2, line 14: What is 'an NP-hard problem'?

**Reply**: *The problem of sensors network optimisationis NP-Hard (i.e. Non-deterministic Polynomial-time hardness) as shown by (Ko et al., 1995), which means that it is difficult for an exhaustive search algorithm to solve all instances of the problem because it's need a considerable time.*

14) Page 2, line 35: 'probabilities' should be 'probability'.

**Reply**: *Corrected. We have now carefully checked the manuscript to eliminate possible linguistic errors.*

15) Page 3, line 8: 'required' should be 'require'.

**Reply**: *Corrected.*

16) Page 3, line 10: 'the continuous' should be 'continuous'.

**Reply**: *Corrected.*

17) Page 3, line 23: 'was' should be 'is'.

**Reply**: *Corrected.*

18) Is the optimisation methodology presented only valid for a single source?

**Reply**: *In this study, the presented optimization methodology is only valid for a single source. Nevertheless, it is possible to consider the optimization for multiple sources. We envisage that evaluation in the future.*

19) Page 7, line 3: The term temperature should be put in quotes as this is not a real temperature in the present context.

**Reply**: *Changed accordingly.*

20) Page 9, line 2: 'stopped' should be 'is stopped'.

**Reply**: *Corrected.*

21) Figures 1 and 3: Why some of the 40 samplers locations do not coincide in these figures?

**Reply:** *In the schematization of the MUST experiment, the position of the tenth detector of the fourth row was slightly shifted from its true position. Figure 1 in actual manuscript version is adjusted accordingly as shown in figure 2 below.*

[Figure]

Figure 2: Schematization of the MUST experiment: (a) correct version (b) adjusted version

22) Is the code for simulated annealing algorithm with the renormalization inversion technique available?

**Reply:** Yes *the codes are available for one trial as an example.*

**References**

*Fluidyn-PANACHE, 2010. User Manual. FLUIDYN France / TRANSOFT International, version 4.0.7 Edition.*

*Hanjalic, K., 2005. Turbulence and transport phenomena: Modelling and simulation. In: Turbulence Modeling and Simulation (TMS) Workshop. Technische Universitat Darmstadt.*

*Issartel, J.-P. Emergence of a tracer source from air concentration measurements, a new strategy for linear assimilation. Atmospheric Chem. Phys. 5, 249–273 (2005). URL* http://www.*atmos-chem-phys.net/5/249/2005/. DOI 10.5194/acp-5-249-2005.*

*Ko, C.-W., Lee, J., Queyranne, M., 1995 An exact algorithm for maximum entropy sampling. Operational Research, 43, 684–691. URL https://doi.org/10.1287/opre.43.4.684.*

*Kouichi, H. (2017). Sensors networks optimization for the characterization of atmospheric releases source, Theses, Université Paris Saclay, France. https://hal.archivesouvertes.fr/tel-01593834v2*

*Kumar, P., Feiz, A.-A., Ngae, P., Singh, S. K., Issartel, J.-P., 2015. CFD simulation of short-range plume dispersion from a point release in an urban like environment. Atmospheric Environment 122, 645 – 656. URL http://www.sciencedirect.com/science/article/pii/S1352231015304465*

*Kumar, P., Singh, S. K., Feiz, A.-A., Ngae, P., 2016. An urban scale inverse modelling for retrieving unknown elevated emissions with building-resolving simulations. Atmospheric environment, 140, 135-146. URL https://doi.org/10.1016/j.atmosenv.2016.05.050*

*Launder, B., 2004. Turbulence modelling of buoyancy-affected flows. In: Singapore Turbulence Colloquium.*

*Qu, Y., Milliez, M., Musson-Genon, L., &Carissimo, B. (2011). Micrometeorological modeling of radiative and convective effects with a building-resolving code. Journal of applied meteorology and climatology, 50(8), 1713-1724.*

*Sklavounos, S., Rigas, F., 2004. Validation of turbulence models in heavy gas dispersion over obstacles. Journal of Hazardous Materials 108 (1), 9 – 20. URL http://www.sciencedirect.com/science/article/pii/S0304389404000494*

*Tauseef, S., Rashtchian, D., Abbasi, S., 2011. CFD-based simulation of dense gas dispersion in presence of obstacles. Journal of Loss Prevention in the Process Industries 24 (4), 371 – 376. URL http://www.sciencedirect.com/science/article/pii/S0950423011000222*

---

## Author Comment (AC4) · 20 Jun 2018

**Response to the Reviewer #3**

The subject of this paper is challenging and very timely; certainly, we would like to know how to monitor the spread of pollutants in the urban milieu as efficiently and accurately as possible. In order to accomplish this task, the Authors derive a method based on the combination of optimization techniques, inverse tracers transport modelling and Computational Fluid Dynamics. The subject is very difficult and there are very few papers addressing the problem in a comprehensive manner; for this reason it is justified to consider publication of this study.

*We would like to thank Dr. Pudykiewicz for his careful consideration of this manuscript and for his helpful and insightful comments. We have carefully considered his comments and worked to include them in the revised version of the manuscript according to the proposed suggestions.*

*Please find below the responses to his comments.*

1) 'The Authors attempt to analyze two canonical problems: - Identification of the unknown source - Optimization of the measuring network. These two problems are mutually exclusive. Furthermore, they have different cost functionals defined on different vector spaces and, consequently, the set of control parameters is not the same for each case. This important distinction is overlooked in the paper and it is advisable to modify the text by precisely defining the functionals and the control variables.'

**Reply**: *In this study two canonical problems are considered:*

*a) Identification of the unknown source: the source term estimation STE is studied in the framework of a parametric approach using the renormalization technique. Here the challenge is to determine the parameters of the source (intensity and position) using any measurements vector (in practice the number of measurements is limited). Based on retroplumes (using sensors locations and CFD model in backward mode), we first determine the optimal renormalized Gram matrix $H_w$, for which an optimal weight function is required. This optimal weight function that verifies the renormalization condition minimize the information retrieved from the observations thus avoiding inversion artifacts close to the detectors positions. As the renormalization is a data assimilation method, the cost function to minimize is defined as the quadratic distance between the observed and the simulated measurements according to the $H_w$ norm.*

*b) Optimization of the measuring network: here, the optimization consists of selecting the best positions to be instrumented by the sensors among potential locations. This choice is operated in a space of search constituted of all possible networks (of a specific size) and based on a cost function that describes quantitatively the quality of the networks. The optimal network has the lowest quadratic distance between the observed and the simulated measurements according to the $H_w$ norm. This optimal (or near-optimal) network is obtained using the*

*Simulated Annealing (SA) algorithm. The data here are the measurements and the according sensors locations.*

*These canonical problems are coupled at each iteration during the searching process.*

*The text of the revised manuscript is accordingly modified to clarify this important point.*

2) '**The problem of optimization of the network is solved using the simulated annealing algorithm. The technique has been introduced to the computational physics over sixty years ago in the classic paper: Metropolis, N.; Rosenbluth, A. W.; Rosenbluth, M.; Teller, A. H.; and Teller, E. "Equation of State Calculations by Fast Computing Machines." J. Chem. Phys. 21, 1087-1092, 1953. Despite that the original formulation is rooted in the basic principles of physics, the reviewed paper, concerned with the network optimization, is missing the physical interpretation of the Simulated Annealing. The description of the technique can read as follows: The algorithm of simulated annealing is initiated by starting from an admissible network. At the subsequent steps, the system moves to another feasible network, according to a prescribed probability, or it remains in the current state. It is crucial to explain how this probability is calculated. The mobility of the random walk depends on a global parameter T which is interpreted as temperature. The initial values of T are large, allowing free exploration of large extents of the state space (this corresponds to the "melted state" in terms of the kinetic theory of matter). In the subsequent steps, the temperature is lowered allowing the algorithm to reach a local minimum.'**

**Reply**: *The SA algorithm is a random optimization technique based on an analogy with thermodynamics. For the SA, each network of p sensors is considered as a state of a virtual physical system, and the objective function is interpreted as the internal energy of this system in a given state. According to statistical thermodynamics, the probability of a physical system to being in a state β follows the Boltzmann distribution $P_\beta = \frac{1}{Z} exp(\frac{-\Delta E_\beta}{K_B T})$ , where Z is the partition function, $E_\beta$ is the internal energy, T is the temperature at the state β and $K_B$ is the constant of Boltzmann. By analogy, the physical quantities (temperature, energy, etc.) become pseudo-quantities and during the minimization process, the probabilistic treatment consists to accept a new network selected in the neighborhood of the current network following the probability $P = exp(\frac{-\Delta J}{T})$, where ΔJ is the cost difference between the new and the current configurations. At high temperature, the SA performs a coarse search of the space of global states, avoids local minima and finds a good minimum. As the temperature is lowered, the search becomes fine in the neighborhood of the already determined minimum and the SA reaches a better minimum.*

*As suggested, we have included this physical interpretation of the Simulated Annealing in the revised manuscript.*

3) The main characteristic of SA is relatively fast convergence but, unfortunately, it is not possible to prove that the minimum of the cost functional is global. There are several others stochastic minimization methods which can be explored; it is possible that they are potentially more applicable in the context of the monitoring of air pollutants.

**Reply**: *It is clear that for all the metaheuristic algorithms (such as the SA), it is not possible to prove that the minimum of the cost functional is global. This question is crucial for us, for this reason, we plan in the future to study the degree of confidence on the 'optimal networks'. Nevertheless, before retaining the SA as an optimization technique, we tested and compared the results obtained by Genetic Algorithm (GA) and SA based on the normalized error cost function (Kouichi, 2017). These algorithms of different search technics (SA probabilistic & GA evolutionary) are evaluated for the reconstruction of a source in a wind tunnel (DYCE experiment (Lepley et al., 2011). The optimization consisted in selecting the best positions for sensors implantation among 27 potential positions scattered in the Wind Tunnel. The results show that the optimal networks retained by the GA and the SA are quantitatively (figure 1) and qualitatively (figure 2) comparable. The errors in source parameters estimation by using the optimal networks of 3 to 13 sensors are presented in figure 1 below. The SA has advantageous because it is relatively easy to implement and takes smaller computational time in comparison to GA. Both SA and GA optimization algorithms in the framework of our approach (based in the renormalization theory) has little influence on the estimation of the parameters of a source.*

[Figure]

Figure 1: Error of source parameters estimation for (a) SA (b) GA in the DYCE wind tunnel experiment. Here m is the number of sensors, $E_l$ and $E_q$ respectively denote the location error (m) and the ratio of the estimated to true source intensity

[Figure]

Figure 2: Optimal networks (m = 3, 6, 9 and 12) obtained by (a) Simulated Annealing SA and (b) Genetic Algorithm GA

4) The problem of selection of the initial admissible network and the role of stratification should be discussed. It is well known that the flow around and above complicated structures is characterized by a complex topology. After some analysis of the literature, I'm convinced that the solution of the network optimization depends strongly on the flow Froude number. The relevant information on the flow in the vicinity of a structure is discussed in the literature, please see for example https://link.springer.com/article/10.1007/s10652-016- 9470-3. It would be interesting to present some figures describing both wind and potential temperature fields from the CFD model used in the study.

**Reply**: *The initial admissible network is selected following the trends of location error ($E_l$) and ratio of the estimated to true source intensity ($E_q$) with the number of sensors from 4 to 16 are performed and the results are presented in Kouichi (2017). As already mentioned in the manuscript, the number of sensors in the optimized networks were reduced to 1/3$^{rd}$ (13 sensors) and 1/4$^{th}$ (10 sensors) of the total number of sensors (40) originally deployed because for some cases a small number of sensors could not allow to correctly reconstruct the source and divergences of the calculations have been noted. As an example for Trial 14, reconstructing the source by using a small number on sensors is not appropriate since 4, 5, or 6 sensors are not enough ($E_l$> 100 m and log($E_q$) > 10E+17). Also, after a certain number of sensors in the network, the source term estimation is not improved significantly (see Figure 3). Thus, selecting 10 (1/4$^{th}$) and 13 (1/3$^{rd}$) number of sensors in the optimal networks ensures an acceptable estimate of the source for all the trials. These points are more clearly discussed in the revised manuscript.*

[Figure]

Figure 3: Errors in the estimation of the source (a) position and (b) intensity in Trial 14. Here p is the number of sensors

*The atmospheric stability effects in the CFD model fluidyn-PANACHE were included through the inflow boundary conditions. We had already analyzed the importance of the proper inflow boundary conditions for wind and turbulence variables on forward and backward atmospheric dispersion in an urban area (Kumar et al., 2015). Accordingly, Gryning et al. (2007) wind profile and an approximate analytical solution of the one-dimensional k-ε prognostic equation (Yang et al., 2009) for the turbulence profiles were used for inflow*

*boundary. Gryning et al. (2007) wind profile is composed of the three different length scales in the surface, middle, and upper layers of the atmospheric boundary layer (ABL), and is applicable in the entire ABL. It was also noted that Gryning's wind profile is not applicable in the trials (number 4, 5, and 6 of the MUST field experiment) of very stable atmospheric conditions. Thus, a wind profile based on the similarity function proposed by Beljaars and Holtslag (1991) was used in these trials. The Monin-Obukhov similarity theory-based logarithmic temperature profile was used to describe its vertical variation in neutral and stable conditions in the MUST field experiment. Since the coefficients in approximated analytical profiles of k and ε are estimated by fitting the observed values of k, the turbulence profiles follow the actual representation of k in each trial of the MUST experiment (Kumar et al., 2015).*

*More generally, in pollutant dispersion problems, when a proper level of turbulence intensity is important at the upwind side of the obstacles, the commonly used techniques are based on setting up simplified forms of inlet TKE (Santos et al., 2009), dynamical recycling (Tomas et al., 2015) or smooth inflow with generic downwind roughness elements (Tomas et al., 2016). Such conditions mostly affect the intensity of vertical mixing and the rate of boundary layer growth, decisive factors in determining concentrations of pollutants emitted within the urban canopy (Korycki et al., 2016). We think that these effects are beyond the scope of this work and could be further explored for the future. These discussions are now included in the revised manuscript,*

*We also present in the revised manuscript figures describing wind fields from the CFD model for the trials 4, 11 and 19. As an example, in figure 4 below is showed the wind velocity vectors around some containers for the trial 11.*

[Figure]

Figure 4: Wind velocity vectors for the trial 11

**References**

*Beljaars, A., Holtslag, A., 1991. Flux parameterization over land surfaces for atmospheric models. J. Appl. Meteorol., 30 (3), 327-341, http://dx.doi.org/10.1175/1520-0450(1991)030<0327:FPOLSF>2.0.CO;2*

*Gryning, S.-E., Batchvarova, E., Brmmer Jr., B., Gensen, H., Larsen, S., 2007. On theextension of the wind profile over homogeneous terrain beyond the surfaceboundary layer. Bound. Layer Meteorol. 124 (2), 251-268. http://dx.doi.org/10.1007/s10546-007-9166-9*

*Korycki, M., Łobocki, L., Wyszogrodzki, A., 2016. Numerical simulation of stratified flow around a tall building of a complex shape Environ Fluid Mech, 16, 1143–1171, DOI 10.1007/s10652-016-9470-3*

*Kouichi, H., Jul. 2017. Sensors networks optimization for the characterization of atmospheric releases source. Thesis, Université Paris Saclay, France. URL https://hal.archives-ouvertes.fr/tel-01593834*

*Kumar, P., Feiz, A.-A., Ngae, P., Singh, S. K., Issartel, J.-P., 2015. CFD simulation of short-range plume dispersion from a point release in an urban like environment. Atmospheric Environment 122, 645 – 656.*
*URL http://www.sciencedirect.com/science/article/pii/S1352231015304465*

*Lepley, J. J., Lloyd, D., Robins, A., Wilks, A., Rudd, A., Belcher, S., (2011). Dynamic sensor deployment for the monitoring of chemical releases in urban environments (DYCE), Proc. of SPIE, 8018(12), 1-11, doi:10.1117/12.883373.*

*Santos, J.M., Reis, N.C., Goulart, E.V., Mavroidis, I., 2009. Numerical simulation of flow and dispersionaround an isolated cubical building: the effect of the atmospheric stratification. Atmos Environ43:5484–5492. doi:10.1016/j.atmosenv.2009.07.020*

*Tomas, J.M., Pourquie, M.J.B.M., Jonker, H.J.J., 2015. The influence of an obstacle on flow and pollutantdispersion in neutral and stable boundary layers. Atmos Environ 113:236–246, https://doi.org/10.1016/j.atmosenv.2015.05.016*

*Tomas, J.M., Pourquie, M.J.B.M., Jonker, H.J.J., 2016. Stable stratification effects on flow and pollutant dispersion in boundary layers entering a generic urban environment. Bound Layer Meteorol 159(2):221–239, https://doi.org/10.1007/s10546-015-0124-7*

*Yang, Y., Gu, M., Chen, S., Jin, X., 2009. New inflow boundary conditions formodelling the neutral equilibrium atmospheric boundary layer in computationalwind engineering. J. Wind Eng. Industrial Aerodyn. 97 (2), 88-95.*
*http://www.sciencedirect.com/science/article/pii/S0167610508001815.*

---

## Author Response (AR2)

**Reply to the Editor's Comments**

The paper "Optimization of an Urban-like Monitoring Network for Retrieving an Unknown Point Source Emission" by Hamza Kouichi et al. has been re-assessed in light of the accepted Ngae et. al (2019) manuscript and the previous published articles Kumar et al. (2015a, 2015b, 2016). The current case was discussed with other topical editors and also with executive editors in addition to referee opinions.

The paper presents a further analysis of the renormalisation technique to the MUST dataset using the CFD model fluidyn-PANACHE (already published). The main result consist in reducing the network size form 40 to 13 and 10 sensors.

The points yet to be addressed belong to four categories:

1 - Technical: pertinence of using the same measurements for optimisation and validation and other minor points
2 - Originality of the results, analysis and conclusions.
3 - Applicability in practice of the results, analysis and conclusions
4 - Code description and availability

**Reply:** *We would like to thank the editors for carefully reading our manuscript and for giving constructive comments in light of our precedent papers and in addition to the referees' opinions. Indeed, this work is a continuation of our previous studies and through which we have tried to provide answers to the questions raised by the editor and reviewers. We have revised the text to include these remarks and made the manuscript more focused on the problem of reducing the size of an existing monitoring network. We have also changed the title of the paper which clearly explains the problem of reducing the size of an existing network. Each of these points is addressed in responses to your following comments.*

One reviewer has a point that has been partially answered by the authors, namely that the optimisation evaluation is done using the MUST set of measurements and this makes it more likely that the resulting sensor configuration performs well reconstructing the source (that "the same measurements shouldn't be used for the optimisation and for the reconstructions"). A robust reply to this criticism is the core result of the QJRMS manuscript, that can therefore not be used as a reply in the context of the current review.

**Reply:** *First of all we wish to clarify that the core results recently published in the QJRMS are not exactly the reply to the criticism of this paper. As also mentioned in the responses to comments of the previous reviewer and already cited in this gmdd paper, this work including the results in the QJRMS paper were parts of the PhD thesis of the first author of this paper Hamza Kouichi (available online on:* https://www.biblio.univ-evry.fr/theses/2017/2017SACLE020.pdf*). So the framework of the research problem published in the QJRMS paper was already independently developed and the results were available even before submitting any of these manuscripts. As the subject is very complex and*

*there are very few papers addressing the problem in a comprehensive manner, now we wish to clarify why we dealt with these two research problems separately.*

*In order to develop a robust methodology for the complex problem of optimal sensor placements, the first idea was to utilize the information available from the concentration measurements from a larger network to reduce its size. Similar information has been previously utilized by some researchers while dealing with the optimization problem over flat terrains by developing different approaches. For this purpose, in this first study, we developed and implemented this new methodology for the optimal sensor placement in an urban-like environment by combining the optimization techniques, inverse tracers transport modelling, and Computational Fluid Dynamics. Results from this study were encouraging as we have been able to reduce the number of sensors by 1/3rd and 1/4th of the original network with almost the same level of source detection ability as the original larger number of samplers. And this was an important step to the solution of a complex problem of the optimal sensor designing, especially in the urban-like environment. The real-world applications of this independent research problem are discussed in response to one of your next comment.*

*In order to apply the above-defined research problem for more general practical application, it was noted that in this study we utilized the concentration measurements in the optimization process to determine the optimal configurations of the networks. However, a priori information about the concentration measurements may not be available in some practical applications for the deployment of the sensors in an optimal way. In order to relax this limitation of dependency on prior concentration measurements in some practical applications, in the QJRMS paper we tried to develop another methodology for determining an optimal sensors network using only the available meteorological conditions. The research problem in the QJRMS paper was designed for different application point of view and is independent of the methodology presented in this study.*

*We partially agree with the reviewer and editor that the optimisation evaluation is done using the MUST set of measurements and this makes it more likely that the resulting sensor configuration performs well reconstructing the source (that "the same measurements shouldn't be used for the optimisation and for the reconstructions"). However, this doesn't limit the application of the proposed methodology for some important practical applications like the accurate emissions estimation. In fact, this is a limitation of the MUST data for this application as for a complete process of the designing and the evaluation one requires a sufficiently long set of measurements so that the whole data can be divided into two parts: (i) first part for the designing an optimal sensor network and (ii) the second part for the evaluation of the designed optimal network. However, the durations of the releases in the MUST field experiment were not sufficiently large to divide the whole data from a test release separately into two parts for designing the optimal sensor network and then its evaluation. However, in further evaluation of the resulting optimal sensor configuration, a different set of the concentration measurements can be constructed by adding some noise to the measurements. For a continuous release in steady atmospheric conditions, the average value of the steady concentration in a test release is not expected to deviate drastically from the mean values in each segment of the complete data. So this new set of the concentration*

*measurements with added noise can partially fulfil the purpose of the evaluation of a designed optimal network.  As shown in Table 2, the errors in the estimated source parameters are small even with the new sets of concentration measurements constructed by adding 10% Gaussian noise. We have also evaluated the obtained optimal networks with the other two sets of the concentration measurement, which are generated by adding 15% and 20% random noise to the original concentration observations. The source parameters are still estimated with reasonably good accuracy for these two scenarios. This whole exercise shows that even if we have utilized a different set of the measurements for the evaluation of the optimal networks, the optimal networks have almost the same level of the source detection ability in an urban-like environment.*

Some additional reviews considered that the distinct novelty of the manuscript under consideration in not put forward clearly enough with respect to the manuscripts Ngae et. al (2019) and Kumar et al. (2015a, 2015b, 2016). On the other hand, the scope can be adjusted in underlying that the current work addresses only part of the problem. In this case the text has to be adapted accordingly. The current presentation (and the title) suggests a very general treatment of the problem. This is contrasts with the fact that part of the results seem to be published in Ngae et. al (2019) and Kumar et al. (2015a, 2015b, 2016). In general it has to be highlighted so it becomes immediately clear to an average reader what is new in this manuscript with respect to the other two.

**Reply:** *It should be noted that the earlier studies mentioned by the editor were conducted for different research problems only for (i)  to study the forward dispersion in an urban-like environment using a CFD model by utilizing the MUST diffusion experiment (Kumar et al., 2015a) and (ii) describing methodologies to utilize the renormalization inversion technique with combination of CFD method for localizing and quantifying an unknown ground level or elevated continuous point sources in an urban-like environment (Kumar et al., 2015b, Kumar et al., 2016). However, as editor has also pointed out, that the aim of this study is to demonstrate how the renormalization inversion technique can be applied to optimally placing a smaller number of concentration samplers for quantifying a continuous point source in an urban-like environment with almost the same level of source detection ability as the original larger number of samplers. For this purpose, in this study, we proposed a methodology for designing the optimal monitoring network in an urban-like environment, which is based on the combination of optimization techniques, inverse tracers transport modelling, and Computational Fluid Dynamics. This attests to a new development of a methodology to optimally design the sensors monitoring network in an urban-like environment. The novelty of the manuscript under consideration with respect to the manuscripts Ngae et. al (2019) is already explained in the response to your previous comment. As suggested, we have modified the text accordingly in the revised version of the manuscript in order to adjust the scope and we highlighted that the current work addresses only part of the problem. Title of the paper is now changed.*

Otherwise, it may be put forward how in the view of the authors the results and conclusions can be applied in practice.

**Reply:** *This study presents a practical method for managing realistic situations. This study shows that it is possible to reconstruct a source of atmospheric emissions with a limited number of concentration measurements and presents a methodology for selecting the 'best' sensors positions based on an optimality criterion. The applications of the proposed method for the optimization of a sensor network with limited numbers can be very useful and demanding in many real-world problems. For example, in oil and gas industries, estimation of the emissions of the greenhouse gases (GHG) like methane ($CH_4$) is a challenging problem. In order to utilize an inversion method to estimate the $CH_4$ emissions, accurate measurements of the methane at a network of the high precision sensors, downwind of a possible source, is a prerequisite. However, in order to obtain these accurate $CH_4$ measurements, the cost of high precision methane sensors like Cavity Ring Down Spectroscopy/Spectrometer (CRDS) or similar other sensors is currently so high that we cannot deploy these sensors in a large number. However, alternatively, we can obtain the initial measurements by deploying a sufficiently large number of low-cost sensors (which may not be as high precision as CRDS or others) on a large monitoring network. Using these less accurate $CH_4$ measurements and the proposed optimization methodology in this study, we can quickly design an optimal network, which provides the 'best' sensors positions with the reduced number of sensors. Then, high precision sensors can be deployed on this obtained optimal network to measure the accurate $CH_4$ measurements. These concentration measurements can be utilized in an inversion method to estimate the accurate $CH_4$ emissions. A similar and very useful application of the method proposed here can be applied for the estimation of the methane emissions from landfills. We have revised the text and given more explanation regarding the usefulness of the methodology and its practical applicability.*

Remember that the article cannot be published if the authors do not make their code available. There appears to not even be a code availability section. This is not only referring to the CFD code, which is not the core of the article and has been published before. The authors have to make the pieces of code used in the core results, including the renormalisation and the optimisation. If possible, the scripts used for preparing the figures presenting the main results have to be provided as well. The executive editors clearly state: "The paper must be rejected if the authors refuse to comply with requests to make the code accessible within the requirements of GMD."

**Reply:** *We should mention here that a large part of the code was already made available at the time of revision as a compressed folder in the supporting information. We have further improved the readability of the code by introducing detailed comments.   The updated full version of the code is now made available along with the manuscript. The MATLAB version of the code is executable for coupling between the optimization algorithm (SA) (the renormalization algorithm) and the CFD retroplumes calculated for a sample test trial 14 of the MUST field experiment.*

I suggest the authors to revise the manuscript in a way that makes the criticisms irrelevant. This includes reworking the introduction (and also the discussion) narrowing the scope in order to describe more accurately what the paper delivers and how it is related to the previously published works. I suggest to explain and discuss more explicitly the results and

conclusions of the works previously published by the group (i.e. Ngae et. al (2019) and Kumar et al. (2015a, 2015b, 2016).) and precise the relationship of the results and conclusions presented in the current manuscript.

**Reply:** *As suggested, we have revised the text accordingly. Most of these papers are already being referred in the previous version and now we have also described our recent paper Ngae et. al (2019) and how it is related to the current study.*

The main outstanding point is however the code availability. The paper cannot be published without it. A possible way forward in the context of GMD is to put the accent in the description of the code (and reproducibility) as it seems that none of the previously published works the code is sufficiently documented.

**Reply:** *The code is now available with detailed comments for more comprehension.*

minor/specific points:

p 1 l1 "sensors measuring the polluting substances" > sensors measuring polluting substances

**Reply:** *Corrected*

l2 "environment with a view to estimate an unknown" > environment in order to estimate an unknown

**Reply:** *Corrected*

l3 " The methodology was presented by coupling the" > The methodology is presented by coupling the optimal network was analyzed by > optimal network is analyzed by
**Reply:** *Corrected*

l10 "In 80% trials, emission rates with the 10 and 13 sensors networks were estimated within a factor of two which are also comparable to 75% from the original network." this phrase is not clear enough for the abstract, please clarify.

**Reply:** *The phrase is modified in the revised version for more clarification.*

l11 The last phrase: "This study presents an application of the renormalization data-assimilation theory for determining the optimal monitoring networks to estimate a continuous point source emission in an urban-like environment." is background/introduction information, and should be combined with the first-second phrase.

**Reply:** *Modified accordingly.*

l17 "by the concern authority." The concerned authorities?

**Reply:** *Corrected.*

p 2 l1 "The problem to optimize a monitoring network is common and consists in reducing the size of a network of sensors at the level of a city, county, or a neighborhood while retaining its properties." This statement contrasts with the following aspects of monitoring

networks optimization: First deployment, Updating an existing network, reducing the size of an existing network, increasing the size of an existing network.

**Reply:** *The text is revised to avoid the contrast and clarification are included.*

Please clarify specifying the actual contribution of this work in the general context.

**Reply:** *The actual contribution of this work in the general context is clarified in the revised version.*

"This study will be focused with an objective to reconstruct an unknown continuous point source's release in an urban-like environment." This contrasts with the stated narrower focus of reducing an existing network of sensors for source location and intensity estimation. This observation is relevant because the inverse methodology was already introduced in Ngae et. al (2019) and Kumar et al. (2015a, 2015b, 2016). These papers are about inverse modelling on fluidyn-PANACHE fields in the MUST case applying the renormalisation inversion technique. It has to be clear from the beginning what is the new contribution of this work that adds to what has already been published or has been submitted and accepted (namely the network reduction?).

**Reply:** *The text is revised to avoid the contrast and clarification are included.*

l6 "an optimally monitoring sensors network" something seems missing in this grammatical construction.

**Reply:** *Corrected.*

l9 "This study presents a methodology for the sensor's locations choice, leading to the best network for the estimation of 10 an unknown point point source in a geometrically complex urban environment." You have to

**Reply:** *This phrase is revised and modified for more clarification.*

The "the" article in "The SA algorithm was designed for the statistical physics" is spurious > "The SA algorithm was designed in the context of statistical physics?"

**Reply:** *Corrected.*

p3 l13 "advanced search algorithm*s* like genetic algorithm "

**Reply:** *Corrected.*

l18 replace "..., etc." with a concrete reference or omit.

**Reply:** *Omitted.*

p3 l8 " This study deals with a case of reducing the number of sensors in order to obtain an optimal network from an existing network." This sentence is key, because it describes precisely what is done in the study. This information has to be clear much earlier and in the abstract.

**Reply:** *This information is more highlighted in the revised manuscript and in abstract.*

Sections 2 and 3: provide the working code. This is a fundamental GMD requirement.

**Reply:** *The code is now available with detailed comments for more comprehension.*

Section 5: You don't have to provide the CFD code if it has been published elsewhere, but if the code described in 2 and 3 requires sample input files, those have to be included, at least for testing purposes.

**Reply:** *The code is now available and executable for the CFD retroplumes for the trial 14.*

Section 6: Ideally scripts for the results should be provided for reproducibility.

**Reply:** *The code is now available.*

**References**

Kumar, P., Feiz, A. A., Ngae, P., Singh, S. K., & Issartel, J. P. (2015a). CFD simulation of short-range plume dispersion from a point release in an urban like environment. Atmospheric Environment, 122, 645–656. http://doi.org/10.1016/j.atmosenv.2015.10.027

Kumar, P., Feiz, A. A., Singh, S. K., Ngae, P., & Turbelin, G. (2015b). Reconstruction of an atmospheric tracer source in an urban-like environment. Journal of Geophysical Research, 120(24), 12,589-12,604. http://doi.org/10.1002/2015JD024110

Kumar, P., Singh, S. K., Feiz, A. A., & Ngae, P. (2016). An urban scale inverse modelling for retrieving unknown elevated emissions with building-resolving simulations. Atmospheric Environment, 140, 135–146. http://doi.org/10.1016/j.atmosenv.2016.05.050

Ngae, P., Kouichi, H., Kumar, P., Feiz, A.-A., & Chpoun, A. (2019). Optimization of an urban monitoring network for emergency response applications: An approach for characterizing the source of hazardous releases. Quarterly Journal of the Royal Meteorological Society. http://doi.org/10.1002/qj.3471

**Reply to the Reviewer#3**

The paper addresses optimization of a reduced set of receptors among an established arrays of detectors in an urban like monitoring network for retrieving an unknown point source emissions. This task is achieved here by coupling a simulated annealing algorithm, an inversion technique and a CFD model. The study claims to propose an optimal determination of monitoring network, however, this optimality is never achieved. The reallocated network structures are biased towards the source parameters. This is also argued by the previous reviewers which is still not addressed in the revised version. Moreover, there are several instances where results are simply stated without clarification (only some are referred in my comments).

**Reply:** *Please find below a point by point response to the reviewer's comments. We are grateful for the reviewer's comments and suggestions and appreciate the efforts he/she has made to improve the quality of the manuscript.*

1. Technically, the fundamental limitation of this study lies in the fact that they did not treat the two problems "design of reduced network" and "point source estimation" independently whereas the two problems are mutually exclusive. Same cost function is minimized for both the problems which gives a biased estimate of source parameters.

**Reply:** *We agree with the reviewer's remarks that in a network optimization problem, finding an optimal rearrangement of a set of receptors and then, point source estimation are two independent sets of problems. In fact, we also followed the same procedure. The choice of the optimal sensors network was not determined based on source estimation. The network optimization problem was independently presented and performed before any evaluation by estimating the point source parameters using the measurements from the sensors in the obtained optimal network. It is very clearly explained in the flow diagram of the methodology in Figure 2 and shows that a point source was estimated only when we obtained the optimal monitoring network.  In this work, the first step is to find the best configuration of a limited set of sensors using the meteorological data, the positions of the existing sensors on the instrumented area, a CFD technique and the concentration observations. The second step is to evaluate a posteriori the performance of the optimal networks in comparison with the original network used in the MUST field experiment.*

*Also, in order to support the utility of the proposed optimization methodology in current framework, here we also provide an example of an important practical application. In the oil and gas industries, estimation of the emissions of the greenhouse gases (GHG) like methane (CH4) is a challenging problem. In order to utilize an inversion method to estimate the CH4 emissions, accurate measurements of the methane at a network of the high precision sensors, downwind of a possible source, is a prerequisite. However, in order to obtain these accurate CH4 measurements, the cost of high precision methane sensors like Cavity Ring Down Spectroscopy/Spectrometer (CRDS) or others are currently so high that we cannot deploy these sensors in a large number. However, alternatively, we can obtain the initial measurements by deploying a sufficiently large number of low-cost sensors (which may not be as high precision as CRDS or others) on a large monitoring network. Using these less accurate CH4 measurements and the proposed optimization methodology in this study, we can quickly design an optimal network which provides the 'best'*

*sensors positions with the reduced number of sensors. Then, high precision sensors can be deployed on this obtained optimal network to measure the accurate CH4 measurements. These concentration measurements can be utilized in an inversion method to estimate accurate CH4 emissions. A similar and very useful application of the method proposed here can be applied for the estimation of the methane emissions from landfills.*

*Here the low-cost sensors can rapidly be deployed specifically for collecting the information (i.e., measurements) to be used for a specific need (neutralize the source, refurbishment an installation on industrial site, etc.). In this case, the meteorological conditions (as wind speed and direction, etc.) can be known in real time from the available observations or from numerical weather forecasting models and can be assumed as stationary. The optimization, in this case, can be performed in real time if the interesting area is not complex and the calculation can be conducted quickly in a very short time (using Gaussian model and an optimization algorithm for example) using the measurements from low-cost sensors. If the domain is complex (i.e., contains several obstacles), CFD model should be used to include the effect of the obstacles. However, for an area of interest in a complex urban or industrial environment, an archive database of the CFD calculations can be established for a wide range of meteorological and turbulence conditions and can be utilized in the optimization process.*

2. In the paper, the minimization of cost function (section 3.1) and renormalization process (section 2.2 and 2.3) refers to the same estimate for a point source estimation. So, there is no point in presenting these two as different methods and different optimization. This has already been mentioned in several papers, for example. Sharan et al. (2009), Sharan et al. (2012) or Issratel et al. (2012). The authors have already cited these papers.

**Reply:** *We sincerely acknowledge reviewer's concern  that the mathematical illustration of the optimization techniques utilized in the present study is already presented in the earlier studies. . However, we are of the view that in order to present the optimization methodology, the presentation of a brief mathematical formulation of the renormalisation technique is necessary.*

*The optimality criterion cannot be defined in a consistent manner without presenting  the origin and physical significance of the optimization algorithm. If we present directly the adequate cost function (i.e. normalized errors) this cannot be appropriate for readers that don't have any information about the renormalisation method. Further,  the detail of point source estimation is presented in the previous version of the manuscript    due to the fact that this method has been utilized for the evaluation analyses (i.e. performance in source parameters estimation). However, as suggested by the Reviewer, we have further made an attempt to reduce the size of these sections considerably in the updated manuscript. In the revised manuscript section 3.1 is completely moved to Appendix A.*

3. Mathematically, reducing the set of measurements is critical in ill-posed problems since each measurement is a significant entity or information while estimating the characteristics of unknown

space or parameters. In the paper, authors did not analyze how their total measurement information is varying or reducing with their subjective choice of the receptors. The accuracy or closeness of the source parameters can not be the only criterion to believe in a reduced set of network. Note that reducing the measurements will increase the degree of freedom in space and such solutions will be prone to the noise in the model, their variables and measurements. Such issues are never analyzed or taken into account here.

**Reply:** *We agree with the reviewer's remark about the effects of reducing the set of measurements in an ill-posed problem while estimating the characteristics of unknown parameters. As the reviewer has also pointed it out that reducing the measurements will be prone to the noise, we have performed a posterior analysis of the estimated parameters from the noisy measurements. As shown in Table 2, the errors in the estimated source parameters are small even with the new sets of concentration measurements constructed by adding 10% Gaussian noise to the original measurements. We have also evaluated the obtained optimal networks with two more sets of the concentration measurements, which were generated by adding 15% and 20% random noise to the original concentration observations. The source parameters are still obtained with reasonably good accuracy for these two scenarios. This whole exercise shows that even if we utilize a noisy set of the measurements for the evaluation of the optimal networks, the optimal networks would have almost the same level of the source detection ability in an urban-like environment.*

4. The exact convergence of the simulated annealing algorithm can not be guaranteed. Especially, in case of ill-posed problems, it is highly probable in different simulations to produce several sets of reduced receptors of same size that can have minimum of the cost function. In this case, uniqueness of the selected set of reduced network can be challenged. The paper never discussed such issues which are more common in case of ill-posed problems.

**Reply:** *Throughout the text, we mentioned this issue that there is no guarantee in the convergence of the SA and we confirmed (based on the adequate bibliographical references) that the obtained network can be the optimal or the near-optimal one. This complex combinatorial optimization approach retained big attention in the literature and the SA is selected following the recommendations from more than one research work of networks optimization in the framework of the atmospheric dispersion context (Abida et al., 2008; Abida and Bocquet, 2009; Jiang et al., 2007; etc.). Nevertheless, before utilizing the probabilistic algorithm SA, we tested its performance in comparison with the Genetic Algorithm GA of evolutionary research technique (Kouichi, 2017). Concerning the statistical study after the achievement of the optimization, we plan to perform this investigation as continuity of this first study. Some of these issues are already discussed in the manuscript and we again discussed in the revised manuscript.*

5. In computation, the computation of weight matrix and gram matrix depends on the number of measurements utilized during the inversion process. Accordingly, these two matrices seems to be changing with each iteration of the simulated annealing algorithm. It seems that the weight matrix is a priori information introduced in the space however, this will be reduced (or at least vary

drastically) with the reductions in the number of measurements. How their variation is affecting the retrieval of source vector or point source location is never explored or discussed here while they are mentioned important in cost function minimization.

**Reply:** *As mentioned before in response to one of the comments from the reviewer, we did not estimate the point source parameters during the optimization process of a monitoring network. The choice of the optimal sensors network was not determined based on source estimation. So it was not necessary to analyze the effect of variation of the weight matrix or Gram matrix on the point source location during the optimization process. To analyze the effect of weight matrix or Gram matrix on the estimation of a point source location is a completely different exercise. This doesn't need an optimization process and can simply be verified by taking the different number of measurements in the source inversion problem. It should be mention here that source estimation was not the main objective of this study and it was conducted only to evaluate the performance ability of the obtained optimal networks.*

6. The study do not bring any significant outcome in terms of methodology, optimality criterion or source localization features. Their discussion is mostly similar to a source estimation study like in Kumar et al. (2015b). In addition, it does not provide any insight related to their sensitivity of source localization with respect to the difficulties faced in the real urban scenarios, model errors / uncertainties, meteorological variability, etc.

**Reply:** *This point was already responded in detail to one of the comment of Dr. Sarvesh Kumar Singh on the previous version of the manuscript. If we just leave aside the optimization problem, even source reconstruction in a complex urban environment is itself a very complex problem. We think that addressing a far more complex problem than the source estimation in an urban area by combining concepts from an inversion technique, CFD, and optimization algorithm in a comprehensive manner is itself a significant contribution. As responded before, in the present study, the obtained optimal networks were analyzed qualitatively and quantitatively for all the trials of the MUST field experiment. The dispersion of the sensors in the urban-like environment was critically analyzed according to the source position and the meteorological conditions. A fine analysis is performed to highlight the common structures in the optimal networks. Also, a posteriori study is realized in order to evaluate the performance of the optimal networks. For this, the errors in source parameters estimation are compared with the errors obtained from the original network which leads to the important conclusions in networks size reduction in the framework of source reconstruction in an urban environment. As the applicability of the obtained monitoring networks is validated and analyzed by estimating the source parameters from the concentration measurements from the optimal networks, it is obvious to present and analyzed the source reconstruction results and compare these with the one obtained in the previous study. We do not agree with the reviewer's point that this study does not highlight any significant contribution related to optimal networking. In fact, using the proposed methodology, we were accurately able to estimate the source parameters using the measurements only from 1/4th and 1/3rd sensors with approximately similar accuracy compare to the network of the original number of sensors. This is a significant contribution that reduces the number of sensors in a complex urban-like environment*

*and without compromising the ability of the network with a minimal number of sensors to estimate an unknown source. As discussed before in response to the reviewer's first comment, from application point of view this method can be very useful and demanding for the accurate methane emissions estimation in oil and gas industries and also from landfills.*

7. Page 15, line 20. The sentence "This tendency makes it possible …." is not clear. The tendency to reallocate the sensors is prior to the knowledge of source location and depends on retro-plumes or mainly flow and dispersion characteristics. So, such tendencies can never be explained in terms of their proximity to the source location. In lines 21-26, the paragraph "The visibility function includes ..." is repetition about the visibility function from previous papers without bringing any new conclusions.

**Reply:** *This sentences are either modified for more clear presentation or removed in the revised manuscript.*

8. Page 15, line 16, why larger location errors do not necessarily correspond to high intensity errors. This is not obvious. Another example is in line 26, while saying "visibility functions have significant levels." What does it mean?. These are just few examples where results are stated without exploring the reasoning behind them.

**Reply:** *Here we have just presented a summary of the results from the evaluation exercise. These or similar sentences are modified and clearly explained in the revised manuscript.*

9. Page 16, lines 25-28. I do not see "conditions for near overall optimum" in this paper and how one or more optimal network can satisfy it just by having a common set of few sensors or skeletons.?

**Reply:** *Modified in the revised version.*

10. Similar to comment #7, Page 16, lines 29- 32. The explanations given here do not signify anything related to the uncertainties of the network structures. The table 2 represents uncertainties for source parameters not for the network structures.

**Reply:** *The discussion is modified for more clarity.*